# TNF alpha unmasks enteric malate aspartate shuttle dysfunction bridging Parkinson disease and intestinal inflammation

Bruno Ghirotto [1,2], Luís Eduardo Gonçalves[3], Vivien Ruder [1], Christina James [1], Elizaveta Gerasimova[4], Tania Rizo[1,15], Holger Wend[1], Michaela Farrell[1], Juan Atilio Gerez [5], Natalia Cecilia Prymaczok[5], Merel Kuijs [6,7,8], Maiia Shulman[6,7,8], Anne Hartebrodt[9], Iryna Prots [4], Arne Gessner [10], Michael Vieth[11], Friederike Zunke [12], Jürgen Winkler [12], David B. Blumenthal [9], Fabian J. Theis[6,7,8], Roland Riek [5], Claudia Günther[3,13], Markus Neurath [3], Pooja Gupta [1,16] & Beate Winner [1,13,14,16] ✉

Gastrointestinal dysfunction often precedes motor symptoms in Parkinson's disease (PD), suggesting the enteric nervous system (ENS) is central to early pathogenesis. How α-synuclein contributes to ENS dysfunction, and how inflammation modulates this, remains unclear. Here we show that Tumor Necrosis Factor alpha enhances α-synuclein accumulation in induced pluripotent stem cell-derived enteric neurons and glia, and impairs the malate-aspartate shuttle, a key pathway for mitochondrial energy production. This drives a metabolic shift toward glutamine oxidation in patient cells. This metabolic impairment reduces overall mitochondrial function, which is partially rescued by the neuroprotective compound Chicago-Sky-Blue 6B. Furthermore, transcriptomic and histological analyses of human gut tissue from inflammatory bowel disease patients reveal that inflammation-associated metabolic suppression and α-synuclein upregulation occur beyond PD, representing general hallmarks of intestinal inflammation. These findings highlight a conserved metabolic vulnerability in the ENS and establish patient-derived enteric lineages as a robust platform to model inflammatory ENS pathology.

Parkinson disease (PD) is the most frequent movement disorder. The motor deficits - hypokinesia, rigidity, and tremor - are caused by dopaminergic neuronal loss in the substantia nigra[1]. However, non-motor symptoms, particularly those affecting the gastrointestinal (GI) tract, are increasingly recognized as early and clinically significant manifestations of PD[2,3]. Chronic constipation and abdominal bloating often precede motor deficits by years[4], suggesting a role for the enteric nervous system (ENS) in early disease pathogenesis. The ENS, a vast network of over 500 million enteric neurons and glia within the enteric wall, regulates motility, secretion, and blood flow while also modulating intestinal immunity and interacting with the microbiota[5]. Its

striking similarities to the central nervous system (CNS) in terms of neuronal diversity and signaling suggest that ENS dysfunction could contribute to early GI symptoms in PD and influence disease progression.

Accordingly, a leading hypothesis suggests that in the body-first subtype of PD, pathology originates in the GI tract and spreads to the brain via autonomic nerves, including the vagus nerve[6]. Central to this hypothesis is α-synuclein (α-syn), whose misfolding and aggregation defines PD pathology[1]. While α-syn toxicity has been extensively studied in CNS[7], its impact on the ENS remains poorly understood. Histopathological studies confirmed α-syn accumulation in enteric

neurons of PD patients[8], and animal models set the foundation for better understanding of mechanisms linking ENS dysfunction to GI symptoms[9,10]. Notably, injection of α-syn epitopes in humanized mice triggered intestinal inflammation, constipation, and enteric neuronal loss, demonstrating that α-syn can directly drive ENS dysfunction and GI pathology[11].

Neuroinflammation is increasingly recognized as a key feature of PD[12,13], with cytokines playing a key role in α-syn-driven pathology[14,15]. We previously demonstrated that α-syn increases the susceptibility of iPSC-derived cortical neurons to cytokine-induced toxicity, particularly IL-17A[15], suggesting that both α-syn accumulation and inflammatory stimuli contribute to disease progression. In the CNS, proinflammatory cytokines increase α-syn-induced mitochondrial dysfunction, oxidative stress, and synaptic impairments[1,14,15]. Similarly, α-syn has been implicated in intestinal inflammation[9,11,16,17], but most studies rely on animal models, which do not fully recapitulate human pathology. While both immune activation and mitochondrial dysfunction are recognized PD hallmarks, how cytokines and α-syn interact to drive pathology in the human ENS remains unknown. Importantly, enteric neurons show high dependence on mitochondrial respiration to sustain function and viability, as shown in Hirschsprung's disease[18]. Disruptions in metabolic homeostasis may therefore amplify their vulnerability to inflammation, further aggravating ENS dysfunction. PD studies have demonstrated disease-induced alterations in mitochondrial respiration, amino acid metabolism, and energy substrate utilization[19,20], but the interplay of intestinal inflammation and neurodegeneration has not been resolved.

Studying α-syn pathology in the human ENS faces significant challenges. Unlike classical gastrointestinal diseases like inflammatory bowel diseases (IBD), GI specimens from PD patients are rarely available. ENS complexity and its unique microenvironment limit insights from histopathological studies, which confirmed α-syn accumulation[8], but not its functional impact. Additionally, interspecies differences in animal models hinder the translation of findings to human disease[21]. Recent advances in stem cell research offer a promising platform for modeling α-syn pathology in both the CNS and ENS. Induced pluripotent stem cell (iPSC)-derived neurons enable patient-specific studies[22], and protocols for generating enteric neural lineages (ENLs) from iPSCs[23] have advanced the study of Hirschsprung disease[24]. Nevertheless, this approach remains unexplored for PD.

To address these gaps, we utilized iPSC-ENLs from PD patients with α-syn gene triplications (SNCA 3x). Employing multi-omics and functional analyses, we examined whether an increase in α-syn contributes to early PD pathology in the ENS and if this process is intensified by proinflammatory cytokines. We demonstrated that SNCA 3x ENLs exhibit enhanced Tumor Necrosis Factor alpha (TNF) susceptibility, leading to increased mitochondrial α-syn accumulation. This triggered TNF-induced metabolic reprogramming, impairing the malate-aspartate shuttle (MAS) and tricarboxylic acid (TCA) cycle, thereby increasing reliance on glutamine oxidation. These metabolic shifts impaired cellular energy metabolism, leading to increased enteric neuronal vulnerability. The effect could be rescued by targeting glutamate metabolism with Chicago-Sky-Blue 6B (CSB6), a competitive inhibitor of vesicular glutamate transporters. Our data highlight iPSC-derived ENLs as a powerful platform for studying human synucleinopathies and identify cytokine signaling and metabolic dysfunction, specifically MAS impairment, as critical drivers of enteric pathology in early PD, with broader relevance to intestinal inflammation.

## Results

### iPSC-ENLs as a model for studying synucleinopathies
To study the enteric nervous system in synucleinopathies, we generated ENLs from three SNCA 3x iPSC lines and their isogenic controls (Iso)[25]. The differentiation protocol starts from iPSCs and involves the induction of neural crest cells followed by enteric lineage specification to recapitulate key developmental pathways in vitro[23]. Within the differentiation process towards ENLs we assessed the expression of marker genes at three time points: iPSC at an early stage of ENC induction (day 6), a middle time point during ENL differentiation (day 40) and full differentiation at day 70 (Fig. 1a). All iPSC lines used passed the initial quality control analysis for pluripotency markers Nanog and Lin-28A, without differences between groups (Fig. S1A). As expected, already at day 40 of ENL differentiation, Reverse Transcription Polymerase Chain Reaction (RT-qPCR) confirmed the presence of the neuronal marker *TUBB3* and enteric neuronal markers *ELAVL4*, *PHOX2B*, and *HOXB3* in both Iso and SNCA 3x lines (Fig. 1b, S1B). Similarly, ENLs showed expression of the glial marker *GFAP* at day 70, with no group-level differences (Fig. 1b, S1B). Immunostaining confirmed the presence of GFAP and the enteric neuron marker HuC/D in both groups, along with a specific increase in α-syn levels in SNCA 3x ENLs at day 70 (Fig. 1c).

Cellular diversity and gene expression patterns in iPSC-ENLs at day 70 of differentiation were additionally explored using single-cell RNA sequencing (scRNAseq). Through clustering at a broad resolution and subsequent annotation, we determined three primary cell populations: enteric neurons, glia, and proliferating cells (Fig. 1d, S1C). At this broad population level, no significant differences in abundance were observed between Iso and SNCA 3x, as assessed by scProportion test with a log2FC threshold of 1 (Fig.1e), and the complementary neighbourhood analysis using MiloR (Fig. S1D, Supplementary Data 5). Each cluster expressed its classically described canonical markers[23,26] (Fig. 1f). We also measured *SNCA* expression levels in the ENS. As expected, enteric neurons had the highest *SNCA* expression, followed by glia, and proliferating cells showed the lowest expression (Fig. 1g). Between the two conditions, SNCA 3x glia exhibited higher *SNCA* expression compared to Iso (Fig. S1E).

To highlight the broader relevance of our iPSC ENL model, we subclustered the scRNAseq enteric neurons and glia cell populations (Fig. 1d) and compared them to the previously published single-cell atlas of human gut tissue[27]. We identified 14 neuronal subclusters and 13 glial subclusters (Fig. 2a, e), with top markers associated with these subclusters depicted in Fig. S2A–B and Supplementary Data 2. Annotation using SingleR successfully mapped several distinct neuronal and glial subtypes to the reference human gut tissue (Fig. S2 and Supplementary Data 3). This included putative excitatory motor neuron 3 (PEMN3), expressing the D2 dopamine receptor and putative inhibitory motor neuron 3 (PIMN3), marked by adrenomedullin. The analysis also identified putative sensory neurons (PSN), which sense and respond to chemical and mechanical stimuli and interneurons (PIN), involved in intercellular communication (Fig. 2a and S2C–D). Most enteric glia subclusters matched the Glia 1 subtype, linked to GDNF receptor *GFRA2* expression[27] (Fig. 2e and S2F-G). Following, we merged these annotations onto our UMAP plots (Fig. 2a, e).

Overall, these results indicate that both Iso and SNCA 3x iPSC lines differentiate effectively into ENLs, comprising a mixed culture of enteric neurons and glial cells, which recapitulate the cellular diversity of the human ENS. Moreover, the presence of α-syn expression in our model highlights its relevance not only for studying synucleinopathies but also for broader ENS-related research. Crucially, the lack of significant differences between Iso and SNCA 3x at the broad population level prompted us to investigate abundance changes at the subcluster level, as detailed below.

### SNCA 3x drives mitochondrial alterations at the transcriptional level and alters the subcluster composition of enteric neurons and glia
We investigated whether increased α-syn alters gene expression and cellular abundance in enteric neuron and glial subpopulations using the scRNAseq data. Visualization of *SNCA* expression across enteric

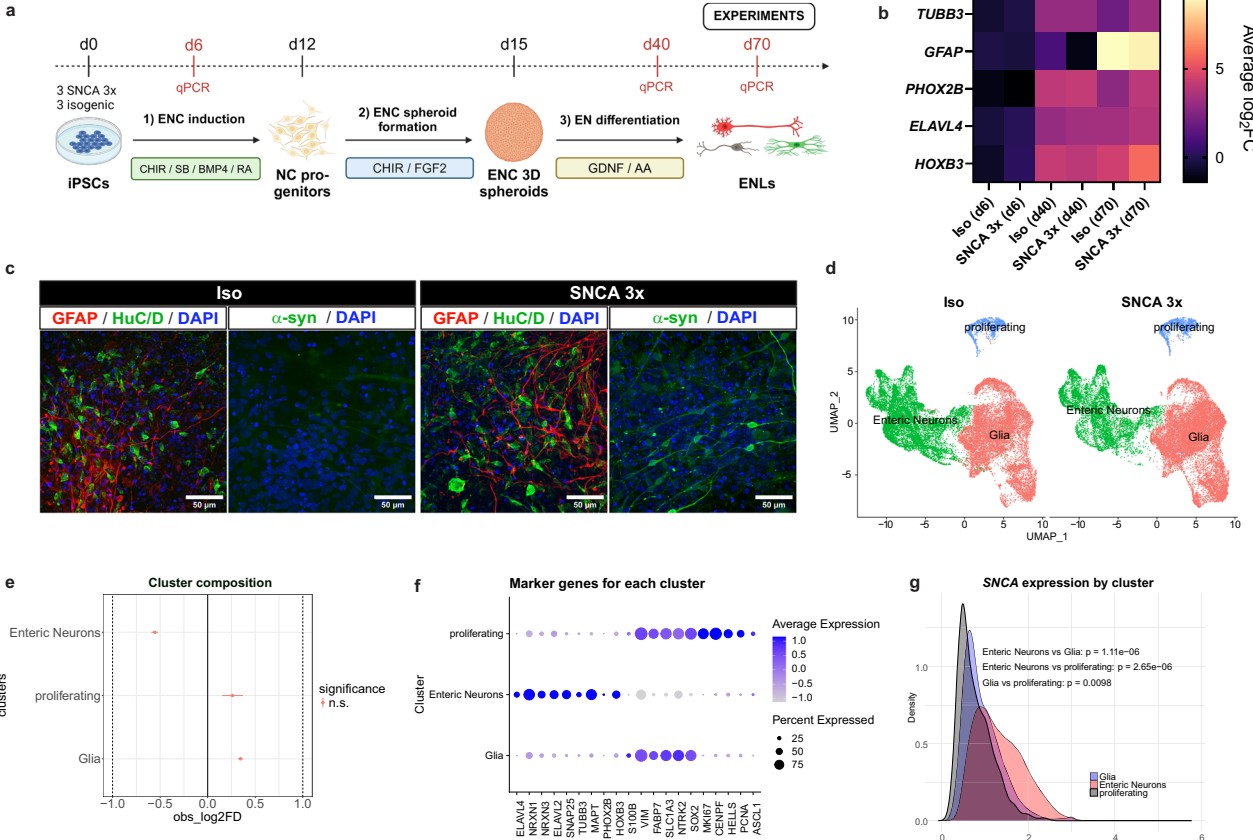

**Fig. 1 | iPSC-ENLs as a model for studying synucleinopathies. a** Paradigm illustrating the differentiation of ENLs from iPSCs. BMP4, Recombinant human bone morphogenetic protein-4; CHIR, CHIR 99021; RA, Retinoic Acid; SB, SB431542; FGF2, Recombinant Human FGF Basic; AA, ascorbic acid; GDNF, Recombinant Human Glial Derived Neurotrophic Factor; NC, neural crest; ENC, enteric neural crest; ENL, enteric neural lineages. Created in BioRender. Winner, B. (2026) https://BioRender.com/652io1g (**b**) Heatmap showing gene expression of neuronal marker *TUBB3*, glial marker *GFAP* and enteric neuronal markers *PHOX2B*, *ELAVL4* and *HOXB3* in iPSC-derived ENLs at days 6, 40 and 70 of differentiation using RT-qPCR. Log₂ fold change was calculated in relation to the Iso group at day 6 of differentiation and averaged for $n = 3$ independent SNCA 3x and 3 isogenic lines per group, from two independent differentiations. See also Fig. S1B. Source data are provided as a Source Data file. **c** Immunocytochemistry characterization of iPSC-derived ENLs at day 70 after start of differentiation. Panels show the glial marker GFAP (red), the enteric neuronal marker HuC/D (green), α-syn (green, 2A7

antibody, Novus, Cat#NBP1-05194) and DAPI (blue) for cell nuclei staining for both Iso (left) and SNCA 3x (right). Scale bar = 50 μm. Figure is representative of two independent experiments. **d** UMAP plots obtained from scRNAseq analysis of Iso and SNCA 3x ENLs at day 70 after start of differentiation, separated by the clusters identified after annotation. **e** Compositional analysis plot showing the absence of significantly changed clusters between Iso and SNCA 3x. Data are presented as observed log₂ fold change (center points) +/− 95% confidence intervals (error bars). Confidence intervals were calculated using bootstrap resampling. *P*-values were determined via permutation testing and adjusted for multiple comparisons using the Benjamini-Hochberg method (adj. *p*-value). FDR=False Discovery Rate; log2FD=log₂Fold Difference. n = 3 biological replicates per group. **f** Dotplot showing the average and percentage of expression of canonical marker genes for each cluster identified in (D). **g** Ridgeplot comparing the expression of *SNCA* between each cluster identified in (D). $n = 3$ biological replicates per group, p-values calculated by unpaired two-tailed Student's *t* test.

neuron (Fig. S2E) and glia subclusters (Fig. S2H) revealed a shift in the expression distribution in the SNCA 3x line, which was highly heterogeneous when correlated to subcluster abundance. Differential expression analyses (Supplementary Data 4) revealed mitochondrial dysfunction, a PD hallmark, as a key feature of SNCA 3x ENLs. Accordingly, our integrated enrichment analysis highlighted several metabolic-associated pathways as significantly changed, both in enteric neurons (Fig. 2d, Supplementary Data 6) and enteric glia (Fig. 2h, Supplementary Data 6).

In enteric neurons, the subclusters 3, 4, and 9 were most strongly associated with these pathways, which included *"fatty acid transport"*, *"acetyl-CoA metabolic process"* and *"steroid metabolic process"* (Fig. 2d). When plotting the genes directly linked to the enriched pathways (Supplementary Data 6) on a heatmap, we noticed a uniform decrease across all three subclusters in the expression of key genes associated with acetyl-CoA metabolic process (*HMGCS1, MVD, ACSS2, MVK*), steroid metabolic process (*HMGCS1, FDFT1, SQLE, MSMO1, LDLR,*

*INSIG1, HMGCR, IDI1*), and fatty acid transport (*ACSL3*) in SNCA 3x enteric neurons (Fig. 2b).

In enteric glia, subclusters 0, 1, 5, 6, and 11 exhibited strong associations with mitochondrial pathways, including terms like *"mitophagy"*, *"mitochondrial ATP synthesis coupled electron transport"*, *"release of cytochrome c from mitochondria"*, *"response to oxidative stress"* and *"mitochondrial protein import"* (Fig. 2h). To further investigate the underlying stress mechanisms, we visualized the key genes driving these enrichments on a heatmap and observed a general upregulation of the *CHCHD2* gene, associated with cellular responses to oxidative stress. Additionally, we detected increased expression of the mitophagy-associated gene *MAP1LC3A* in subclusters 0, 6, and 11, and of the antioxidant gene *MT3* in subclusters 0 and 6 of SNCA 3x enteric glia (Fig. 2f). Moreover, subcluster 11 exhibited elevated expression of mitochondrial protein import genes (*DNAJC19, PAM16,* and *TIMM13*) in SNCA 3x glia (Fig. 2f). Finally, a module score analysis comparing key genes regulating cholesterol metabolism and

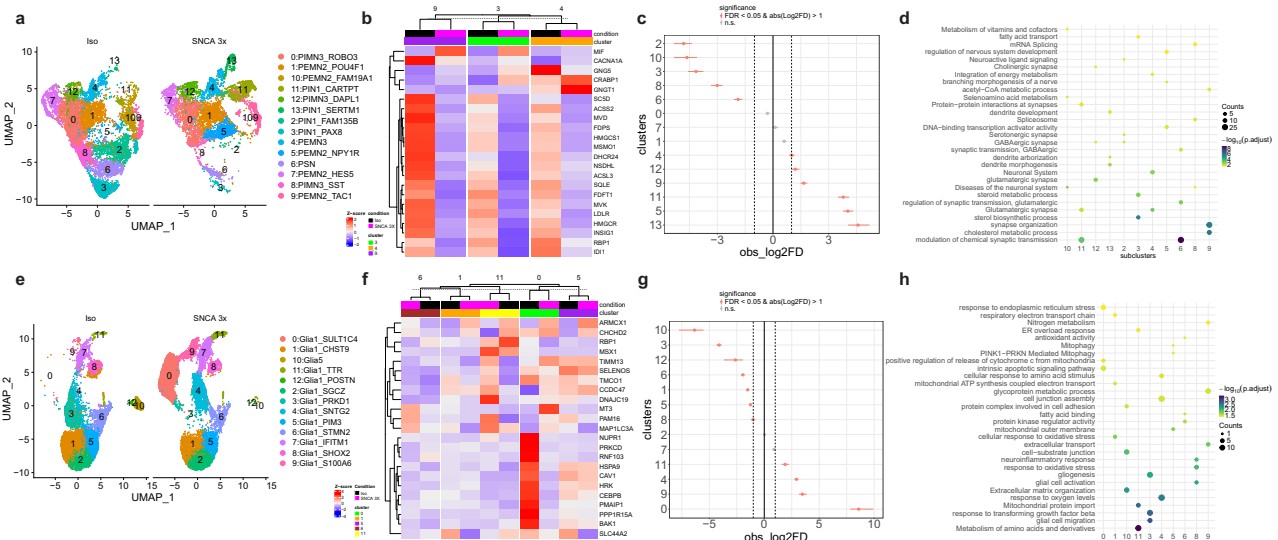

**Fig. 2 | SNCA 3x drives mitochondrial alterations at transcriptional level and alters subpopulation composition of enteric neurons and glia.** UMAP plots obtained from subclustering analysis of the enteric neurons (**a**) and glia (**e**) (see Fig. 1d) of Iso and SNCA 3x ENLs at day 70 after start of differentiation. Annotations were performed using SingleR based on Drokhlyansky et al.[27]. Heatmap obtained from differential gene expression analysis of Iso vs SNCA 3x enteric neurons (**b**) and glia (**f**) characterizing the expression of transcripts associated with the regulation of mitochondrial-associated pathways depicted in (**d**) and (**h**). Compositional analysis plot showing the significantly changed enteric neuron (**c**) and glia (**g**) subclusters between Iso and SNCA 3x. Data are presented as observed log₂ fold change (center points) +/− 95% confidence intervals (error bars). Confidence intervals were calculated using bootstrap resampling. *P*-values were determined via permutation testing and adjusted for multiple comparisons using the Benjamini-Hochberg

method (adj. *p*-value). FDR=False Discovery Rate; log2FD=log₂Fold Difference. *n* = 3 biological replicates per group. Integrated enrichment analysis of differentially expressed genes in enteric neuron (**d**) and glia (**h**) subclusters. Analysis includes Reactome, KEGG, and Gene Ontology (Biological Processes, Molecular Function, and Cellular Component) based on differentially expressed genes between SNCA 3x and Iso, separated by subcluster (see Supplementary Data 4 for gene lists and Supplementary Data 6 for full enrichment results). Statistical significance was determined using a one-sided hypergeometric test (Over-Representation Analysis) via the clusterProfiler R package. *P*-values were adjusted for multiple comparisons using the Benjamini-Hochberg method, with a significance threshold of adjusted *p* < 0.05. The background universe for all tests was defined as all genes detected in the single-cell RNA-sequencing experiment.

mitochondrial stress responses (Fig. S3A–B) revealed a significant negative correlation between these two modules specifically in subcluster 12 of SNCA 3x glia, a population markedly reduced in this genotype (Fig. 2g).

Compositional analysis (Fig. 2c, g), performed using the scProportion test with a log2FC threshold of 1, revealed several alterations in subcluster proportions driven by α-syn in both enteric neurons and glial cells. For enteric neurons, we highlight a decrease in a population of sensory neurons (subcluster 6) together with an increase in an interneuron subtype (subcluster 11) that is highly associated with glutamatergic synapses (Fig. 2d) in SNCA 3x lines, suggesting a functional link between α-syn accumulation and altered gastrointestinal motility mediated by the regulation of the enteric glutamatergic circuitry[28]. Concurrently, SNCA 3x enteric glia exhibited a pronounced increase in the proportion of subcluster 0 (Glia 1), a population strongly associated with pathways of endoplasmic reticulum stress and the release of cytochrome c from the mitochondria (Fig. 2h), alongside a decline in subcluster 3 (Glia 1), linked to response to TGF-β (Fig. 2h), an immunoregulatory cytokine.

Next, we first sought to determine if the α-syn-driven differences in cell abundance were due to localized changes within specific cell neighborhoods using MiloR differential abundance analysis. MiloR identified localized neighborhood-level changes specifically within the neuronal subclusters (minimal spatial FDR = 0.15; Fig. S2I, Supplementary Data 5). The neuronal neighborhood changes correspond to the same subclusters identified as decreased (e.g., subcluster 2) or increased (e.g., subcluster 13) using the scProportion test in SNCA 3x cells. In contrast, MiloR analysis of the glial subset failed to detect localized changes (Fig. S2J, Supplementary Data 5), suggesting the glial proportional shifts are more global.

Finally, to compare SNCA 3x-associated neuronal dysfunction across distinct cellular contexts, we mapped our iPSC-ENL scRNAseq data to published midbrain (Patikas et al.[29]) and cortical (Jin et al.[30]) SNCA 3x organoid signatures using SingleR. Our enteric neurons primarily matched the dopaminergic neuron 2 (oDAn2)[29] (Fig. S3C–D) and the inhibitory neuron (INs)[30] clusters (Fig. S3F–G). Integrated enrichment analysis using these cross-study neuronal signatures revealed common dysregulated pathways in SNCA 3x ENLs, predominantly related to synaptic function and cholesterol metabolism (Fig. S3E, S3H). Furthermore, leveraging the glial component in the cortical model, our enteric glia matched predominantly the astrocyte (AS) cluster (Fig. S3I-J). Enrichment analysis of the AS signature in our enteric glia identified shared metabolic and mitochondrial dysfunction modules (e.g., acetyl-CoA metabolic process, regulation of glucose and lipid metabolism, and cellular response to oxygen levels), consistent with the specific phenotypes observed in our enteric glial subclusters (Fig. S3K).

These data suggest that SNCA 3x disrupts mitochondrial homeostasis in enteric neurons and glia at the transcriptional level, altering subpopulation dynamics and inducing metabolic stress-driven vulnerability.

## SNCA 3x drives basal mitochondrial dysfunction and alters enteric neuron-glia communication in iPSC-ENLs

Given the extensive transcriptional changes in mitochondrial-linked genes and the enrichment of metabolic pathways in SNCA 3x enteric neurons and glia (Fig. 2) and their relevance to PD[1,19,31] we wanted to validate the findings by analyzing different aspects of mitochondrial morphology. Using TOM20 immunostaining and automated image analysis, we assessed several mitochondrial parameters (Fig. 3a–b).

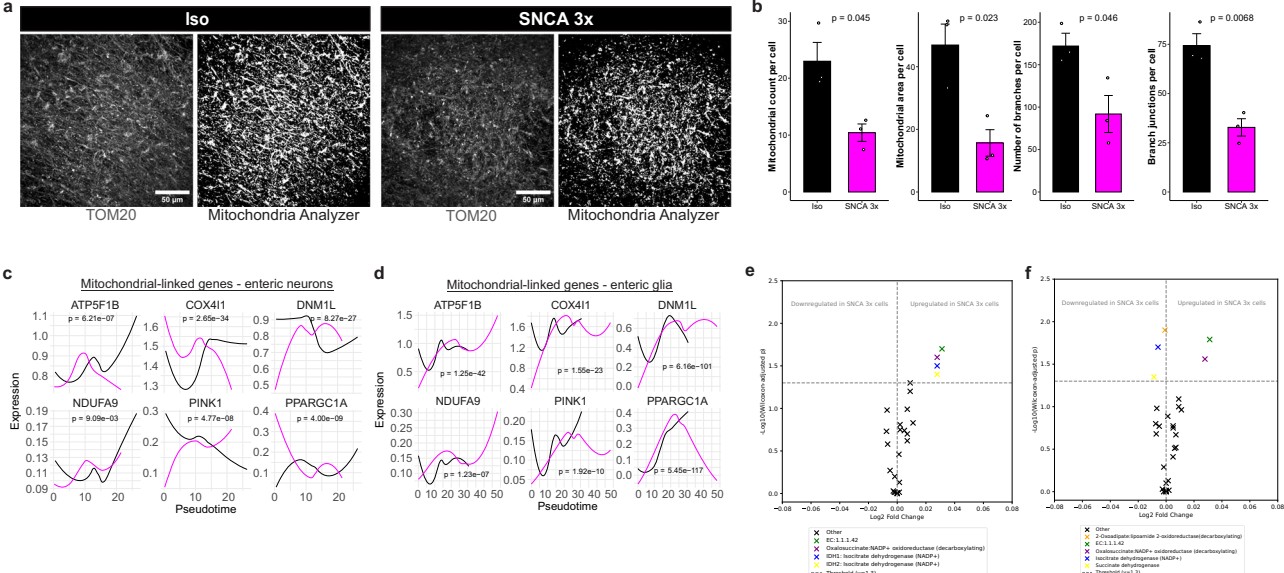

**Fig. 3 | SNCA 3x drives basal mitochondrial dysfunction and alters enteric neuron-glia communication in iPSC-ENLs. a** Representative panels of TOM20 immunostaining and the mitochondrial mask created with the Mitochondria Analyzer plugin for FIJI. Left panels represent Iso and right panels SNCA 3x ENLs, as indicated. Cells were analyzed on day 70 after start of differentiation. Scale bar=50 μm. **b** Quantification of mitochondrial count, area, number of branches and branch junctions in iPSC-ENLs calculated with the Mitochondria Analyzer plugin for FIJI. All data were normalized to the number of cell nuclei per image, calculated using DAPI staining in CellProfiler. The total number of cells analyzed per cell line and used to normalize the data was 1276 (Iso-1), 1534 (SNCA 3x-1), 810 (Iso-2), 1511 (SNCA 3x-2), 614 (Iso-3) and 1083 (SNCA 3x-3). Data points in the graphs represent *n* = 3 independent SNCA 3x and 3 isogenic lines per group, from one differentiation, mean ± SEM, *p*-values calculated by unpaired two-tailed Student's *t*-test. Source data are provided as a Source Data file. Expression of mitochondrial-linked genes in SNCA 3x vs. Iso enteric neurons (**c**) and glia (**d**) along pseudotime. Gene expression values were modeled using linear regression across pseudotime, with FDR correction for multiple comparisons. Metabolic flux analysis of mitochondrial function in SNCA 3x vs Iso enteric neurons (**e**) and glia (**f**). We calculated the logarithm of the Wilcoxon adjusted *p*-value to filter reaction differences. A threshold of 1.3 (corresponding to an adjusted *p*-value of 0.05) was used to identify significant reactions.

Interestingly, our data detected a significant decrease in mitochondrial count and area per cell in SNCA 3x ENLs (Fig. 3b), suggestive of mitochondrial stress and mitophagy. Additionally, the number of branches and the branch junctions per cell were also significantly decreased in SNCA 3x ENLs (Fig. 3b), indicating shortening of the mitochondrial network, which is a classical feature of mitochondrial fission.

Next, we analyzed mitochondrial-linked gene expression along pseudotime (Fig. 3c–d), using Slingshot[32]. We selected the root of the trajectories based on the enteric neuron and glial subclusters which showed the highest entropy levels (Fig. S4A-B). Our results revealed that in SNCA 3x enteric neurons, genes controlling oxidative phosphorylation[33], ATP production[33] and mitochondrial biogenesis[34] were downregulated (*ATP5F1B*, *COX4I1*, *NDUFA9*, *PPARGC1A*), while those controlling mitophagy[35] (*PINK1*) and fission[36] (*DNM1L*) were upregulated (Fig. 3c). Conversely, in glial cells, *ATP5F1B*, *COX4I1*, and *NDUFA9* were upregulated along with *PINK1* and *DNM1L*, whereas *PPARGC1A* showed a transient peak before downregulation (Fig. 3d). Complementarily, we performed a metabolic flux analysis focusing on mitochondrial function, which suggested an upregulation of Oxalo-succinate:NADP+ oxidoreductase in both SNCA 3x enteric neurons (Fig. 3e) and glia (Fig. 3f), indicating a shift toward NADPH production and highlighting potential compensatory metabolic adaptations to mitochondrial dysfunction.

Using the scRNAseq data, we built on the previous analyses and explored how SNCA 3x affects cellular communication and signaling in iPSC-ENLs using CellChat[37]. Globally, we observed an overall increase in the total number and strength of inferred intercellular interactions in the SNCA 3x group (Fig. S1F−G). However, this was driven by different changes at the cell type level: enteric neurons exhibited a net decrease in the number of inferred interactions (Fig. S4C−D), while enteric glia showed a net increase in interaction strength (Fig. S4G−H).

This context is crucial for interpreting subsequent specific pathway changes. Analysis across all cell types highlighted marked alterations in VEGF-related signaling, key regulators of cell survival[38], vascularity[38] and neuroinflammation[39,40] (Fig. S1H, Supplementary Data 7) in SNCA 3x cells. Specifically, we observed an increased *VEGFB-VEGFR1* signaling predominantly mediating glia-glia communication, and a corresponding increase in *VEGFA-VEGFR1* signaling directing communication between enteric neurons and glia (Fig. S1I, Supplementary Data 7). Focusing on enteric neuronal communication, we detected the de novo emergence of Bone Morphogenetic Protein (BMP) signaling, associated with neurite fasciculation and orientation within the bowel[41], exclusively in the SNCA 3x enteric neurons (Fig. S4E, Supplementary Data 7), with the entire source originating from subcluster 10 (Fig. S4F, Supplementary Data 7), a population simultaneously majorly depleted (Fig. 2c). Specifically, we observed a high communication probability for *BMP7* (source: subcluster 10, identified as excitatory motor neurons) signaling to *BMPR1A + BMPR2* (target: subcluster 2, identified as interneurons), with subcluster 2 also being the most significantly depleted neuronal subpopulation in SNCA 3x neurons (Fig. 2c). Finally, focused analysis of enteric glial signaling revealed a de novo activation of Epidermal Growth Factor (EGF) signaling (Fig. S4I, Supplementary Data 7), previously linked to intestinal mucosal healing[42], which was exclusively sourced by the vulnerable subcluster 8 in the SNCA 3x line (Fig. S4J, Supplementary Data 7). The *EGF-EGFR* signaling from subcluster 8 showed the highest communication probability to subcluster 11 (Fig. S3J, Supplementary Data 7), which is slightly upregulated in SNCA 3x (Fig. 2g) and is strongly associated with mitochondrial protein import (Fig. 2h). Subcluster 8 drove the entire EGF signal and was transcriptionally associated with glial activation and neuroinflammatory response (Fig. 2h).

Our findings demonstrate that α-syn accumulation not only induces significant mitochondrial and metabolic stress but also

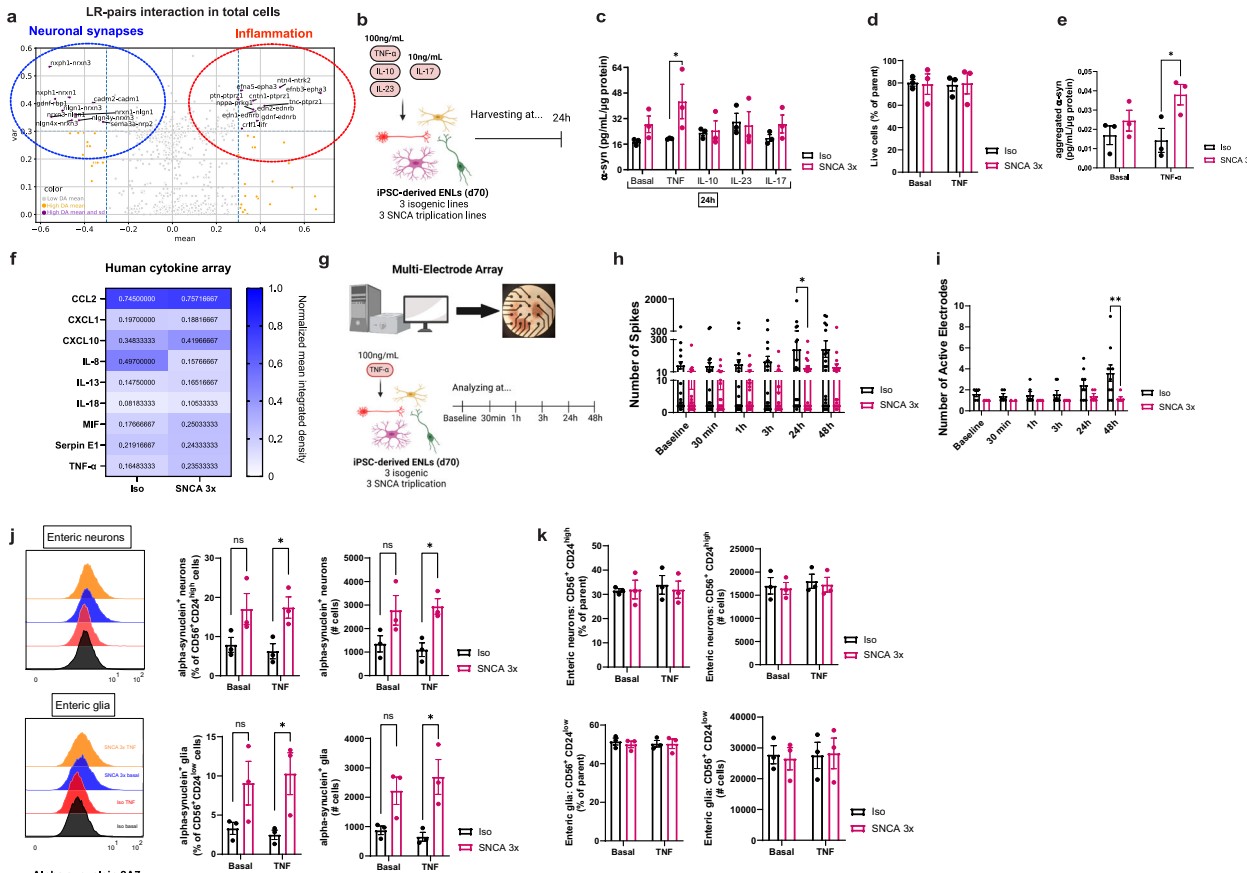

**Fig. 4 | TNF uncovers genotype-specific α-syn accumulation and synaptic dysfunction in SNCA 3x ENLs. a** Ligand-Receptor pair analysis of scRNAseq data prior to subclustering. Pairs with large mean change and high variance are purple. Pairs failing variance (≥ 0.3) or mean (≥ 0.3) cut-offs are orange and gray, respectively. **b** Experimental paradigm depicting iPSC-ENL stimulation with cytokines. Created in BioRender. Winner, B. (2026) https://BioRender.com/a2k344q. **c** ELISA of total α-syn, normalized by total protein. *n* = 3 independent SNCA 3x and 3 isogenic lines per group, from two independent differentiations, mean ± SEM, *\*p* = 0.0433 by two-way ANOVA with Sidak post-hoc. **d** Flow cytometry analysis using Live/Dead staining. *n* = 3 independent SNCA 3x and 3 isogenic lines per group, from two independent differentiations, mean ± SEM. **e** ELISA of aggregated α-syn, normalized by total protein. *n* = 3 independent SNCA 3x and 3 isogenic lines per group, from two independent differentiations, mean ± SEM, *\*p* = 0.0303 by two-way ANOVA with Sidak post-hoc. **f** Human cytokine array quantification of the supernatants of iPSC-ENLs previously stimulated for 24 h with 100 ng/ml TNF. Data represent the mean of *n* = 3 independent SNCA 3x and 3 isogenic lines per group,

from one differentiation. **g** Experimental paradigm for the MEA experiments. Created in BioRender. Winner, B. (2026) https://BioRender.com/vwugkr4. Quantification of the number of spikes (**h**), and active electrodes (**i**), *n*=wells of a CytoView MEA 48-well plate, representative of *n* = 3 independent SNCA 3x and 3 isogenic lines per group, from two independent differentiations, mean ± SEM, *\*p* = 0.046, *\*\*p* = 0.0031 by two-way ANOVA with Sidak post-hoc. **j** Flow cytometry analysis of the percentage and total of α-syn⁺ (2A7 antibody) enteric neurons (CD56⁺CD24^high) and enteric glia (CD56⁺CD24^low). *n* = 3 independent SNCA 3x and 3 isogenic lines per group, from two independent differentiations, mean ± SEM, *p* = 0.0419 (% α-syn⁺ neurons); *p* = 0.0294 (# α-syn⁺ neurons); *p* = 0.0478 (% α-syn⁺ glia); *p* = 0.0123 (# α-syn⁺ glia) by two-way ANOVA with Sidak post-hoc. **k** Flow cytometry analysis of the percentage and total enteric neurons (CD56⁺CD24^high) and enteric glia (CD56⁺CD24^low). *n* = 3 independent SNCA 3x and 3 isogenic lines per group, from two independent differentiations, mean ± SEM. Source data are provided as a Source Data file in **c**–**f**, **h**, **I**, **j**, **k**. Basal: DPBS + 0.1% BSA.

fundamentally remodels the inter and intra-cellular communication architecture of the ENS, favoring signaling pathways associated with stress, neuroinflammation, and survival. These changes set the stage for further metabolic and functional impairments under inflammatory conditions, as explored in the following section.

## TNF uncovers genotype-specific α-syn accumulation and synaptic dysfunction in SNCA 3x ENLs

The ENS, as a vital component of the intestinal microenvironment, plays a role in coordinating local immune responses[43], besides other functions. Cytokines, as key regulators of immune signaling, mediate both physiological and pathological processes within the ENS[43].

Our scRNAseq data revealed that SNCA 3x alters biological pathways and intercellular communication. Additional ligand-receptor pair analysis in total cells identified activated inflammatory pairs (e.g., *EDN1-EDNRB*[44], *CRLF1-LIFR*[45]) and repressed synapse-regulating pairs (e.g., *NLGN1-NRXN3*[46], *NXPH1-NRXN1*[47]) (Fig. 4a) in SNCA 3x ENLs. Of

note, most of these synapse-associated pairs were downregulated in enteric neurons (Fig. S6A) whereas the inflammation associated pair *EFNB3-EPHA3*[48] was present both in enteric neurons and glia (Fig. S6A-B). This prompted us to investigate the impact of various cytokines (TNF, IL-10, IL-23, IL-17) - relevant both to PD and regulation of mucosal immunity[1,43] - on α-syn biology and neuronal function in iPSC-ENLs. Cytokine treatments revealed a genotype-dependent effect (F = 5.539, *p* = 0.0289), as only TNF significantly increased total α-syn levels in SNCA 3x ENLs compared to TNF-treated Iso controls, as measured by ELISA (Fig. 4b-c). Interestingly, at a basal level, our scRNAseq detected that SNCA 3x cells express generally higher levels of TNF-linked genes (*TNFRSF1A*, *NFKBIA*, *NFKB1*, *RELA*, *MIF*) across most enteric neuronal and glial subclusters (Fig. S5A–D and S5F), indicating that both populations retain the molecular machinery required for TNF responsiveness. Importantly, treatment with TNF did not alter cell viability for both genotypes (Fig. 4d). Dot blot analysis further confirmed higher α-syn levels in TNF-treated SNCA 3x ENLs compared to

TNF-treated Iso ENLs (Fig. S6C). However, α-syn levels in TNF-treated SNCA 3x ENLs were not significantly different from those at baseline (Fig. 4c, S6C), suggesting that the SNCA 3x genotype primarily drives α-syn elevation, while TNF further increases levels relative to controls. Given previous reports of α-syn aggregates in the gut of PD patients[8], we assessed its solubility using sarkosyl fractionation[49]. α-Syn signal was detected in the pellet fractions of SNCA 3x ENLs, but no significant differences were observed between Iso and SNCA 3x under basal conditions or following TNF stimulation (Fig. S6D). We investigated further potential changes using ELISA, which revealed a modest increase in aggregated α-syn in SNCA 3x ENLs upon TNF stimulation (Fig. 4e). Cytokine array showed no significant changes in cytokine production between groups upon TNF treatment (Fig. 4f). This suggests that SNCA 3x ENLs exhibit an intrinsic susceptibility to TNF, with selective increase of α-syn species, occurring independently of the induction of a broad inflammatory response.

Subsequently, to assess how TNF impacts enteric neuronal function over the time, we performed a functional electrophysiology assay to measure neuronal activity using the Multielectrode array (MEA) platform (Fig. 4g). Interestingly, while the Iso ENLs reacted to TNF treatment by increasing the number of spikes and active electrodes over time, SNCA 3x ENLs showed no specific response (Fig. 4h–i). There was a significant difference between Iso and SNCA 3x ENLs after 24 h and 48 h of TNF exposure in number of spikes (Fig. 4h) and active electrodes (Fig. 4i), respectively. No differences were observed between Iso and SNCA 3x ENLs regarding weighted mean firing rate measurements (Fig. S6E). Accordingly, our scRNAseq data already detected that at a basal level there is reduced expression of synaptic and neuronal identity markers such as *DSCAM* and *GABRB3* (Fig. S5E and Supplementary Data 4) across enteric neuronal subclusters induced by α-syn triplication.

In line with these data, we utilized a previously published flow cytometry pipeline for studying the ENS[50], in which we could clearly separate the enteric neuron and glial populations based on their CD24 expression levels (Fig. S6F), also supported by our scRNAseq data (Fig. S6G-H), observing that TNF stimulation unmasks total α-syn accumulation both in enteric neurons and enteric glial cells (Fig. 4j). No changes were observed in the proportions of enteric neurons and glia between Iso and SNCA 3x, both under basal and TNF stimulation conditions (Fig. 4k), in accordance with our scRNAseq data (Fig. 1e, Fig S1D).

These findings suggest that SNCA 3x ENLs exhibit an impaired capacity to modulate their neuronal activity in response to TNF, which contrasts with the adaptive response observed in Iso ENLs. This functional deficit, coupled with selective α-syn increase in SNCA 3x ENLs, further supports an intrinsic vulnerability of SNCA 3x ENLs to inflammatory stress, reinforcing a link between α-syn accumulation and synaptic dysfunction in the ENS.

### TNF rewires the metabolism of SNCA 3x ENLs by disrupting the malate-aspartate shuttle and impairing TCA cycle flux

Building on the links between α-syn and mitochondrial dysfunction identified in our scRNAseq data, we used multiomics analyses to further examine the impact of TNF on the iPSC-ENL proteome and metabolome (Fig. 5a).

Data independent acquisition (DIA)-based shotgun proteomics coupled to label-free quantification was used to quantify the proteomes of Iso and SNCA 3x ENLs in a basal level and upon TNF stimulation. When these proteomes were compared (Fig. 5B–C and Supplementary Data 8), differentially expressed proteins (DEPs) were identified ($p < 0.05$, |log2 FC| ≥ 0.8). The effect of SNCA 3x (SNCA 3x versus Iso, basal) resulted in the most robust proteomic alteration with 116 DEPs. This indicates a strong effect of α-syn in the proteome of iPSC-ENLs. Interestingly, only one DEP was identified when Iso cells were treated with TNF compared to Iso basal, while 67 DEPs were

identified in SNCA 3x cells upon TNF treatment compared to SNCA 3x at a basal level and 46 DEPs were detected when comparing SNCA 3x versus Iso under TNF treatment (Supplementary Data 8). This strongly suggests that α-syn overexpression predisposes iPSCs-ENLs to cytokine-induced metabolic stress, establishing a specific vulnerability linked to the SNCA 3x genotype. Gene Ontology Molecular Function analyses revealed a significant enrichment in synaptic functions, including SNAP receptor and SNARE binding in basal SNCA 3x ENLs, suggesting neurotransmitter release and vesicle trafficking alterations[51] (Fig. 5D, Supplementary Data 9). TNF exposure shifted the profile toward oxidative stress and mitochondrial dysfunction, marked by increased peroxidase, antioxidant, and oxygen transport functions, indicating impaired mitochondrial respiration (Fig. 5D, Supplementary Data 9). This progression from synaptic dysfunction to metabolic disruption underscores the impact of inflammatory stress.

LC-Orbitrap-MS metabolomics highlighted 9 differentially expressed metabolites ($p < 0.05$, |log2FC| ≥ 0.5) between Iso and SNCA 3x ENLs. The analysis revealed reductions in aspartate, glutamate, malate and glutamine levels in SNCA 3x ENLs treated with TNF, suggesting malate-aspartate shuttle (MAS) impairment, a key process for tricarboxylic acid (TCA) cycle flux and mitochondrial redox balance[52] (Fig. 5E, Supplementary Data 10). Accordingly, $NAD^+$ levels were significantly reduced in these cells, with no changes in NADH levels, suggesting altered $NAD^+$ homeostasis, potentially due to impaired regeneration or increased utilization in response to metabolic stress (Fig. 5E, Supplementary Data 10). Metabolite-set enrichment analysis on the differentially expressed metabolites (DEMs) (Supplementary Data 10) confirmed MAS as a top affected pathway (Fig. 5F), supporting inflammation-driven metabolic reprogramming.

When checking for the levels of key proteins associated with MAS in our proteomics data (Supplementary Data 8), cytosolic malate dehydrogenase (MDHC) was significantly decreased by TNF specifically in SNCA 3x ENLs, indicating a treatment effect within this genotype, but with no differences between genotypes upon TNF exposure (Fig. S7A). Mitochondrial malate dehydrogenase (MDHM) showed consistent differences between SNCA 3x and isogenic controls under both basal and TNF conditions, suggesting a baseline genotype effect rather than a TNF-specific response (Fig. S7B). Mitochondrial aspartate aminotransferase (AATM) did not change with TNF in either genotype (Fig. S7C). By contrast, aspartate aminotransferase (AATC) (Fig. 5G) exhibited the clearest pattern of selective vulnerability: it was significantly reduced by TNF in SNCA 3x and also differed between genotypes under TNF exposure. Taken together, these data indicate that MAS alterations are not uniform but rather enzyme-specific, with AATC emerging as the most compelling candidate linking TNF exposure to MAS dysfunction in the SNCA 3x background. We then validated our findings using western-blotting for AATC, which showed a trend for decreased levels in SNCA 3x already at a basal level ($p = 0.2775$), with a higher tendency noted upon the addition of TNF ($p = 0.1870$) (Fig. 5H), in line with the proteomics finding. Interestingly, when looking at the activity of key TCA enzymes, we observed a trend for decreased citrate synthase in SNCA 3x ENLs, especially upon TNF stimulation (Fig. S7D), with no alterations in succinate dehydrogenase between groups (Fig. S7E). Finally, integrated pathway analysis linked TNF exposure to extracellular matrix receptor interactions and glutamate metabolism (Fig. 5I).

These findings demonstrate that TNF disrupts metabolic homeostasis in SNCA 3x ENLs by impairing MAS and the TCA cycle, leading to mitochondrial dysfunction.

### TNF increases α-syn-mitochondria interactions and highlights oxidative stress-associated populations in SNCA 3x ENLs

Given that α-synuclein binds to mitochondria and that mitochondrial dysfunction can promote protein aggregation and oxidative stress[31], we next investigated whether the metabolic alterations observed at

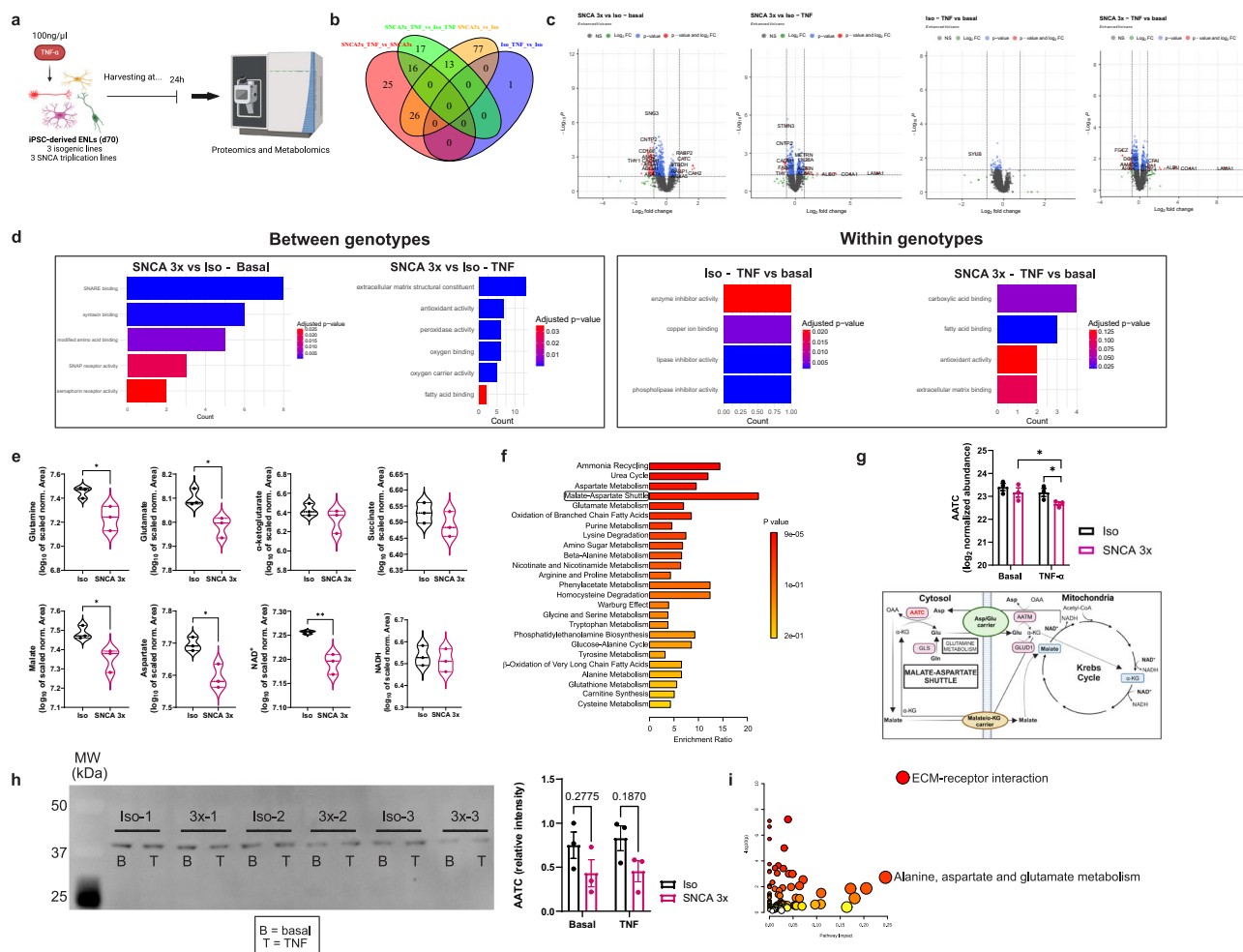

**Fig. 5 | TNF rewires the metabolism of SNCA 3x ENLs by disrupting the malate-aspartate shuttle and impairing TCA cycle flux. a** Experimental paradigm for proteomics (basal and TNF) and metabolomics (TNF only) of iPSC-ENLs. Created in BioRender. Winner, B. (2026) https://BioRender.com/6491x2m. **b** Venn diagram of differentially expressed (DE) proteins (FC > 1.75, $p < 0.05$) across four comparisons: SNCA 3x vs. Iso (Basal, yellow; TNF, green) and treated vs. basal (SNCA 3x, red; Iso, blue). DE proteins identified via two-sided empirical Bayes moderated $t$-tests (limma package, unadjusted $p$-values). Basal: vehicle control (DPBS + 0.1% BSA). See Supplementary Data 8. **c** Volcano plots highlighting DE proteins for the four comparisons in (**b**). $X$-axis: log2 fold change; $Y$-axis: -log10($p$-value). Statistical significance (dashed lines: unadjusted $p < 0.05$, |log2FC| >= 0.8) determined via empirical Bayes moderated t-test (limma R package). Data: $n = 3$ biological replicates/condition. **d** Molecular function enrichment analysis of the comparisons in (**b**) (Supplementary Data 9). Significance determined by one-sided hypergeometric test (enrichGO, clusterProfiler R package); p-values adjusted via Benjamini-Hochberg method (padj<0.05). **e** Quantification of intracellular metabolites in TNF-treated iPSC-ENLs. Raw p-values calculated scaling for donor effect ($n = 3$ lines/group, one differentiation). Unpaired two-tailed Student's $t$-test: $p = 0.0281$ (glutamine); $p = 0.0105$ (glutamate); $p = 0.029$ (malate); $p = 0.0168$ (aspartate);

$p = 0.006$ (NAD + ). Data: $n = 3$ biological replicates/condition. **f** Metabolite-set enrichment analysis of DE proteins from metabolomics (Supplementary Data 10) using SMPDB database (MetaboAnalyst). Analysis via over-representation analysis (one-sided hypergeometric test). Bars colored by unadjusted $p$-value ($p < 0.05$ threshold). (**g**) Cytosolic aspartate aminotransferase (AATC) quantification from proteomics ($n = 3$ lines/group, one differentiation). Mean ± SEM; $p = 0.021$ (Iso vs. SNCA 3x, TNF); $p = 0.029$ (SNCA 3x, TNF vs. Basal) via moderated t-tests (limma, unadjusted $p$-values). Metabolic map (below) shows MAS, TCA cycle, and glutamine metabolism. OAA=oxaloacetate, Asp=aspartate, Gln=glutamine, Glu=glutamate, αKG = α-ketoglutarate, GLS=glutaminase, GLUD1=glutamate dehydrogenase, AATM=mitochondrial aspartate aminotransferase. AATC is highlighted in red. Created in BioRender. Winner, B. (2026) https://BioRender.com/g006z5p.
**h** Western blot validation of AATC levels ($n = 3$ lines/group, one differentiation). Mean ± SEM, two-way ANOVA with Sidak post-hoc. **i** Integrated pathway analysis merging DE proteins and metabolites (SNCA 3x vs. Iso, TNF) via MetaboAnalyst 6.0 Joint Pathway Analysis module (one-sided hypergeometric test and degree centrality). Dots colored by unadjusted $p$-value (p < 0.05). Source data are provided as a Source Data file in **e**, **g**, **h**.

basal conditions (Fig. 3) and after TNF treatment (Fig. 5) contribute to mitochondrial α-syn accumulation and oxidative stress (Fig. 6A). Proximity ligation assays (PLA) of α-syn and TOM20 confirmed increased α-syn–mitochondria interactions in SNCA 3x ENLs at baseline. TNF stimulation further enhanced the PLA signal in both genotypes, with a greater increase in SNCA 3x ENLs compared to isogenic controls (Fig. 6B–C).

To examine mitochondrial oxidative stress at single-cell resolution, we analyzed mitoSOX fluorescence by flow cytometry in iPSC-derived enteric neuron-like (ENL) cultures. Unsupervised t-distributed

stochastic neighbor embedding (t-SNE) of mitoSOX signals revealed a largely homogeneous distribution under basal conditions in both control and SNCA 3x enteric neurons (Fig. 6D) and glia (Fig. 6E). Following TNF exposure, a distinct mitoSOX-high subpopulation emerged predominantly in the SNCA 3x condition, encompassing both neuronal (Fig. 6D) and glial (Fig. 6E) cells. This cluster was consistently detected across biological replicates, although the overall increase in mitoSOX fluorescence per cell type did not reach statistical significance in bulk quantification. The reproducible appearance of this discrete mitoSOX-high cluster supports a TNF-induced oxidative

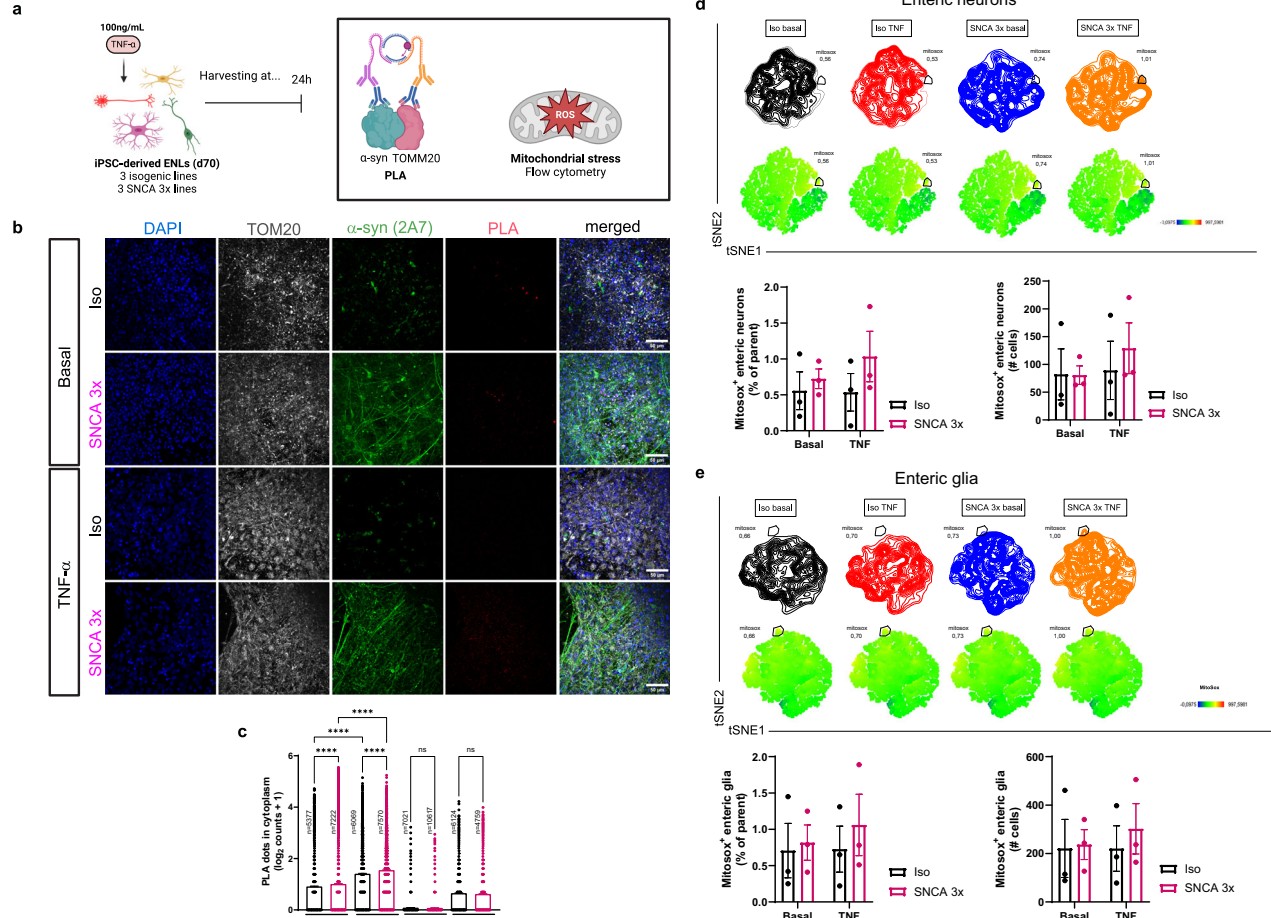

**Fig. 6 | TNF increases α-syn-mitochondria interactions and highlights oxidative stress-associated populations in SNCA 3x ENLs. a** Paradigm describing the proximity ligation assay (PLA) and measurement of mitochondrial ROS experiments. Created in BioRender. Winner, B. (2026) https://BioRender.com/vb5oi60. **b** Assessment of PLA results using immunocytochemistry. Panels are separated in basal and TNF stimulated for both Iso and SNCA 3x groups. Panels show the nuclear marker DAPI (blue), the mitochondrial marker TOM20 (gray), α-syn (green) and PLA dots for TOM20-α-syn (red). Scale bar = 50 μm. Basal refers to cells treated with vehicle used to dilute the TNF (DPBS + 0.1% BSA). **c** Quantification of the PLA dots in the cytoplasm. n=number of total individual cells analyzed per condition indicated

in the figure, from 3 independent SNCA 3x and 3 isogenic lines per group, from one differentiation, with mean ± SEM, ****$p < 0.0001$, by one-way ANOVA with Tukey's post-hoc. Basal refers to cells treated with vehicle used to dilute the TNF (DPBS + 0.1% BSA). Data were normalized to the number of nuclei per image. Source data are provided as a Source Data file. Concatenated t-SNE plots from flow cytometry data of all lines highlighting Mitosox[high] populations across genotypes and conditions and quantification of the percentage and total mitosox[high] cells in enteric neurons (CD56[+]CD24[high]) (**d**) and in enteric glia (CD56[+]CD24[low]) (**e**). $n = 3$ independent SNCA 3x and 3 isogenic lines per group, from two independent differentiations, mean ± SEM. Source data are provided as a Source Data file.

stress response in a subset of SNCA 3x ENL cells, consistent with the mitochondrial vulnerability inferred from reduced AATC abundance and impaired MAS activity.

Collectively, these findings establish a mechanistic link between TNF–induced metabolic reprogramming and mitochondrial pathology, in which enhanced α-syn–mitochondria interactions and increased oxidative stress contribute to impaired bioenergetic function and neuronal vulnerability in SNCA 3x ENLs.

### TNF increases the reliance of SNCA 3x ENLs on glutamine oxidation, which is rescued by CSB6 treatment

To further investigate the metabolic vulnerability of SNCA 3x ENLs and assess the potential rescue by a glutamate metabolism modulator, CSB6, we first performed additional Seahorse Mito Stress and Mito Fuel Flex assays (Fig. 7A). These experiments revealed genotype and condition-dependent effects of TNF and CSB6 on mitochondrial respiration (Fig. 7B–F). Two-way ANOVA identified a significant genotype versus condition interaction for both basal (F = 3.3, $p = 0.0475$) and ATP-linked oxygen consumption rates (OCR; F = 4.677, $p = 0.0157$), indicating that treatment responses differed between genotypes. The

condition main effect was significant across multiple mitochondrial parameters, including basal, ATP-linked, maximal, and non-mitochondrial OCR (all F > 5, $p < 0.01$), consistent with broad treatment-induced changes in mitochondrial function.

TNF significantly reduced ATP-linked respiration in both genotypes, an effect fully rescued by CSB6 treatment (Fig. 7B–C). For proton leak, CSB6 significantly counteracted TNF-induced decrease only in SNCA 3x ENLs, indicating a genotype-selective recovery of mitochondrial reserve capacity. Post-hoc analyses further revealed that under CSB6-only conditions, SNCA 3x ENLs displayed higher basal ($p = 0.0163$) and ATP-linked OCR ($p = 0.0060$) than isogenic controls, despite the absence of a main genotype effect across all conditions (Fig. 7B–C). This suggests that CSB6 enhances mitochondrial respiration in both genotypes but provides a higher benefit to the metabolically compromised SNCA 3x cells. In the mitofuel assays (Fig. 7D), we again observed a significant genotype versus condition interaction for glutamine dependency (F = 4.258, $p = 0.0217$). Specifically, TNF significantly increased glutamine dependency in SNCA 3x ENLs, an effect fully rescued by CSB6 (Fig. 7E). Analysis of fatty acid oxidation (FAO) dependency revealed no significant differences between genotypes

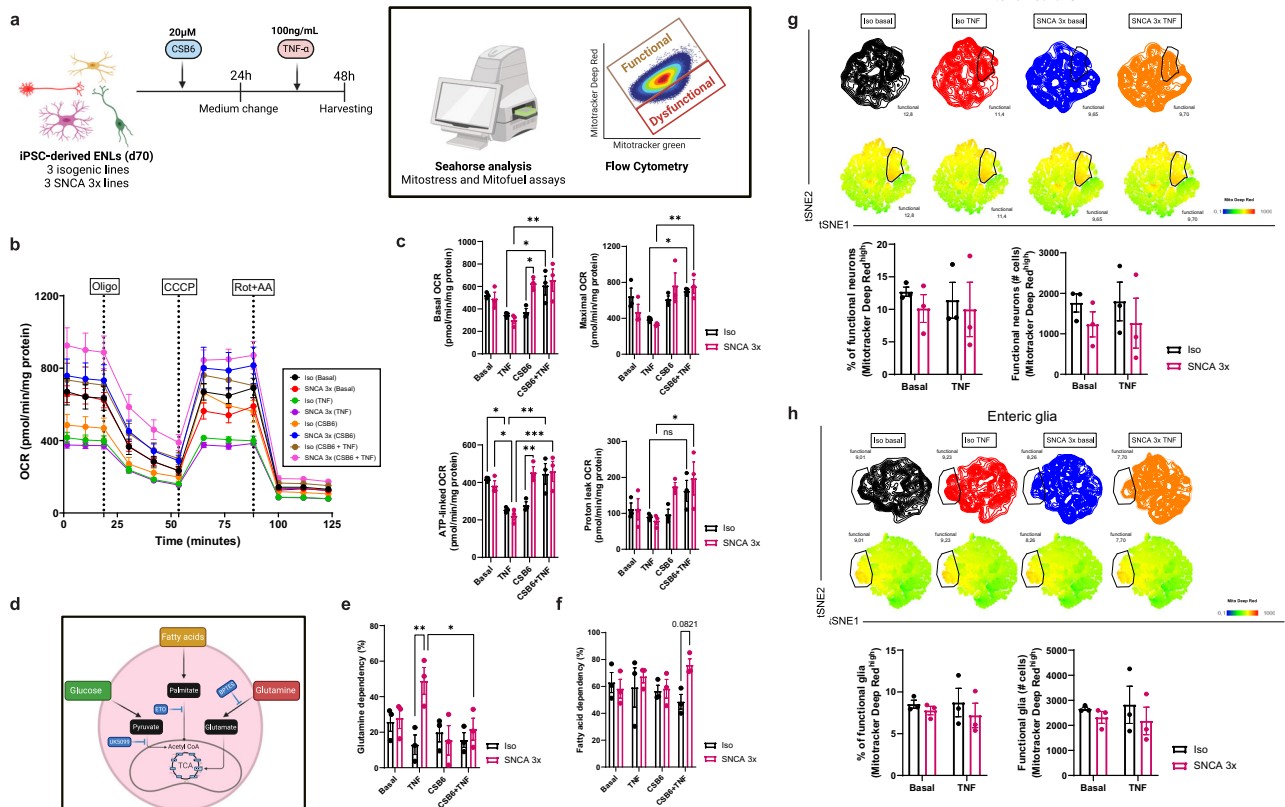

**Fig. 7 | TNF increases the reliance of SNCA 3x ENLs on glutamine oxidation, which is rescued by CSB6 treatment. a** Paradigm of seahorse and mitotracker assays. iPSC-ENLs were treated with Chicago Sky Blue 6b (CSB6) for 24 h, then the medium was replaced by one containing TNF for further 24h. Created in BioRender. Winner, B. (2026) https://BioRender.com/yvzd0da. **b** Mitostress assay. Curves showing the overall OCR for Iso and SNCA 3x ENLs with the different treatments specified. Oligo: Oligomycin; CCCP: carbonyl cyanide m-chlorophenylhydrazone; AA+Rot – antimycin-A plus Rotenone. **c** Quantification of parameters in (**b**), normalized to total protein ($n = 3$ lines/group, three differentiations). Mean ± SEM, two-way ANOVA (Sidak/Tukey's post-hoc). Basal respiration: Iso TNF vs. Iso CSB6 + TNF, $p = 0.0171$; Iso CSB6 vs. SNCA 3x CSB6, $p = 0.0163$; SNCA 3x TNF vs. SNCA 3x CSB6 + TNF, $p = 0.0014$. Maximal respiration: Iso TNF vs. Iso CSB6 + TNF, p = 0.0385; SNCA 3x TNF vs. SNCA 3x CSB6 + TNF, $p = 0.0048$. ATP-linked OCR: Iso Basal vs. Iso TNF, p = 0.0134; Iso TNF vs. Iso CSB6 + TNF, p = 0.0038; SNCA 3x Basal vs. SNCA 3x TNF, p = 0.0145; SNCA 3x TNF vs. SNCA 3x CSB6 + TNF, p = 0.0005.

Proton leak: SNCA 3x TNF vs. SNCA 3x CSB6 + TNF, $p = 0.0148$. **d** Paradigm of seahorse mitofuel assay. BPTES= Bis-2-(5-phenylacetamido-1,3,4-thiadiazol-2-yl) ethyl sulfide, ETO=etomoxir, TCA=tricarboxylic acid cycle. Created in BioRender. Winner, B. (2026) https://BioRender.com/z0fla0d. Quantification of the glutamine (**e**) and fatty acid (**f**) dependency analysis. $n = 3$ independent SNCA 3x and 3 isogenic lines per group, from two independent differentiations, mean ± SEM, by two-way ANOVA with Sidak and Tukey's post-hoc. Glutamine: Iso TNF vs SNCA 3x TNF, $p = 0.003$; SNCA 3x TNF vs SNCA 3x CSB6 + TNF, $p = 0.029$. Concatenated t-SNE plots from flow cytometry data of all lines highlighting Mito DeepRed[high] populations across genotypes and conditions and quantification of the percentage and total Mito DeepRed[high] cells in enteric neurons (CD56[+]CD24[high]) (**g**) and in enteric glia (CD56[+]CD24[low]) (**h**). $n = 3$ independent SNCA 3x and 3 isogenic lines per group, from two independent differentiations, mean ± SEM. Source data are provided as a Source Data file in **b**, **c**, **e**, **f**, **g** and **h**. For all figures, basal refers to cells treated with vehicle used to dilute the TNF and CSB6 (DPBS + 0.1% BSA).

---

under basal, TNF, or CSB6-only conditions, but showed a trend toward increased FAO dependency in SNCA 3x ENLs in the CSB6 + TNF group (Fig. 7F), possibly reflecting a compensatory metabolic adaptation.

To further characterize mitochondrial functional heterogeneity, we conducted flow cytometry analyses of Mitotracker Deep Red fluorescence, which is linked to changes in mitochondrial membrane potential, and applied t-SNE to visualize cell populations in iPSC-ENLs under basal and TNF–stimulated conditions. In line with Seahorse findings, SNCA 3x ENLs exhibited a trend toward reduced mitochondrial function already under basal conditions (Fig. 7G–H). Following TNF treatment, t-SNE contour maps revealed a visually distinct loss of Mitotracker[high] subpopulations, most evident in SNCA 3x enteric glia (Fig. 7H), though this was not statistically significant in bulk quantification.

Complementing these findings, t-SNE analysis of carnitine palmitoyltransferase IA (CPT1α) expression indicated that CPT1α was more prominently expressed in enteric neurons (Fig. S7F) than in glia (Fig. S7G). Following TNF exposure, t-SNE contour plots showed a

visually reduced CPT1α [high] population in SNCA 3x neurons, accompanied by a notable appearance of CPT1α [high] cells in glia (Fig. S7F–G). Although these population shifts were not significant in bulk analysis, their consistent visualization across replicates suggests a potential redistribution of FAO capacity between neurons and glia under inflammatory conditions, in line with the TNF–induced metabolic rewiring observed in SNCA 3x ENLs.

Collectively, these results demonstrate that TNF drives a shift toward glutamine oxidation in SNCA 3× ENLs, accompanied by altered mitochondrial integrity and a redistribution of CPT1α expression between neurons and glia. CSB6 treatment restores mitochondrial function and normalizes substrate utilization, underscoring its potential to counteract cytokine-induced metabolic reprogramming in α-synuclein-linked enteric pathology.

**Inflammation induces MAS suppression in the human gut**
To extend our observations to human intestinal inflammation, we examined independent transcriptomic datasets alongside tissue-level staining of ulcerative colitis (UC) patients' biopsies.

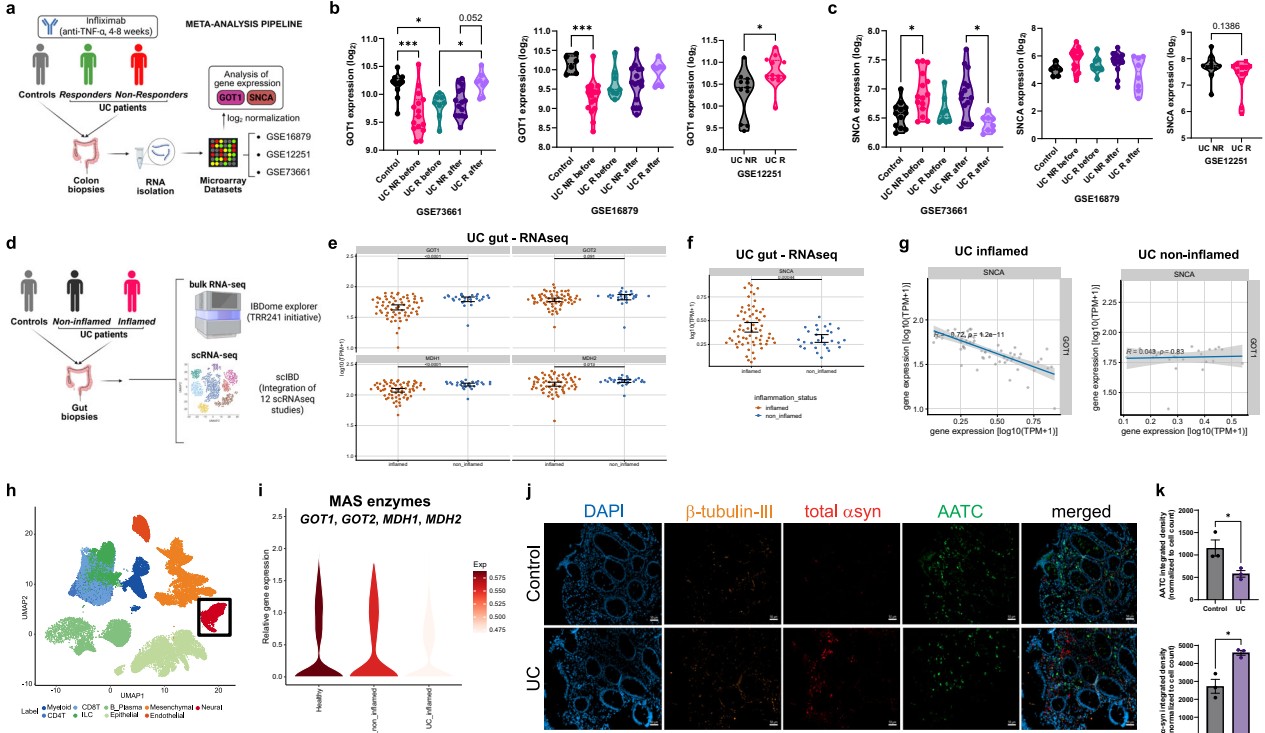

**Fig. 8 | Clinical relevance of MAS dysfunction in human inflamed gut.**
**a** Paradigm of meta-analysis of independent UC patient cohorts, responders or non-responders to infliximab therapy and controls (GSE16879, GSE12251, GSE73661). Created in BioRender. Winner, B. (2026) https://BioRender.com/jkisazh. **b** *GOT1* expression (log2 normalized) across datasets. Analyzed by Student's *t*-test or one-way ANOVA (Tukey's post-hoc). GSE73661: Control vs. UC NR before, *p* = 0.0002; Control vs. UC R before, *p* = 0.035; UC R before vs. UC R after, *p* = 0.0421. GSE16879: Control vs. UC NR before, *p* = 0.0007. GSE12251: *p* = 0.0158. **c** *SNCA* expression (log2 normalized) across datasets. Analyzed by t-test or ANOVA. GSE73661: Control vs. UC NR before, *p* = 0.0237; UC R after vs. UC R after, *p* = 0.0112. **d** Bioinformatic analysis paradigm of UC cohorts (inflamed/non-inflamed) using bulk RNA-seq (IBDome, https://ibdome.org/) and scRNAseq (scIBD, http://scibd.cn/). Created in BioRender. Winner, B. (2026) https://BioRender.com/h6tkzor. **e** Expression of MAS genes (*GOT1*, *GOT2*, *MDH1*, *MDH2*) in inflamed vs. non-inflamed UC tissue (IBDome, *n* = 66 inflamed, *n* = 28 non-inflamed). TPM normalized, log10 scale. Two-sided Wilcoxon-Mann-Whitney test. Error bars: 95% CI. Exact p-values are indicated in the figure. **f** *SNCA* expression in inflamed vs. non-inflamed UC tissue (IBDome, *n* = 66 inflamed, *n* = 28 non-inflamed). TPM normalized, log10 scale. Two-sided Wilcoxon–Mann–Whitney test. Error bars: 95% CI. **g** Correlation analysis between GOT1 and SNCA in inflamed and non-inflamed tissue (IBDome, *n* = 66 inflamed, *n* = 28 non-inflamed). Pearson test. The shaded area denotes the 95% confidence interval of the slope of the linear model. **h** UMAP plot generated by integrating the 12 scRNAseq studies from the scIBD platform, highlighting the neural cluster in red. **i** Mean expression value of the MAS enzymes *GOT1*, *GOT2*, *MDH1* and *MDH2* in healthy, UC non-inflamed and UC inflamed tissue. Data came from the scIBD platform. **j** Immunocytochemistry of healthy controls and UC patient gut tissue: β-tubulin III (orange), total α-syn (red), AATC (green), DAPI (blue). Scale bar: 50 μm. **k** Quantification of the immunocytochemistry shown is (J) for *n* = 3 controls and n = 3 UC patients, mean ± SEM. Integrated density was calculated using FIJI and normalized to the total cell count per image. Data analyzed by unpaired two-tailed *t*-tests. AATC, *\*p* = 0.04; α-syn, *\*p* = 0.01. Source data are provided as a Source Data file in **b, c, e, f, g, k**.

First, we assessed the expression of *GOT1* (AATC) and *SNCA* across three datasets of UC patients treated with infliximab. *GOT1* was consistently downregulated in patients who were non-responders (NRs). Critically, the GSE73661 dataset showed a significant restoration of GOT1 in clinical responders (Rs) following therapy (Fig. 8B). Conversely, *SNCA* trended toward upregulation in NRs, and in the same GSE73661 cohort, its expression was significantly lower in Rs compared to NRs post-treatment (Fig. 8C). This crucial inverse pattern defines patients who fail to respond to anti-TNF therapy. It directly links persistent TNF-driven inflammation to MAS suppression and α-syn accumulation.

We next confirmed these findings using our in-house bulk RNA-seq dataset (IBDome[53]) and the integrated scIBD database (Fig. 8D). In IBDome, MAS enzymes (*GOT1*, *MDH1*, *MDH2*) were reduced in inflamed versus non-inflamed tissue (Fig. 8E), whereas *SNCA* was increased in UC inflamed (Fig. 8F), also showing a negative correlation with *GOT1* which disappeared in the absence of inflammation (Fig. 8G). At the single-cell level, MAS enzyme expression was significantly decreased in neural clusters under inflammatory conditions (Fig. 8H–I).

Finally, immunostaining of colon biopsies from UC patients and non-UC controls revealed significantly reduced AATC and increased α-syn levels, along with an overall co-localization of α-syn and the neuronal marker beta-III tubulin, providing evidence for transcriptional findings at the tissue level (Fig. 8J-K).

In summary, our findings demonstrate that TNF reprograms metabolism in SNCA 3x ENLs by impairing MAS, reducing TCA cycle flux, and shifting mitochondrial dependency toward glutamine oxidation as a compensatory mechanism. This metabolic stress is linked to increased mitochondrial α-syn accumulation, oxidative stress, and loss of mitochondrial membrane potential. CSB6 counteracts these TNF-driven alterations by restoring mitochondrial respiration, reducing glutamine dependency and preventing mitochondrial dysfunction (Fig. 9). Importantly, by validating these signatures in human UC patients, across bulk and single-cell transcriptomics and immunostaining, our study reveals a conserved inflammation-induced MAS dysfunction linked to α-syn accumulation, extending the relevance of our findings beyond PD to broader contexts of intestinal inflammation in the human gut.

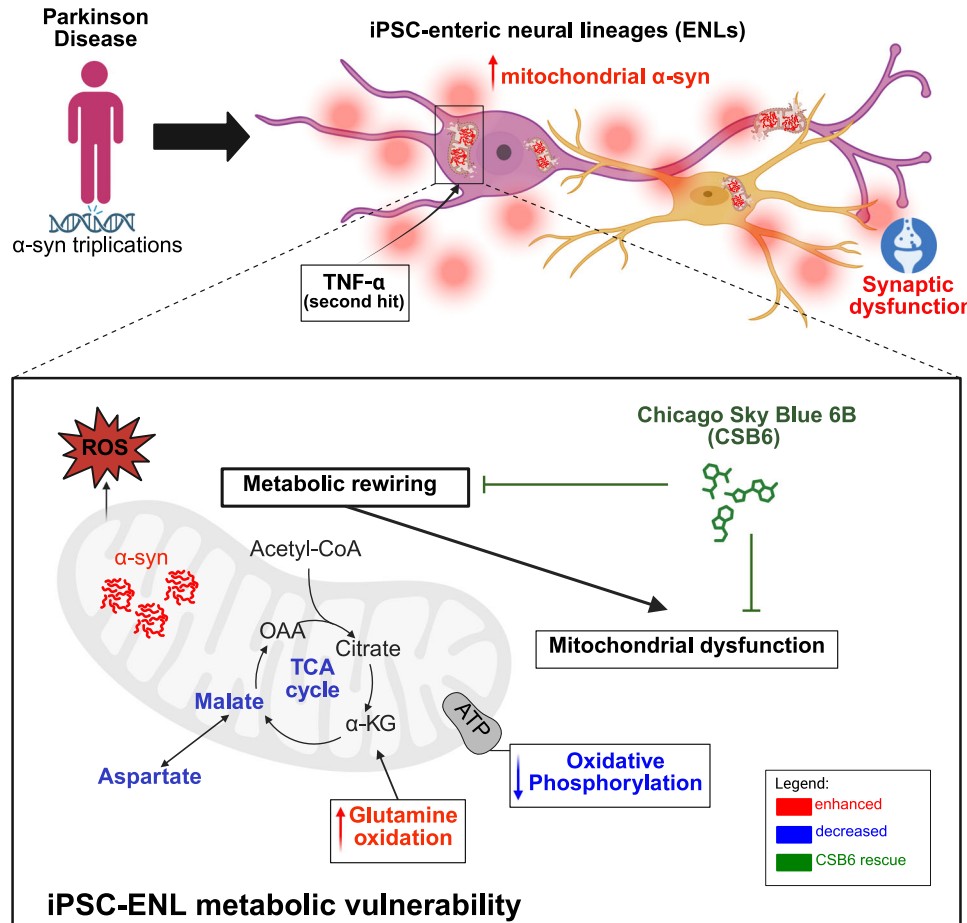

**Fig. 9 | Summary of findings.** Graphical summary of findings. SNCA 3x drives inflammatory-metabolic vulnerability in the ENS. Our multi-omics and functional approach using iPSC-derived ENLs revealed that α-syn accumulation induces basal mitochondrial and synaptic dysfunction, altering cell composition and transcriptional profiles in both enteric neurons and glia. Challenging these cells with the proinflammatory cytokine TNF acts as a second-hit, unmasking selective vulnerability in SNCA 3x ENLs, marked by enhanced α-syn-mitochondria interactions and a corresponding surge in oxidative stress-associated populations. This functional collapse is mechanistically driven by a disruption of the malate-aspartate shuttle

and the TCA cycle, forcing cells to become critically reliant on glutamine oxidation for energy. We demonstrate that the glutamate metabolism modulator Chicago Sky Blue 6B effectively restores mitochondrial function, reverses the TNF-driven glutamine dependency, and enhances metabolic flexibility, suggesting a targeted therapeutic strategy for α-syn-driven enteric pathology. Finally, we validate this core mechanism by showing that MAS suppression, linked to α-syn accumulation, is a conserved signature in inflamed human gut tissue. Created in BioRender. Winner, B. (2026) https://BioRender.com/d91886y.

## Discussion

In this study, we employed a multiomics-approach integrating single-cell transcriptomics, proteomics, metabolomics, and functional assays to gain a comprehensive understanding of α-syn-induced dysfunction in the human ENS, something that has been challenging to achieve with previous methodologies[54]. Our findings reveal that α-syn accumulation establishes a pre-existing metabolic vulnerability in iPSC-derived ENLs which TNF then exploits to drive mitochondrial dysfunction and metabolic reprogramming. This is characterized by MAS and TCA cycle disruption, forcing the ENLs to rely on glutamine oxidation for energy production. These data demonstrate that α-syn accumulation in ENLs drives vulnerability in the ENS through this inflammatory-metabolic crosstalk. Notably, targeting glutamate metabolism with CSB6 restored mitochondrial metabolism, highlighting a potential therapeutic approach for early PD-related ENS dysfunction.

Our scRNAseq analysis demonstrated that increased α-syn levels in SNCA 3x induce mitochondrial and metabolic stress in a cell-type-specific manner. Enteric neurons show downregulation of genes controlling acetyl-CoA, and steroid metabolism, alongside upregulation of mitophagy and fission regulators, consistent with mitochondrial fragmentation and energy insufficiency that may underlie the selective

vulnerability of specific neuronal subpopulations[55]. In contrast, enteric glia exhibited transcriptional signatures indicative of mitochondrial adaptation, including increased expression of OXPHOS, antioxidant, and mitochondrial protein import genes. These responses, while suggestive of a potential protective adaptation, remain open for further investigation. Beyond these intrinsic alterations, and consistent with PD-related synaptic remodeling[56], SNCA 3x also reshaped inter and intracellular communication, modulating enteric neuron-glia interactions[57] through VEGF, EGF, and BMP pathways, linking subpopulation vulnerability to rewired signaling networks. Interestingly, cross-comparison with cortical and midbrain SNCA 3x organoids revealed conserved disruptions in synaptic and cholesterol metabolism pathways, underscoring shared α-syn-driven vulnerabilities across neuronal systems. Collectively, these findings suggest that α-syn accumulation perturbs mitochondrial homeostasis and enteric neuron–glia communication, establishing a mechanistic framework for early ENS dysfunction in PD.

TNF unmasked genotype-specific α-syn accumulation in SNCA 3x ENLs, despite not significantly changing the immunoreactivity of the cells. Importantly, when analyzing α-syn levels at the cellular level using flow cytometry, we observed that TNF exposure unveiled

increased α-syn levels in both enteric neurons and glia, indicating that both are affected by the pathology. Interestingly, differences in α-syn aggregation were only detectable using highly sensitive ELISA assays, potentially indicating that our model represents a pre-aggregation stage. In contrast to classical CNS models[25], where microglia actively regulate α-syn aggregation and propagation[14], the presence of enteric glial cells in ENLs suggests the activation of compensatory mechanisms that may help buffer excessive α-syn aggregate formation. Additionally, α-syn was previously described primarily in its monomeric form in the ENS, implying that the native state of the protein differs from what is observed in the CNS[58]. Notably, TNF levels are elevated in the serum of PD patients and correlate with disease progression[59,60]. Moreover, TNF is synthesized by intestinal epithelial cells[61], suggesting that it can locally impact gut inflammation and enhance α-syn accumulation in the ENS. A compelling link between intestinal TNF and PD pathophysiology comes from studies showing that treatment with gold-standard anti-TNF therapies in IBD patients reduced PD risk by nearly 80%[62]. Although TNF did not exert major effects on neuronal activity, our MEA data suggested a pronounced basal synaptic dysfunction phenotype in SNCA 3x ENLs, an important observation given that over 80% of PD patients experience constipation, which can be directly associated with ENS activity dysfunction[63].

Our multiomics analysis not only validated the scRNAseq findings but also provided deeper insights into a TNF-induced metabolic rewiring in SNCA 3x ENLs, bringing important insights into metabolic vulnerability in the ENS in the context of early PD. Using proteomics, we observed that increased α-syn impacted neuronal function, in line with the previously mentioned transcriptomics signatures as well as the neuronal activity data. This is also in line with recent findings using cortical organoids[30]. The addition of TNF highlighted oxidative stress and mitochondrial dysfunction as a response to inflammation-driven α-syn toxicity. Additionally, using metabolomics, we uncovered a hypometabolic state in SNCA 3x ENLs, characterized by a major downregulation in metabolites related to TCA and MAS. Given that NAD+ is a critical cofactor for TCA cycle enzymes such as malate dehydrogenase and its role in electron transport chain function, its depletion could directly impair mitochondrial substrate oxidation and ATP production. In particular, we found reduced NAD+ levels, with NADH remaining unaltered, suggesting impaired NAD+ regeneration under inflammatory stress, as observed in PD and other neurodegenerative diseases[64–66]. The failure to replenish NAD+ pools impairs mitochondrial respiration, likely through disruptions in the MAS, which facilitates the transfer of cytosolic NADH into mitochondria to maintain redox balance and sustain OXPHOS. These findings align well with a previous study using iPSC-derived neuronal models of sporadic PD[67], which also highlighted cellular metabolic disruption as a clear phenotype. Interestingly, in line with our proteomics data, expression of the *GOT1* gene, coding for AATC, was lower in the brain of PD patients compared to controls[68]. Also corroborating these results, a study using a mouse model for prodromal PD showed that TCA dysfunction influences the onset and progression of α-syn pathology[69]. Therefore, our findings emphasize the inflammatory-driven metabolic reprogramming in SNCA 3x ENLs, pointing to the interplay of neuroinflammation, mitochondrial dysfunction, and MAS/TCA cycle disruption in enteric α-synucleinopathies.

Deepening into the mechanism, our data indicated that already at basal levels, there is an increased interaction between α-syn and mitochondria in SNCA 3x ENLs, consistent with previous findings[70], and this effect was further amplified upon TNF stimulation. TNF exposure unmasked specific mitochondrial superoxide-producing populations in SNCA 3x enteric neurons and glia, reinforcing the role of inflammation in driving oxidative stress and mitochondrial dysfunction[71]. Functionally, TNF broadly impaired mitochondrial respiration in both genotypes, most notably ATP-linked oxygen consumption. Using a data-driven approach informed by our single-cell

analysis, we selected CSB6, a vesicular glutamate transporter inhibitor previously shown to exert neuroprotective effects in PD models[72], to test whether modulation of glutamate metabolism could rescue this metabolic phenotype. TNF selectively increased the reliance of SNCA 3x ENLs on glutamine oxidation, consistent with an anaplerotic shift compensating for impaired MAS activity[73]. CSB6 effectively normalized this metabolic profile, reinstating balanced substrate utilization and enhancing overall mitochondrial function. Notably, even in the absence of TNF, CSB6 alone increased both basal and ATP-linked respiration in SNCA 3x ENLs, suggesting an intrinsic capacity to boost mitochondrial performance in metabolically compromised cells. Although FAO dependency did not differ significantly across genotypes or treatments, a mild trend toward increased FAO reliance was observed in TNF-treated SNCA 3x ENLs under CSB6, potentially reflecting partial restoration of metabolic flexibility[74]. In parallel, single-cell flow cytometry analyses revealed a TNF-dependent redistribution of CPT1α expression, with CPT1α[high] populations decreasing in neurons and emerging in glia. While not significant in bulk quantification, this reproducible pattern across replicates may indicate a cell-type–specific reallocation of FAO capacity under inflammatory stress. Together, these findings indicate that TNF drives a glutamine-dependent metabolic reprogramming in SNCA 3x ENLs, accompanied by altered mitochondrial integrity and a redistribution of FAO potential between neurons and glia, while CSB6 mitigates the associated mitochondrial and bioenergetic deficits.

Importantly, this inflammatory MAS-suppression mechanism was also observed in vivo. We analyzed gut tissue from UC patients, where MAS-associated enzymes expression was downregulated while α-syn was upregulated, particularly in those with persistent inflammation or non-responders to infliximab therapy. This aligns with recent observations of elevated pathological α-syn in the colon of UC patients[75]. Therefore, these findings extend the relevance of our model beyond PD and highlight the clinical translatability of targeting inflammation-induced metabolic dysfunction in the ENS.

In conclusion, our findings indicate that α-syn accumulation and TNF-induced inflammation synergize to drive ENS dysfunction, supporting a dual-hit hypothesis for PD onset. This interplay reinforces the ENS as an active player in PD pathophysiology, shifting the paradigm from a passive receptor of inflammatory signals to an initiator of metabolic disruption. Our iPSC-derived ENL model provides a powerful platform for investigating early-stage PD mechanisms and suggests that targeting glutamate metabolism may offer a new therapeutic perspective for ENS-related dysfunction in PD. Furthermore, further investigations on how this metabolic vulnerability impacts the spread and propagation of α-syn pathology represents a crucial next step toward fully understanding disease progression especially in the context of body-first PD.

## Limitations of the study

While the iPSC-derived ENL broadly recapitulates the human ENS, the presence of mixed neuronal and glial cells in iPSC-ENLs makes it challenging to separate mechanistic effects between these cellular populations. We tried to address this by extensively analyzing our scRNAseq data and by including a flow cytometry analyses followed by separation of the TNF effects on the two cell types that were incorporated throughout the manuscript. Nevertheless, some functional analyses such as the MEA and Seahorse cannot be deconvoluted, representing rather the overall phenotype of iPSC-ENLs. Moreover, the high variability between differentiated iPSC lines is a well-known problem in the field and cannot be excluded from data interpretation[76]. Furthermore, the small number of biological replicates used limited the statistical power for our omics analyses, therefore the findings should be interpreted as discovery-level candidates. Additionally, it is very difficult to obtain gut specimens from early PD patients that fully capture the ENS extension.

 

CSB6 is a compound that increased overall mitochondrial function in our model and there is evidence for effectiveness coming from other PD models[72]. Thinking about clinical translation, CSB6 served as a proof-of-concept in our study and previously approved compounds could be repurposed to counteract the metabolic alterations observed in the iPSC-ENLs in future approaches. However, CSB6 showed a high basal fluorescence in the APC channel, which hampered flow cytometry analyses. Despite these limitations, our study provides a mechanistic framework for understanding enteric neural vulnerability in PD and highlights MAS disruption as a tractable metabolic node. Importantly, the conservation of this mechanism in intestinal inflammation extends its relevance beyond PD, suggesting broader implications for ENS dysfunction in chronic inflammatory disorders.

## Methods

### Ethics statement

All protocols used in this study as well as the use of human material were approved by the Ethics Committee of the Medical Faculty of the Friedrich-Alexander-Universität Erlangen-Nürnberg and the Universitätsklinikum Erlangen (#22-289-Bp and #49_16B). The human cells used in this study were handled in accordance with the principles outlined in the Declaration of Helsinki. All iPSC lines used in this study are commercially available.

### iPSC generation, characterization and culture methods

**iPSC reprogramming, generation of isogenic controls.** B-lymphocytes containing a triplication in the *SNCA* gene were obtained from the Coriell NINDS and NIGMS Human Genetic Cell Repositories: GM15010 (3x-1), ND00196 (3x-2), ND00139 (3x-4, referred to as 3x-3 throughout this paper). Further phenotypic and genotypic details on these lines and subjects are available on https://www.coriell.org under the aforementioned accession numbers. See key resources table for more details. Reprogramming into iPSC was performed by transfection of B-lymphocytes with episomal plasmids containing Oct3/4, L-Myc, Sox2 and Klf4, as previously described[25]. Isogenic controls were generated using a dual nickase CRISPR/Cas9 strategy to disrupt exon 2 of the *SNCA* gene, fully characterized and validated in a previous publication[25]. Copy number variation and cell karyotyping analyses revealed absence of chromosomal abnormalities. Iso and SNCA 3x iPSC lines were generated in the laboratory of Joseph R. Mazzulli (The Ken and Ruth Davee Department of Neurology, Northwestern University Feinberg School of Medicine, Chicago, IL 60611, USA) and provided to this study by Friederike Zunke.

### iPSC culture maintenance

iPSCs were cultured in Geltrex (Thermo Fisher)-coated 6-well plates with Essential 8 Flex medium (Thermo Fisher), maintained at 37 °C and 5% $CO_2$, with media change performed every two days. For thawing, cryovials were warmed in a 37 °C water bath, and cells were resuspended in fresh medium containing 10 µM ROCK inhibitor (Tocris, Cat#1254). Passaging occurred at 80%-90% confluency using ReLeSR (Stemcell, Cat#100-0483), with cells split at a 1:2 or 1:3 ratio. Cells were routinely tested for mycoplasma infection. For freezing, cells were detached with ReLeSR and resuspended in a freezing medium containing knockout serum replacement (KOSR, Thermo Fisher, Cat#10828028) + 10% DMSO before storage at −150 °C.

### iPSC characterization

iPSCs were characterized using flow cytometry staining for Nanog and Lin28A. iPSCs were initially treated with Accutase (Gibco, USA), washed with 1× PBS, centrifuged at 300 g at room temperature for 5 min and then resuspended in FC buffer (2% FCS, 0.01% sodium azide in PBS). We then counted 500,000 cells per line and proceeded with fixation and permeabilization with 100 µl BD Fixation/Permeabilization Solution (BD Bioscience) for 10 minutes, followed by addition of 1 ml

BD Perm/Wash Buffer (BD Bioscience), incubation for 5 minutes and centrifugation at 300 g for 3 min. For intracellular staining of iPSCs, we used a combination of Nanog and Lin28A primary antibodies (RRID: AB_2738303 and RRID: AB_2784439, both 1:100) in BD Perm/Wash Buffer for 30 min, followed by washing and resuspension in 300 µl FACS buffer. Cells were then acquired using a Cytoflex S cytometer (Beckman Coulter) and analyzed with the CytExpert 2.4 software. At least 30,000 events per sample were analyzed.

### Differentiation of iPSCs into ENLs

The ENS differentiation protocol was adapted from Barber et al.[23], using the novel chemically defined method. ENS induction began on day 0 with a nearly confluent iPSC monolayer and continued for 12 days with stepwise culture transitions. On day 0, Essential 8 medium (Thermo Fisher) was replaced with 2 ml of Cocktail A [Essential 6 medium (Thermo Fisher), 10 µM SB431542 (Miltenyi Biotec, Cat#130-106-543), 600 nM CHIR99021 (Tocris, Cat#4423), 1 ng/ml BMP4 (Peprotech, Cat#120-05ET), 100µg/mL normocin (InvivoGen, Cat#ant-nr-2)], followed by a switch to Cocktail B [Essential 6 medium (Thermo Fisher), 10 µM SB431542, 1.5 µM CHIR99021, 100µg/mL normocin] on day 2, with media renewal on day 4. On day 6, Cocktail B was replaced with 4 ml of Cocktail C [Essential 6 medium (Thermo Fisher), 10 µM SB431542, 1.5 µM CHIR99021, 1 µM RA (Sigma, Cat#R2625), 100µg/mL normocin], with subsequent exchanges on days 8 and 10. On day 12, ENC spheroids were formed by detaching the ENC monolayer using Accutase for 30 min at 37 °C, followed by centrifugation at 300 g for 3 min, resuspension in NC-C medium [Neurobasal medium (Thermo Fisher), 1x N2 supplement (Thermo Fisher, Cat#17502048), 1x B27 supplement w/o Vit. A (Thermo Fisher, Cat#12587010), 1x GlutaMax (Thermo Fisher, Cat#35050038), 1x MEM NEAA (Thermo Fisher, Cat#11140050), 10 ng/ml FGF2 (Peprotech, Cat#100-18B), 3 µM CHIR99021, 100µg/mL normocin], and transfer to ultra-low attachment (ULA) plates. Spheroids formed within two days, with media replacement using NC-C medium. On day 15, EN induction began, involving dissociation of spheroids using Accutase, centrifugation at 300 g, and resuspension in EN-C medium [Neurobasal medium (Thermo Fisher), 1x N2 supplement, 1x B27 supplement with Vit. A (Thermo Fisher, Cat#17504044), 1x GlutaMax, 1x MEM NEAA, 100 µM AA (Sigma, Cat#A4544), 10 ng/ml GDNF (Peprotech, Cat#450-10), 100µg/mL normocin]. Cells were counted using a fluorescence cell counter and seeded onto Geltrex-coated plates at 60,000 cells/cm².

### Human tissue samples

Human tissue samples were obtained from patients undergoing routine biopsy collection at the Universitätsklinikum Erlangen, after written informed consent was obtained from the patients. Gut tissues were used for paraffin-embedded sections. The diagnoses of UC were based on clinical, endoscopic, radiology, and histologic findings. Data were pseudonymized. All controls used were non-UC samples. Metadata including diagnosis, age, sex and disease duration is provided in Supplementary Data 11.

### RNA extraction, cDNA synthesis and RT-qPCR

Reverse transcription-quantitative PCR (RT-qPCR) analysis was performed starting with RNA extraction using Trizol (Invitrogen) and the RNeasy Mini Kit (Qiagen, Cat#74104). Cells were homogenized in Trizol, followed by chloroform addition and phase separation by centrifugation at 12,000 g, 4 °C for 15 min. The RNA-containing aqueous phase was collected, mixed with 70% ethanol, and processed through RNeasy columns with sequential washes using buffer RW1 and buffer RPE. RNA was eluted with RNase-free water, its concentration measured using the NanoPhotometer NP80 (Implen), and stored at −80 °C. For cDNA synthesis, the QuantiTect Reverse Transcription Kit (Qiagen, Cat#205311) was used, with 1000 ng starting RNA concentration per reaction. After DNA removal with gDNA wipeout buffer, reverse

transcription was carried out at 42 °C for 30 min, followed by inactivation at 95 °C for 3 min, and storage at −20 °C. RT-qPCR was performed in triplicates for target genes (*TUBB3*, *GFAP*, *PHOX2B*, *ELAVL4*, *HOXB3*), with *HPRT* and *RPLPO* as housekeeping genes. Primer sequences used are listed in Supplementary Data 1. A primer mix (10 μM) was prepared, and 1 μl cDNA per sample was loaded onto a 384-well plate. The qPCR master mix was added, the plate was sealed, centrifuged, and run on the LightCycler480 equipment (Roche). Ct values were analyzed using the delta-delta Ct method, and fold change values were calculated based on the expression values of the Iso group at day 6 of differentiation.

### Immunocytochemistry

Cells were fixed on day 70 after start of differentiation using 4% paraformaldehyde (PFA) for 30 min at room temperature (RT) (500 μl per well in an ibidi 24-well plate), washed three times in DPBS, and stored at 4 °C until staining. Permeabilization was performed with 0.3% TritonX-100 in DPBS for 10 min at RT, followed by blocking with 5% donkey serum in DPBS with 0.3% TritonX-100 for 30 min. Cells were then incubated overnight at 4 °C with the primary antibodies (mouse anti-a-syn 2A7 1:100, RRID: AB_1555287; mouse anti-HuC/D 1:200, RRID: AB_221448; rabbit anti-GFAP 1:500, RRID: AB_10013382). The following day, cells were washed, incubated for 2 h at RT with secondary antibodies (AF488 anti-mouse; AF647 anti-rabbit, all 1:500), and counterstained with DAPI (1:5000 in DPBS) for 2 min. After final washes, samples were mounted in Mowiol (Sigma, Cat#81381) for long-term storage at 4 °C. Imaging was performed using the Super Resolution Evident SR Spinning Disk microscope (Olympus) and the cellSens Software and processing was performed using FIJI (ImageJ).

Paraffin-embedded tissue sections from UC patients and controls were dewaxed and rehydrated. Samples were then fixed in a methanol solution for 20 min and permeabilization was performed with 0.1% TritonX-100 in TBS for 10 min at RT, followed by blocking with 10% Roti-block (Roth, Cat#A151.1, in a 2%BSA TBST solution for 15 min. The sections were then incubated overnight at 4 °C with the primary antibodies (mouse anti-tubulin beta-III, 1:250, RRID: AB_2313773; rat-anti-alpha-synuclein 15G7, 1:10, RRID: AB_11180660; rabbit-anti-aspartate aminotransferase, 1:100, RRID: AB_3598484). The following day, cells were washed, incubated for 2 h at RT with secondary antibodies (AF488 anti-rabbit, RRID: AB_2535792; AF546 anti-mouse, RRID: AB_11180613; AF647 anti-rat, RRID: AB_2910635, all 1:500), and counterstained with DAPI (1:2500 in TBS) for 5 min. Finally, sections were mounted with DAKO fluorescence mounting medium (Dako, Cat #S3023) and kept at 4 °C. Immunofluorescence images were acquired using a fluorescence microscope (Observer. Z1, Zeiss) and processing was performed using FIJI (ImageJ). In total, we analyzed $n = 5941$, $n = 3795$ and $n = 4220$ cells for controls 1, 2 and 3, respectively and $n = 5797$, $n = 5519$ and $n = 6089$ cells for UC 1, 2 and 3, respectively.

### Single cell RNA sequencing

**Sample preparation and sequencing.** On day 70 of differentiation, we obtained single cells from iPSC-ENLs using a papain-DNAse digestion protocol. Cells were then cryopreserved in a medium containing 90% fetal bovine serum and 10% DMSO and sent out for library preparation and sequencing at GENEWIZ (Leipzig, Germany) with 6000 cells targeted per sample and 50,000 reads targeted per cell. RNA samples were quantified using Qubit 4.0 Fluorometer (Life Technologies, Carlsbad, CA, USA) and RNA integrity was checked with RNA Kit on Agilent 5300 Fragment Analyzer (Agilent Technologies, Palo Alto, CA, USA).

rRNA depletion was performed using NEBNext rRNA Depletion Kit (Human/Mouse/Rat). RNA sequencing library preparation used NEBNext Ultra RNA Library Prep Kit for Illumina by following the manufacturer's recommendations (NEB, Ipswich, MA, USA). Briefly, enriched RNAs were fragmented according to manufacturer's instruction. First

strand and second strand cDNA were subsequently synthesized. cDNA fragments were end repaired and adenylated at 3′ends, and universal adapter was ligated to cDNA fragments, followed by index addition and library enrichment with limited cycle PCR. Sequencing libraries were validated using NGS Kit on the Agilent 5300 Fragment Analyzer (Agilent Technologies, Palo Alto, CA, USA), and quantified by using Qubit 4.0 Fluorometer (Invitrogen, Carlsbad, CA).

The sequencing libraries were multiplexed and loaded on the flow cell on the Illumina NovaSeq X plus instrument according to manufacturer's instructions. The samples were sequenced using a 2 × 150 Pair-End (PE) configuration v1.5. Image analysis and base calling were conducted by the NovaSeq Control Software v1.7 on the NovaSeq instrument. Raw sequence data (.bcl files) generated from Illumina NovaSeq was converted into fastq files and de-multiplexed using Illumina bcl2fastq program version 2.20. One mismatch was allowed for index sequence identification.

### Initial processing, quality control, and clustering of scRNAseq data

The fastq files were processed using CellRanger (v7.1.0) from 10X Genomics, with reads aligned to the GRCh38 genome assembly to generate gene counts per cell. Subsequent processing, analysis, and visualization were performed in *R* (v4.3.3) using Seurat (v5.1.0). To ensure quality control, outlier cells with <200 genes, > 7500 reads or a mitochondrial percentage > 5% were filtered out. Potential doublets were removed using *scDblfinder*. Following quality control, batch effect across samples was mitigated using Seurat's integration workflow. Data normalization was carried out using the *SCTransform* algorithm. Features to use when integrating multiple datasets were identified using *SelectIntegrationFeatures*. Integration anchors computed using the *FindIntegrationAnchors* function were subsequently used by the *IntegrateData* function to generate a batch-corrected, integrated expression matrix. Principal component analysis (PCA) was applied for dimensionality reduction. Clustering was subsequently performed computing the shared nearest neighbor (SNN; *FindNeighbors*, k = 20) and applying the graph-based Leiden algorithm (*FindClusters*, resolution = 0.1) to determine the cell subtypes, followed by visualization with Uniform Manifold Approximation and Projection (UMAP). Initial annotation of the clusters was performed based on the expression of canonical markers[23,26].

### Subclustering analysis

To further resolve cellular heterogeneity, subclustering was performed on the enteric neurons and glia cell populations using Seurat (v5.1.0). Here, the enteric neurons and glial clusters were extracted from the integrated Seurat object using the *subset* function. Data normalization was conducted using the *NormalizeData* function, followed by identification of highly variable features via *FindVariableFeatures* with the 'vst' selection method. As in the previous analysis, the data were scaled, and PCA-based dimensionality reduction was performed. Clustering was carried out by computing the SNN (*FindNeighbors*, k = 20) and applying the Leiden algorithm (*FindClusters*, resolution 0.5). The processed objects were saved in H5Seurat format using the *SaveH5Seurat* function.

### Mapping of scRNAseq dataset to a reference dataset

Mapping of the scRNAseq dataset to the reference scRNAseq dataset from Drokhylansky et al.[27], Patikas et al.[29] and Jin et al.[30] was performed using SingleR[77] for cell-type annotation. The raw counts and the corresponding cell-type annotations from the reference dataset were converted to a SingleCellExperiment (SCE) object and log-normalized using *logNormCounts*. Query datasets of enteric neurons and glia were similarly transformed into SCE objects and subsetted into Iso and SNCA 3x conditions. SingleR classification was conducted separately for Iso and SNCA 3x samples to obtain cell-type predictions.

Visualization included heatmaps of classification scores (*plotScoreHeatmap*) and log-transformed assignment frequencies (*pheatmap::pheatmap*), as well as score distribution plots (*plotScoreDistribution*).

## Marker identification and enrichment analysis for the subclusters

Using the *FindMarkers* function in Seurat, markers associated with the subcluster of the enteric neuronal and glial populations were identified. The analysis was limited to genes expressed in at least 25% of cells (min.pct = 0.25) and exhibiting a minimum log fold-change (logFC) of 1.0 between groups. Only genes with an adjusted $p < 0.05$ were considered significant markers. Enrichment analysis was conducted using Gene Ontology (GO), KEGG, and Reactome pathway databases using clusterProfiler (v4.10.1). Enrichment terms with a Benjamini-Hochberg corrected $p < 0.05$ were considered significant. Enrichment results were visualized using *ggplot2* or the *sc.pl.dotplot* function from scanpy (v1.11).

## Differential expression between conditions

Differential expression analysis between the SNCA 3x and Iso conditions was performed using the *FindMarkers* function in Seurat, with a minimum of 25% cells expressing a given gene (min.pct = 0.25) and a minimum absolute log fold-change threshold of 1. Significant genes were selected based on an adjusted $p < 0.05$, calculated using a Wilcoxon Rank Sum test followed by Bonferroni correction. We used ComplexHeatmap (v2.18.0) to display significant mitochondria-related DEGs across the enteric neuron and glia subclusters.

## Compositional analysis

Compositional analysis was performed using the scProportionTest package[78]

(v0.0.0.9000), starting with the metadata containing the subcluster and condition information for enteric neurons and glia. The *sc_utils* function was applied to process the metadata for both datasets, and permutation testing was then conducted to evaluate whether the proportions of cells in specific clusters differed significantly between the two conditions (Iso vs. SNCA 3x). The analysis was run with a false discovery rate (FDR) threshold of 0.05 and a log2 fold-change threshold of 1.0.

The differential abundance analysis was performed using the MiloR[79] method to identify local changes in cell proportions. For our analysis, we used the first 30 PCA dimensions and set the neighborhood size based on k = 50 nearest neighbors with a proportional overlap of 0.1. The number of cells within each neighborhood was then counted per biological replicate. Differential abundance was tested using a generalized linear model that accounts for the batch effect, followed by a spatial FDR correction to the p-values, which allowed us to identify regions on the UMAP plots where the SNCA 3x and Iso cell populations significantly differed.

## Intercellular communication analysis

Cell-cell communication analysis was performed using the CellChat package[37] (v2.1.2) to investigate signaling interactions among proliferating cells, enteric neurons and glia, as well as interactions within the subclusters of the latter two populations. Data from both Iso and SNCA 3x conditions were processed individually. For each condition, the dataset was imported into a CellChat object, followed by standard preprocessing steps including normalization and the identification of overexpressed genes and receptors. Communication probability and cellular communication network were then inferred by calculating the probabilities of ligand–receptor interactions based on cell-specific gene expression profiles. Following network aggregation, a merged CellChat object was generated to compare communication differences between Iso and SNCA 3x. Additionally, number of interactions and interaction strength along with the top-ranked signaling pathways were identified and plotted for visualization.

## Pseudotime analysis of mitochondrial-linked genes

Trajectory analysis of enteric neurons and glia populations was conducted using Slingshot[42] (v2.10.0). Lineages and pseudotime trajectories were inferred with the *slingshot* function, specifying the root cluster based on entropy analysis (TSCAN package v1.40.1). Mitochondrial marker genes (*DNM1L*, *COX4I1*, *ATP5F1B*, *NDUFA9*, *PPARGC1A*, *PINK1*) were analyzed in enteric neurons and glia by assessing their expression across pseudotime in Iso and SNCA 3x conditions. Pseudotime values were extracted using slingPseudotime(), averaged across lineages, and validated against cell counts. Linear regression (lm(*expression ~ pseudotime * group*)) was performed to assess interactions between pseudotime and condition, followed by FDR correction of p-values to analyze the statistical differences in mitochondrial associated genes expression dynamics between conditions.

## Metabolic flux analysis

We used COMPASS[80] (v0.9.10.2) to integrate scRNAseq data with flux balance analysis, inferring metabolic states in SNCA 3x and Iso enteric neurons and glial cells to compare their metabolic differences. Using a genome-scale metabolic model (GSMM), COMPASS allowed us to integrate transcriptomics data with metabolic network constraints to compute "potential activity" scores for thousands of metabolic reactions across single cells. These scores quantify the degree to which the transcriptomics profiles support flux through specific metabolic reactions. We compared the estimated Log2Fold changes of the metabolic reactions for specific pathways and plotted the difference in Log2Fold values to visualize the changes between reaction scores in SNCA 3x vs. Iso cells. We computed the logarithm of the Wilcoxon adjusted *p*-value to filter reaction differences. A threshold of 1.3 (corresponding to an adjusted *p*-value of 0.05) was used to identify significant reactions, where values above this threshold indicated evidence of differential activity between groups.

## ELISA for total and aggregated α-syn

ELISA was used to assess the intracellular levels of total and aggregated α-syn in iPSC-ENLs. Protein was extracted with the buffer provided by the α-syn ELISA kit (#ab260052, Abcam) and quantified using BCA assay (Thermo) for normalization of the data. We used 50 μg/mL of extracted protein for the total α-syn analysis and 300 μg/mL for the aggregated α-syn (#448807, Biolegend) analyses. Reactions were performed according to the manufacturer's instructions, with data analyzed in triplicates.

## Flow cytometry assays

Our flow cytometry data analysis was based on the protocol developed by Windster et al.[50]. Cells were washed twice with PBS 1x and then incubated with Accutase for 30 min at 37 °C and 5% $CO_2$. Subsequently 2 mL of ENC Media was added to inhibit Accutase and cells were centrifuged at 300xg for 3 min, counted and then plated in a 96 well plate. Zombie NIR viability staining dye (Biolegend Cat:423105) was added for 20 min (1:2000) followed by two washing steps with FACS buffer (2% FCS, 0.01% sodium azide in PBS). Next, antibodies and/or mitochondrial probes were incubated for 30 min at 37 °C and 5% $CO_2$ at the following concentrations: Mitotracker Deep Red, Green and MitoSox at 1:10000 (ThermoFisher); CD56 (Biolegend, RRID: AB_2565633, 1:100); alpha-synuclein 2A7 (Novus, RRID: AB_3198368, 1:100); CD24 (BD Biosciences, RRID: AB_2737795, 1:100); CPT1α (BD Biosciences, RRID: AB_3684993, 1:200). Cells were acquired in the MACSQuant Analyzer 16 Flow Cytometer (Miltenyi). Results were anaylzed in FlowJo 10.1 where the dimensional reduction analysis was also employed. For this, we used unsupervised tSNE algorithm where neuronal

(CD56⁺CD24^high) and glial (CD56⁺CD24^low) cells were concatenated and submitted to tSNE analysis. Plots were then generated and populations selected based on the markers of interest.

### Enzyme activity assays

We used two enzymatic assay kits to assess the activity of citrate synthase (abcam, #ab119692) and succinate dehydrogenase (abcam, #ab228560) in iPSC-ENLs, following the manufacture's guidelines. Protein was extracted with the buffer provided by the kits and quantified using BCA assay (Thermo) for normalization of the data. We used 50 μg/mL of protein per well for the citrate synthase assay and 300μg/mL of protein per well for the succinate dehydrogenase assay, with data analyzed in triplicates.

### Dot-blotting analysis

Cell lysates were diluted in ddH$_2$O to achieve 8 μg of total protein in a 3 μL drop, which was then spotted onto a nitrocellulose membrane and air-dried for at least 4 h. Blocking was performed with 5% non-fat dry milk in TBS for 1 h at room temperature, followed by incubation with the primary antibody (mouse anti-Syn-1, RRID: AB_398108, 1:1000) in 5% non-fat milk in 0.1% TBS-Tween20 overnight at 4 °C. After three washes with 0.1% TBS-Tween20, membranes were incubated with an HRP-conjugated secondary anti-mouse antibody (RRID: AB_330924, 1:5000) in 5% non-fat milk in 0.1% TBS-Tween20 for 1 h at room temperature. The blot was then washed three times with 0.1% TBS-Tween20 before signal detection using a chemiluminescence system (ChemiDoc MP, Biorad). Data was normalized to total protein per membrane spot analyzed.

### Fractionation protocol for the enrichment of detergent-insoluble protein aggregates

Detergent-insoluble protein aggregates were enriched using a sarkosyl-based fractionation protocol. Samples were homogenized in ice-cold low salt (LS) buffer (50 mM HEPES pH 7.0, 250 mM sucrose, 1 mM EDTA) with 1X protease and phosphatase inhibitors on ice. Sarkosyl (1% w/v) and NaCl (0.5 M) were added, followed by incubation on ice for 15 min and sonication with Diagenode Bioruptor Pica (setting: 5 s ON, 30 s OFF, 5 cycles, medium frequency). Protein concentrations were determined using the BCA assay, and homogenates were ultracentrifuged at 180,000 × g for 30 min at 4 °C. Sarkosyl-soluble supernatants were collected, while insoluble pellets were washed with sark-buffer and solubilized in urea buffer (50 mM Tris-HCl pH 8.5, 8 M urea, 2% SDS) for 30 minutes at room temperature, followed by low-amplitude sonication (setting: 5 s ON, 30 s OFF, 3 cycles, low frequency). Protein concentrations were normalized based on BCA assay results, and samples were adjusted using sark-buffer before mixing with 5X Laemmli buffer and heating at 95 °C for 5 min.

For Western blot analysis of the samples above, proteins were resolved via SDS-PAGE and transferred onto a membrane using wet transfer. Membranes were blocked with 5% BSA in TBS for 1 h at room temperature, then incubated overnight at 4 °C with primary antibodies (mouse anti-a-syn 2A7, RRID: AB_1555287 and rabbit anti-β-actin, RRID: AB_2305186, both 1:1000). Following three washes in 0.1% TBS-Tween20, membranes were incubated with HRP-conjugated secondary antibodies (anti-mouse, RRID: AB_330924, and anti-rabbit, RRID: AB_2099233, both 1:5000) for 1 h at room temperature. After final washes, protein bands were detected via chemiluminescence (iBright system, Thermo Fisher). Data were normalized to β-actin.

### Human Cytokine array

The supernatant of iPSC-ENLs stimulated for 24 h with 100 ng/mL TNF was collected. We added 1 mL of the supernatant plus 500 μL of array buffer 5 for each membrane of the Proteome Profiler Human Cytokine Array Kit (R&D systems, Cat#ARY005B), which detects up to 36 cytokines at the same time. Each membrane was used for the supernatant coming from one biological sample. Analysis was performed according to the manufacturer's instructions.

### Western blotting

Western blotting was performed according to standard protocols. 4–12% NuPAGE Bis-Tris- gels (Thermo Fischer Scientific) were used to separate proteins by SDS-PAGE. 30 μg of protein was loaded per condition. After transfer to PVDF, the total protein was detected using (No-Stain™ Protein Labeling Reagent (A44449)) using manufacturer's protocol. The membrane was incubated in blocking buffer (4% milk in TBS-T (24.8 mM Tris, pH 7.4, 137 mM NaCl, 2.7 mM KCl, 0.05% (v/v) Tween 20)) for 1 h at room temperature and with primary antibody (anti rabbit Aspartate Aminotransferase Antibody, RRID:AB_3598484; diluted 1:1000 in blocking buffer) overnight at 4 °C. Incubation with anti-rabbit AlexaFluor 750 secondary antibody (diluted 1:5,000 in blocking buffer) for 1 h at room temperature was followed by three washing steps with TBS-T. Proteins were detected using Invitrogen iBright FL1500 Imaging System and analyzed by iBright Analysis (iBA) software (Invitrogen).

### Multielectrode array recording

To measure neuronal activity, iPSC-ENLs were seeded onto a 48-well CytoView MEA plate (Axion Biosystems) pre-coated with geltrex on day 15 of differentiation. Cells were maintained in ENC medium until day 70 (end of differentiation). Recordings were performed for 10 min at 37 °C in an environment of 95% O$_2$ and 5% CO$_2$ using the Maestro MEA System and AxIS software (Axion Biosystems). Signals were sampled at 12.5 kHz with a hardware frequency bandwidth of 200–5000 Hz and later processed using a first-order Butterworth band-pass filter (200–5000 Hz). Spike detection was set at an adaptive threshold of 5.5 standard deviations above background per electrode, with active electrodes defined as those detecting ≥5 spikes per minute. Activity parameters were analyzed using the AxIS Metric Plotting Tool (Axion Biosystems).

### Proteomics analysis

Proteomics data were analyzed as previously published, with some modifications[81]. The cells were lysed with lysis buffer (8 M urea, 0.1 M NH$_4$HCO$_3$) and the concentration of total proteins was quantified using the BCA kit (ThermoFischer). Samples were treated with 12 mM DTT for 30 min at room temperature followed by alkylation with 40 mM Iodoacetamide in the dark for 45 min. After dilution of the sample to 1:5 with 0.1 M NH$_4$HCO$_3$ an overnight digestion with 1 μg Trypsin (Promega) was performed. The tryptic digest was acidified to pH<3 using TFA and desalted using C18 reversed phase spin columns (Harvard Apparatus) according to the manufacturer's protocol. Dried peptides were resolved in 0.1% formic acid containing 1% acetonitrile prior to injection into the mass-spectrometer.

The peptide samples were analysed on an Orbitrap Exploris480 mass spectrometer (Thermo Scientific) equipped with a Waters M-class UPLC system (Waters AG) operating in trap/elute mode. Peptides were loaded and chromatographically separated using a Symmetry C18 trap column (5 μm, 180 μm X 20 mm, Waters AG) and as separation column an HSS T3 C18 reverse-phase column (1.8 μm, 75 μm X 250 mm, Waters AG). The columns were equilibrated with 99% solvent A (0.1% formic acid (FA) in water) and 1% solvent B (0.1% FA in ACN). Trapping of peptides was performed at 15 μl/min for 30 sec and afterwards the peptides were eluted using the following gradient: 1-40% B in 120 min; 40-98% B in 5 min. The flow rate was constant 0.3 μl/min and the temperature was controlled at 50 °C. High accuracy mass spectra were acquired with a Orbitrap Exploris480 operated in data independent acquisition mode. A survey scan was followed by up to 12 MS2 scans. The survey scan was recorded using quadrupole transmission in the mass range of 350-1500 m/z with an AGC target of 3E6, a resolution of 120,000 at 200 m/z and a maximum injection time of

50 ms. All fragment mass spectra were recorded with a resolution of 30,000 at 200 m/z, an AGC target value of 1E5 and a maximum injection time of 50 ms. The normalized collision energy was set to 28%. Dynamic exclusion was activated and set to 30 s. For protein identification and label-free quantification the obtained spectra were analyzed using the software FragPipe. This software was used to automatically generate a spectral library and with this the. Raw files were uploaded as input to calculate protein levels.

Differential expression analysis was performed using the limma package, with linear models and contrasts applied to compare Iso and SNCA 3x groups under various treatment conditions. Differentially expressed proteins (DEPs) were identified by filtering for $p < 0.05$ and log2 fold changes $\geq 0.8$ or $\leq -0.8$. Data visualization included volcano plots to represent significance and fold change, and Venn diagrams to identify overlapping DEPs across conditions. Gene Ontology (GO) enrichment analysis was performed on DEPs using the clusterProfiler package, focusing on the Molecular Function category, and the results were visualized in bar plots. Final results, including protein identifiers, fold changes, p-values, and GO enrichment, were compiled for downstream analysis. Ingenuity pathway analysis was used to merge the top protein-protein interaction networks found when comparing SNCA 3x vs Iso stimulated with TNF vs basal level.

## Metabolomics analysis
Untargeted metabolomics were in general performed as previously described with minor modifications[82]. Cells were lysed in an 80:20 methanol-water mixture (1 mL per sample) containing recovery standards, followed by centrifugation at 24,000 g and 4 °C for 5 min. From the supernatant 350 μL each were used for HILIC and RP chromatography and 230 μL for pooled QC samples, which were prepared the same way as unknown samples after pooling. All supernatants were dried under nitrogen at 30 °C before reconstitution in HILIC or RP eluent. LC-MS analysis was performed using a Dionex Ultimate 3000 system coupled to a Q Exactive Focus mass spectrometer (Thermo Fisher Scientific) with HILIC (Acquity UPLC BEH Amide, 1.7 μm) and RP (Acquity UPLC BEH C18, 1.7 μm) chromatography. The system was operated in positive and negative ionization modes with a mass detection range from 66.7 to 1000 m/z in full-scan mode, with the three most abundant features fragmented in ddMS2 mode. QC samples and blanks were included for system calibration and batch correction. Data were analyzed using Compound Discoverer 3.3, with features identified using a 5 ppm mass tolerance and 0.2 min RT tolerance. Peak areas were normalized to protein concentration, Compound annotations were assigned via ChemSpider, mzCloud™, mzVault, and in-house databases. Only compounds with annotation levels 1 or 2 (See Supplementary Data 10) were included. Analysis of individual metabolites was calculated by performing a scaling of the data to consider the donor effect, followed by Student's unpaired two-tailed t-tests, similar to previous publications[83]. Metabolite-set enrichment analysis was performed using MetaboAnalyst[84], considering the DEMs (raw p-values < 0.05, log2 fold changes $\geq 0.5$ or $\leq -0.5$)

## Integrated pathway analysis
Integrated pathway analysis was generated using MetaboAnalyst[84] by adding as an input the DEPs (raw $p < 0.05$ and log2 fold changes $\geq 0.8$ or $\leq -0.8$) and DEMs (raw $p < 0.05$ and log2 fold changes $\geq 0.5$ or $\leq -0.5$) obtained when comparing SNCA 3x vs Iso stimulated with TNF.

## Proximity ligation assay (PLA)
Imaging 24-well cell plates containing iPSC-ENLs at day 70 (basal and TNF-stimulated) were fixed with 4% paraformaldehyde and permeabilized with 0.4% (v/v) Triton X-100. The Duolink In Situ PLA kit (Sigma Aldrich, Cat#DUO92101) was used for PLA. Cells were blocked and incubated with mouse anti-α-syn (RRID: AB_1555287, 1:50) or rabbit

anti-TOM20 (1:250, RRID: AB_2207530) antibodies and thereafter with the corresponding PLA probes. After ligation and amplification using the kit reagents, cells were counterstained for α-syn and TOM20 and mounted using Duolink mounting medium with DAPI (1:2500). A pipeline to analyze the PLA dots in the cytoplasm of the cells was created using CellProfiler (Broad Institute) and at least 4–5 individual images per individual biological sample under each condition (basal or TNF) were analyzed. Imaging was performed using the Super Resolution Evident SR Spinning Disk microscope (Olympus) and the cellSens Software and processing was performed using FIJI (ImageJ).

## Mitochondrial parameter analysis
Mitochondrial morphology was assessed using the TOM20 immunofluorescence stainings and analyzed in ImageJ (Fiji) with the Mitochondria Analyzer plugin[85]. Images were acquired as TIFF files, and the TOM20 channel was processed for analysis. Briefly, images were converted to 8-bit grayscale, and background was subtracted using a rolling ball algorithm (radius = 50 pixels). A global threshold (Otsu method) was applied to segment mitochondria, followed by binary processing to refine structures. Nuclei were identified from the DAPI channel using CellProfiler, and mitochondrial parameters were normalized to the number of nuclei per image. A minimum of three images per biological replicate (individual Iso or SNCA 3x line) was analyzed, and results were averaged per biological replicate. The total number of cells analyzed per cell line and used to normalize the data was 1276 (Iso-1), 1534 (SNCA 3x-1), 810 (Iso-2), 1511 (SNCA 3x-2), 614 (Iso-3) and 1083 (SNCA 3x-3). Statistical analyses were performed in R, with group comparisons conducted using Student's unpaired two-tailed t-tests.

## Seahorse analyses
For Seahorse assays, we seeded 60,000 ENL precursors per $cm^2$ on Geltrex (Thermo Fisher)-coated Seahorse XF24 cell plates (Agilent, USA) at day 15 of the differentiation and kept them with ENC medium until day 70, when the analyses were performed. For mitostress assays, cells were washed two times and incubated for 1 h at 37 °C in an incubator without $CO_2$, with respiratory medium consisting of DMEM XF (Agilent) supplemented with 10 mM glucose solution (Agilent) plus 1 mM pyruvate solution (Agilent) and 2 mM glutamine (Agilent). After equipment calibration, baseline respiration measurements were followed by injection of 5 μM oligomycin (Sigma), 5 μM carbonyl cyanide m-chlorophenylhydrazone (CCCP) (Sigma) and 5 μM rotenone (Sigma) plus 5 μM antimycin A (Sigma). To perform the mitofuel assays, we used the same medium and conditions of the mitostress assay, but the compounds were the ones provided by the Seahorse XF Mito Fuel Flex Test Kit (Agilent), in the concentrations indicated for the metabolic dependency analysis, which included 3 μM BPTES, 4 μM etomoxir and 2 μM UK5099 (final concentrations). All respiratory modulators were previously titrated, and all plates were normalized to protein content using the BCA kit (Thermo). Analysis was performed using the Wave software (Agilent).

## Bioinformatics analysis of public and in-house datasets
To investigate MAS suppression in the context of human intestinal inflammation, we analyzed three publicly available transcriptomic datasets from ulcerative colitis (UC) patients treated with infliximab (GSE16879, GSE12251, and GSE73661). For GSE73661, $n = 12$ controls, $n = 15$ UC non-responders and $n = 8$ UC responders. For GSE16879, $n = 6$ controls, $n = 16$ UC non-responders and $n = 8$ UC responders. For GSE12251, $n = 11$ UC non-responders and $n = 12$ UC responders. Processed expression matrices were downloaded from the GEO database, and gene expression of GOT1 and SNCA was compared between responders and non-responders. Values were normalized to a log2 scale.

For validation, we analyzed our in-house bulk RNA-seq dataset (IBDome[53] - https://ibdome.org/), which includes inflamed and non-inflamed colonic biopsies from UC patients. The p-values have been calculated using a simple Wilcoxon-Mann-Whitney test. Correlation between *SNCA* and *GOT1* expression in inflamed versus non-inflamed tissue was assessed using Pearson correlation.

To evaluate MAS gene expression at the single-cell level, we accessed the integrated scIBD database (http://scibd.cn/) which compiles single-cell transcriptomic data from 12 IBD studies. We selected the following parameters: Neural (major cluster), large intestine (tissue), adult (developmental stage), then selected all studies, colon and rectum (location), the three groups of interest (healthy, UC non inflamed and UC inflamed) and 5000 downsampled cells per subtype. Average expression of *GOT1*, *GOT2*, *MDH1*, and *MDH2* was compared between inflamed and non-inflamed conditions.

## Statistics and reproducibility

Unless stated otherwise, all quantitative analysis was performed using independently collected material from three isogenic and three SNCA 3x cell lines (our biological replicates, n = 3 per genotype). The number of independent differentiations runs (technical replicates) used to generate the data for each panel is specified in the corresponding figure legend. Each data point in the figures represents the average of these technical replicates for each individual line. Statistical analysis was performed using Graphpad Prism 10, by either two-tailed unpaired Student's *t* test (two groups), one-way ANOVA followed by Tukey's post-hoc test or two-way ANOVA followed by Sidak's, Tukey's or Bonferroni's post hoc test (more than two groups). Error bars represent standard error of the mean (SEM). Statistical analysis for scRNAseq, metabolomics and proteomics analysis were performed by different packages in R as stated. $p < 0.05$ was considered significant. Details of statistics and sample sizes are described in the figure legends. No statistical method was used to predetermine sample size. No data were excluded from the analyses. The experiments were not randomized. The investigators were not blinded to allocation during experiments and outcome assessment.

## Reporting summary

Further information on research design is available in the Nature Portfolio Reporting Summary linked to this article.

## Data availability

The scRNAseq data generated in this study have been deposited in the GEO database under accession code GSE301050. The proteomics data generated in this study have been deposited in the ProteomeXChange database under accession code PXD075048 and in the MassIVE database under accession code MSV000101003 [https://massive.ucsd.edu/ProteoSAFe/dataset.jsp?task=b41f61176673413694a7cf65448de9fe]. The metabolomics data generated in this study have been deposited in the MassIVE database under accession code MSV000098366 [https://massive.ucsd.edu/ProteoSAFe/dataset.jsp?task=85492deedc7c40a39a3568287f5d398b]. Source data are provided with this paper.

## Code availability

All codes used in this publication, together with the Seurat object used to generate all our scRNAseq data are available at [Ghirotto, Bruno (2025), "ENS alpha synuclein paper 2025", Mendeley Data, V2, [https://data.mendeley.com/datasets/v8dknj466y/2].

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

## Acknowledgements

*Funded by the Deutsche Forschungsgemeinschaft (DFG, German Research Foundation) – 505539112 – KFO 5024 (A01, A02, A04, B02, Z01).* Further support came from the Bavarian Ministry of Science and the Arts in the framework of the ForInter network. The LC-MS system used for metabolomic analysis was funded by the Deutsche Forschungsgemeinschaft (DFG, German Research Foundation) – INST 90/1048-1 FUGG. A.H. and D.B.B. were supported by the German Federal Ministry of Research, Technology and Space (BMFTR) – 031L0309A. M.S. and F.J.T.: Funded by the European Union (ERC, DeepCell – 101054957). Imaging was performed at the Optical Imaging Compentence Center Erlangen (OICE) using a DFG-funded microscope system (project number 522417173). P.G. was supported by the AI-PREDICT project (grant no. 01ZU2502) funded by the BMFTR, as well as by Deutsche Forschungsgemeinschaft (DFG, German Research Foundation) through TRR 417 (Project-ID 540805631, Project S03) and SFB 1755 "CASCAID" (Project-ID 550296805, Project Z01). We thank the OICE staff, mainly Dr. Philipp Tripal and Dr. Benjamin Schmid for their technical support. We thank Daniel Beß for technical assistance with the tissue paraffin sections.

## Author contributions

B.G. designed the research, performed experiments, analyzed and interpreted the data, executed bioinformatics analyses, created the figures, and wrote the manuscript. L.E.G., V.R., C.J., E.G, A.G., J.G., N.C.P. performed experiments and analyzed the data. H.W. and M.F. maintained iPSC cultures. M.K., M.S. and A.H. performed bioinformatic analysis on the scRNAseq data. F.Z. generated the iPSC lines used in this study in the laboratory of J. R. Mazzulli, Northwestern University, Chicago. T.R., I.P., J.W., M.V., D.B.B., C.G., F.J.T., R.R. and M.N. provided resources and supervised the project. P.G. wrote the manuscript, analyzed and interpreted data and supervised the bioinformatics analyses. B.W. designed the research, provided resources, interpreted the data, wrote the manuscript, acquired funding and supervised the project. All authors critically reviewed and approved the final version of the manuscript.

## Funding

## Competing interests

F.J.T. consults for Immunai Inc., CytoReason Ltd, Cellarity, BioTuring Inc., and Genbio.AI Inc., and has an ownership interest in Dermagnostix GmbH and Cellarity. The remaining authors declare no competing interests.

## Additional information

[1]Department of Stem Cell Biology, University Hospital Erlangen, Friedrich-Alexander-Universität Erlangen-Nürnberg, Erlangen, Germany. [2]International Max Planck Research School in Physics and Medicine, Erlangen, Germany. [3]Department of Medicine 1, University Hospital Erlangen, Friedrich-Alexander-Universität Erlangen-Nürnberg, Erlangen, Germany. [4]Dental Clinic 1-Department of Operative Dentistry and Periodontology, University Hospital Erlangen, Friedrich-Alexander-Universität Erlangen-Nürnberg, Erlangen, Germany. [5]Institute of Molecular Physical Sciences, ETH Zürich, Zürich, Switzerland. [6]Institute of Computational Biology, Helmholtz Center, Munich, Germany. [7]TUM, School of Computation, Information and Technology, Technical University of Munich, Munich, Germany. [8]TUM School of Life Sciences, Technical University of Munich, Munich, Germany. [9]Biomedical Network Science Lab, Department Artificial Intelligence in Biomedical Engineering (AIBE), Friedrich-Alexander-Universität Erlangen-Nürnberg, Erlangen, Germany. [10]Institute of Experimental and Clinical Pharmacology and Toxicology, Friedrich-Alexander-Universität Erlangen-Nürnberg, Erlangen, Germany. [11]Institute of Pathology, Friedrich-Alexander-Universität Erlangen-Nürnberg, Klinikum Bayreuth, Bayreuth, Germany. [12]Department of Molecular Neurology, University Hospital Erlangen, Friedrich-Alexander-Universität Erlangen-Nürnberg, Erlangen, Germany. [13]Deutsches Zentrum Immuntherapie (DZI), University Hospital Erlangen, Friedrich-Alexander-Universität Erlangen-Nürnberg, Erlangen, Germany. [14]Center of Rare Diseases Erlangen (ZSEER), University Hospital Erlangen, Friedrich-Alexander-Universität Erlangen-Nürnberg, Erlangen, Germany. [15]Present address: Department of Molecular, Cellular, and Developmental Biology, University of Colorado, Boulder, CO, USA. [16]These authors contributed equally: Pooja Gupta, Beate Winner. ✉e-mail: beate.winner@fau.de

