## [Transparent Peer Review file · Nature Communications]

TNF alpha unmasks enteric malate aspartate shuttle dysfunction bridging Parkinson disease and intestinal inflammation

Corresponding Author: Professor Beate Winner

Version 0:

Reviewer comments:

Reviewer #1

(Remarks to the Author)

Ghirotto et al aims to examine how a-synuclein (a-syn) affects the enteric nervous system (ENS) in Parkinson's disease (PD) and the role that inflammation plays in this process. Given the focus on the ENS as a potential originating site of PD a-syn pathology this is an important question to examine. As model system they utilise three induced pluripotent stem cell (iPSC) lines from patients with SNCA triplication and their previously generated isogenic controls, which they differentiate to enteric neural lineage. To my knowledge, this is the first attempt to model PD pathology in iPSC-derived enteric neural cells.

The authors perform a thorough multi-omic analysis including scRNAseq, proteomics and metabolomics as well as multiple functional assays. They confirm the validity of the ENL cultures using scRNA comparison to a human gut atlas. Overall, this is exciting and well-performed work.

However, some of their conclusions and interpretations of the data are not fully supported. This is mainly related to their interpretation of the effect of TNF-a exposure on SNCA 3x cultures where in several assays there is not actually a significant difference between untreated and TNF-a treated SNCA 3x ENLs and/or the isogenic control ENLs response to TNF-a is identical to the SNCA 3x ENLs.

Major concerns/questions:

1. The authors show that the number of enteric neurons is significantly decreased in SNCA 3x ENL cultures at d70 based on scRNAseq whereas numbers of glia and proliferating progenitors are similar. They do not address how this difference in the number of enteric neurons arises between SNCA 3x and isogenic controls. Are fewer neurons generated or have they died during the differentiation?

An important question arising from this, is how the different cellular composition of the ENLs cultures between SNCA 3x and isogenic controls affect the downstream analyses? E.g. in Fig. 3 A-B where the mitochondrial content and -morphology is analysed and found decreased in SNCA 3x. Is this a result of the difference in cell type composition/morphology as mitochondrial morphology and content can differ between neurons and glia? The difference in neuronal numbers is also important to keep in mind in the interpretation of the MEA data.

2. Of the three overall subpopulations of cells, only glia cells show significantly increased SNCA expression based on scRNAseq. This is interesting in relation to the finding of decreased numbers of neurons in SNCA 3x cultures. The authors show immunofluorescent stainings for GFAP, HuC/D and a-syn I Fig. 1C. Can they based on these stainings determine if the cell type specific SNCA expression differences translate to the protein level?

3. The authors conclude that the scRNAseq differential analysis reveal mitochondrial dysfunction in SNCA 3x ENLs, but do not refer to any figure to support this (line 161-162). The enrichment analyses in Fig. 2D+F do not include mitochondrial related terms besides from "regulation of lipid metabolic process" and "integration of energy metabolism". Similarly, the signalling pathways identified as altered by the CellChat analysis as presented in Fig. S1H and Fig. S3E+H do not appear particularly related to mitochondria although this is mentioned in the results section? EGF and NOTCH signalling have many other more prominent functions and the majority of the other factors listed appear more related to axogenesis, neurogenesis and extracellular matrix etc.

4. The proteomics data is reported with p-values and there is no mention of correction for multiple testing. Similarly for the metabolomics data only p-values are reported. In large scale analyses such as these, correction for multiple testing should be applied and q-values/adjusted p-values reported.

5. The authors conclude from the proteomic and metabolomic analyses that MAS and TCA cycle is impaired by TNF- α in SNCA 3x. Besides from AATC are any other MAS or TCA cycle proteins changing with TNF- α exposure? In the analysis of patient data in Fig. 8 additional MAS enzyme expression levels are examined. Are any of these altered in SNCA 3x ENLs? All the metabolites shown in Fig. 5E are lower in SNCA 3x than isogenic control. Could this be a consequence of the generally lower mitochondrial content identified in Fig. 3A-B?

6. The statistical analysis of a-syn-mito co-localisation with PLA in Fig. 6C should be redone to show the average for each of the cell lines – as all other figures in the paper – instead of showing data for individual cells. The figure legends states $n = 3$, but the statistical analysis appears to be performed on individual cells. It is unclear whether TNF- α significantly increases a-syn-mito PLA in SNCA 3x if analysed in this way? However, it appears that TNF- α increases a-syn-mito PLA significantly for isogenic control, which is currently not mentioned/discussed?

Also, could the authors please discuss the discrepancy between the results from Fig. 6C to Fig. 6E where there is no increase of a-syn in the mitochondrial fraction with TNF- α compared to untreated for either SNCA 3x or isogenic control? The western blot in figure 6D should show the starting material / un-isolated samples for comparison to demonstrate that successful isolation of mitochondria has been achieved.

Why are basal conditions without TNF- α not included in Fig. 6F when the earlier data has clearly shown that baseline mitochondrial dysfunction is present in SNCA 3x ENLs? This makes it unclear whether TNF- α is having an effect or the difference is caused by SNCA 3x alone? Please include vehicle treated as well.

Overall, the headline “TNF- α increases a-syn-mitochondria interactions and induces oxidative stress in SNCA 3x ENLs” is currently not fully supported by the presented data.

7. Fig. 7B-C: the authors state that TNF- α -treated SNCA 3x ENLs showed reduced mitochondrial respiration including ATP-linked OCR. However, the effects are close to identical for the TNF- α -treated isogenic control ENLs. This should be reflected in the interpretation of the data. Overall, the effects of CSB6 appear overall similar independent of genotype. Currently, this is only really mentioned in the limitations of the study section. Only the glutamine dependency seems to really differ between SNCA 3x and isogenic control, independently of TNF- α exposure, with normalisation after CSB6 treatment for SNCA 3x. The statement that CSB6 promoted a shift toward fatty acid oxidation is confusing. FAO levels are unchanged for SNCA 3x between the basal, TNF- α - and TNF- α +CSB6-treated conditions – and significantly decreased in isogenic control after CSB6 treatment?

For the Mitotracker green assay the authors note a trend towards an increase in dysfunctional mitochondria in SNCA 3x with TNF- α . However, the trend towards decreased dysfunctional mitochondria at baseline are not mentioned. CSB6 decreases the percentage dysfunctional mitochondria to close to 0 independently of genotype. However, the authors only mention the effect for SNCA ENLs as the result is apparently only significant for this group. The average for TNF- α treated isogenic control is around 20% and with CSB6 around 1 with no visible arrow bars.

8. The authors write in the abstract that they show a-syn upregulation as a general hallmark of intestinal inflammation in human gut tissue. They mention examining SNCA expression data – together with GOT1 expression – in intestinal inflammation patient samples. However, the SNCA expression is only shown in correlation to GOT1 expression and no formal comparison of a-syn levels between inflamed and uninfamed tissue is shown. Could the authors please show the SNCA expression levels the same way they show GOT1 levels in Fig. 8B? Similarly, the SNCA expression data from inflamed and non-inflamed gut tissue from the IBDome dataset should be presented and not only correlations with GOT1. The immunofluorescent staining for GOT1/AATC and a-syn on colon biopsies should be performed on more than one patient and one control if anything is to be concluded from it.

Minor concerns:

1. Fig. 2A+E Can the authors please provide legends with naming of the different subclusters of cells in the scRNAseq data? It is difficult to interpret the data when we only have subcluster numbers.
2. Is there a correlation between the subclusters of cells with the highest changes in DEGs and their SNCA expression? E.g. for neuronal subcluster 8 which is reduced in abundance in SNCA 3x ENLs the SNCA expression levels would be interesting to know.
3. If a-syn has been identified by the proteomic analysis, how are the levels in the different conditions?
4. Could the authors also please report how many proteins and metabolites were identified for the reader to understand how large a proportion of the identified proteins and metabolites differ between the different condition.
5. Can Fig. 5H please be enlarged so that it is more readable.
6. Please define in the results section (line 344) that GOT1 = AATC.

(Remarks on code availability)

Reviewer #2

(Remarks to the Author)

This manuscript presents an ambitious and integrative study using human iPSC-derived enteric neural lineages (ENLs) from SNCA triplication Parkinson's disease (PD) patients to explore the interplay between alpha-synuclein accumulation,

inflammation, and metabolic dysfunction in the ENS. The authors employ a comprehensive multi-omic approach (scRNA-seq, proteomics, metabolomics), functional assays (electrophysiology, Seahorse), and validation in human UC samples to support a model in which TNF- α exposure unmasks mitochondrial and synaptic vulnerability in SNCA 3x ENLs through disruption of the malate-aspartate shuttle (MAS). The study is highly relevant, well-executed, and presents a novel mechanism with broad implications. However, several aspects of the manuscript require clarification or additional analysis to strengthen the conclusions and ensure appropriate interpretation.

1. Several results suggest that SNCA expression is not significantly different at baseline in neurons, despite conclusions about SNCA-driven effects. It would help to show quantifications and expression levels across neuronal and glial subclusters and assess SNCA⁺ cells specifically. Consider module scoring across SNCA⁺ neurons and glia, or correlation with SNCA-interacting genes.
2. Figures 3–7 present pooled ENL results despite many cell-type-specific conclusions. Where feasible, separate neuron vs glia in imaging, flow, and Seahorse analyses or explicitly qualify conclusions that cannot be deconvolved.
3. The cell types and annotations in the scRNA-seq datasets should be described in more detail. What markers were used to define each cluster? This information is critical for evaluating the biological interpretation of the single-cell results.
4. In several figures (e.g., Figure 1E), tests such as Student's t-test are applied to proportions or counts; chi-squared or other appropriate methods should be used. Additionally, comparisons (e.g., Figure 4C) currently conflate genotype and baseline differences. Comparisons should isolate the effect of TNF within each genotype.
5. The strength of the TNF-specific effect in SNCA 3x ENLs is not always evident. Some changes may be shared with ISO. Figure 5 should include the full condition matrix (ISO \pm TNF, SNCA 3x \pm TNF) with clearer annotations.
6. The <5% MT gene threshold may bias against metabolically stressed neurons. Suggest re-analyzing with a relaxed cutoff (e.g., 15–20%) and assessing whether stressed populations or DEGs change.
7. The manuscript suggests enteric glia compensate for neuronal mitochondrial stress, but supporting data are limited. Consider reframing as a hypothesis or provide additional evidence (e.g., selective pathway upregulation, functional protection).
8. It would be great to include CSB6-only controls in functional assays to distinguish TNF-specific rescue from general effects.
9. The proteomic/metabolomic findings should be validated by IF or flow cytometry where possible.
10. Does infliximab treatment alter SNCA-GOT1 correlation in UC datasets?
11. The quantification of GOT1 and SNCA IF in human tissue should be included and the biopsy metadata should be clarified.

Minor Comments

- Figure S2 heatmaps are difficult to interpret due to inconsistent cluster ordering. Reordering or consistent labeling would help.
- Clarify gene/cluster selection in Figures 2B–D, 3C–D.
- In Figure 4A, define 'total cells' and clarify why neuron/glia are pooled.
- Improve figure labeling for Figures 6C, 7E–F; comparisons are unclear.
- In Figure 8, clarify the comparison in panels B–C; what are we meant to conclude?
- Describe IBDome and sclBD datasets in more detail: sample size, condition, analysis methods.

(Remarks on code availability)

N/A

Reviewer #3

(Remarks to the Author)

The manuscript investigates an important understudied area of the pathology of enteric nerves in Parkinson's. The authors describe the iPSC derived model and the cells generated from it in a good level of detail in the isogenic and SNCA triplication lines. Overall the data are extremely interesting and add to the field in terms of inflammation and alpha synuclein in gut neurons. Some specific comments are outlined below:

In Figure 2 the authors describe changes in specific cellular subclusters which link mitochondrial stress, lipid homeostasis, cholesterol metabolism and stress responses. Whilst the authors link many of these together, it would be interesting for the authors to comment on the possible link between the changes in cholesterol metabolism and mitochondrial stress. Although a small part of the cholesterol pathway, mitochondria do play a significant role which is extremely important for bile acid synthesis, also an area of interest in enteric neurons in PD. Hence, could the authors comment on their scRNAseq data in the sub clusters from this point of view?

For the data presented in Figure 3 specifically (but also to other figures throughout) could the authors be clearer in how many cells were quantified and from how many differentiations of each line? At the moment the figure legend is ambiguous with at least 3 images per line, making 3 biological repeats. To ensure the robustness of the data, it needs to be clear from how many differentiations the data was generated.

In Figure 3 the authors postulate that metabolism is shifted in the SNCA triplication lines, and that this may be compensatory. Could the authors expand on why this change would be compensatory response to stress as this is not obvious from the pseudo time experiments shown in Figure 3 without further explanation for the reader.

The authors data showing the changes in proteome after TNF α stimulation are compelling however in the manuscript the authors refer to the SNCA neurons being sensitised to inflammatory stimuli, the data from Figure 3 would suggest this is not general priming but it could be specific, could the authors comment or alter the text.

Throughout the manuscript the figure legends are not detailed enough, only in the Seahorse figure legend of Figure 7 do the authors state that the mean of 3 independent experiments is shown. Could the authors please confirm this for all

experiments.

The data generated are compelling for the change in metabolic status after TNFalpha stimulation. The data in Figures 6 and 7 aim to show a functional effect of this change. However, many of the measurements are of mitochondrial function of OXPHOS rather than of the TCA cycle or beta oxidation. Although the fuel flex experiments show a change, in order to support the data in the previous figures and the conclusion of the his change and critical alterations in the malate shuttle, it would be beneficial if the authors were to functionally measure TCA cycle and beta oxidation activity rather than depend on secondary measurements. Furthermore, the first sets of figures suggest SNCA triplication itself causing mitochondrial abnormalities, however, the functional data in Figure 7 in particular do not support this, could the authors comment. For the CSB6 experiments do the authors show target engagement and therefore can be certain that the CSB6 treatment is having this effect via the known mechanism of action rather than off target effects, as many compounds can affect mitochondria and metabolism.

(Remarks on code availability)

Reviewer #4

(Remarks to the Author)

Ghirotto and colleagues present an iPSC-derived Enteric Neuron System of the alpha-synuclein triplication as a means to investigate the pathogenesis underlying the gut-to-brain hypothesis of PD and potential other synucleinopathies. Their work identifies that soluble TNF-alpha treatment exacerbates mitochondrial dysfunction in enteric neurons. While mitochondrial dysfunction in synucleinopathies is well established, there have been no reports studying this phenomenon in the enteric nervous system.

I have the following comments:

1. Given the observed cellular heterogeneity in the culture (scRNAseq), the cell types could be further refined. a) More fine-level cell typing exists within the Drokhlyansky et al. study. The authors should adequately cell type their cells to the finer levels shown in the manuscript. b) The authors should show the marker genes of the ENs in scRNAseq clusters, such as the markers that they used for their qPCRs (e.g., HOXB3 and TUBB3 are missing from Figure S2A). c) The authors should stain their cells according to the HOXB3, PHOX2B, and TUBB3 to show the percentage of cells expressing those proteins in both cell lines.
2. According to Fig. 1E, SNCA 3x cultures have less enteric neuron proportions. a) Is this something reproducible? b) Is this something technical, or do neuronal populations drop out from the SNCA 3x cultures? In any case, the authors should address how this heterogeneity drives the rest of the conclusions that rely on these bulk assays of these populations. For example, in Figure 5D, is this analysis valid if there are fewer neurons in the SNCA 3x samples?
3. Does treatment of TNF1a induce cell death in the SNCA 3x and/or Iso?
4. It would be interesting to compare the phenotypes of the SNCA 3x neuronal dysfunction with existing studies that investigate similar questions of the SNCA 3x lines but in other neuronal types: (Patikas et al. Neuroscience Research 2023; Jin et al. Science Advances 2024). Comparing the phenotypes of midbrain and cortical organoids could highlight common modules.
5. Since the whole manuscript builds upon the notion that asyn pathology spreads from the gut to the brain, the authors should consider testing asyn fibrils and tnf1a to check whether this pathology can be recapitulated in the Iso cell lines.
6. Cell-to-Cell communication. The way the data is presented, neurons and glia appear to be two separate CellChat analyses. Does this assume that the neurons do not communicate with glial cells? If so, why?
7. L465: "iPSC-ENLs makes it challenging to separate mechanistic effects in a populational level". What does populational level mean?
8. Where is the reported pathology owed? SNCA 3x has fewer neurons at Day 70, so does this signature come from the neurons or the glia populations?
9. L304: Does asyn interact with TOM20 in Iso cell lines?
10. L320: Fig 6C appears to have incorrect statistics; Poisson events (data should be transformed as $\log(\text{counts} + 1)$) should be used prior to statistical testing. A similar problem might be present in the Fig. 1E.
11. L218: "This suggests that SNCA 3x enteric glial cells compensate for neuronal stress by enhancing mitochondrial support.". Not sure how this is supported by the data.
12. Figure S2A: HOXB5 is shown twice.

(Remarks on code availability)

Version 1:

Reviewer comments:

Reviewer #1

(Remarks to the Author)

The authors have fully addressed my previous concerns and the conclusions of the paper have been strengthened by the additional data and analyses.

(Remarks on code availability)

Reviewer #2

(Remarks to the Author)

The authors have adequately addressed all my concerns and comments

(Remarks on code availability)

Reviewer #3

(Remarks to the Author)

The authors have thoroughly addressed the reviewers comments, taking into consideration the comments, adding additional experimental work, detail, changing interpretation of results and overall enhancing the manuscript.

(Remarks on code availability)

Reviewer #4

(Remarks to the Author)

In their rebuttal, the authors have performed additional analyses that substantially address most comments and have improved the quality of their manuscript, thereby strengthening the impact of their findings.

A major concern remains that the scRNAseq data show significant shifts in their neuronal population, which the authors deemed non-significant, citing a log₂ fold change threshold of 1. This threshold is quite stringent for population shifts and, in my opinion, arbitrary. The authors should assess the ramifications of an approximately 30% reduction in neurons (log₂ fold change around -0.5, as shown in the figure for the neurons) observed in their Figure 1E in the rest of the analysis. My impression of a 30% differential abundance is substantial, and in a different research context, it would have been "neuroprotective". While variability between differentiated iPSC lines is a well-known problem (Jerber et al., 2021, Nature Genetics) and is well beyond the scope of this manuscript, the authors should at least address its ramifications in their data interpretation. At a minimum, authors should perform an appropriate analysis to determine which cells are dropping out, using a suitable method such as Milo (Dann et al. Nat Biotech 2022). Then, using known TNF1a pathway genes, they should determine whether these populations express them. Additionally, the limitations sections could be expanded appropriately.

Furthermore, their CD24+ population does not reflect the trends observed in the data and, in my interpretation, may not be significant even at a log₂ fold change > 0 threshold, which raises concerns about how much the flow cytometry experiment can be trusted. Additionally, can the authors provide a CD24 plot to assess how well this marker distinguishes glial from non-glial cells?

At a more granular level, many neuronal subpopulations are differentially abundant (Figure 2C, 2G), and the authors correctly term these "significant" and provide additional interpretations. Still, this potential could be a differentiation artifact, but at least here it is not glossed over. While this may also be due to clonal variability, the authors should assess genes central to their question. In a similar vein to the above, they should check which of those cell types express TNF-alpha genes and could affect the differential responses observed in their bulk assays.

(Remarks on code availability)

Version 2:

Reviewer comments:

Reviewer #4

(Remarks to the Author)

In their rebuttal, the authors have performed additional analyses and provided additional plots to support their claims about the underlying cell-type abundance in the different ENS cultures.

Here are some comments I have:

1) If I understand correctly, scProportion lets the authors set a threshold. This threshold is arbitrary and is used as a null hypothesis. The authors cite a log₂fold change threshold of 1 (a doubling of the population). In the second, they did a more permissive threshold of 0.58 (a 50% increase in population). The second threshold, if I understand the error bars, falls within the scProportion confidence intervals. I would argue that the most rational null is that there is no change in the population, which corresponds to log₂fc=0. I understand that the authors are not concerned with this. However, this needs to be explicitly written in the main text: "with a null hypothesis of 50%..."

2) I don't know why the Milo plots look like this, but having lfc of 0 everywhere (Figure S1D) does not pass the smell test. The authors should double-check that the analysis was conducted properly and that the figure is displayed correctly. In the general clusters Supp Table, deposited log-fold changes range from -8.3 to 7.3. There are similar symptoms in the glia populations.

3) By looking at the Milo tables, it appears that the "permissive" cutoff of 0.2 was selected so that neurons pass this cutoff. If the authors, as they suggest, think their study is not properly powered to support this analysis, perhaps this analysis should be omitted from the manuscript.

4) The flow cytometry plots are helpful. The authors measure ~80% of their cells. The scRNAseq data support the existence of a CD24^{high}/low plot. However, it is not clear how the high- and low-population groups map to the data. While I can definitely infer the CD24-high from the contour lines. However, judging from the contour lines, the CD24-low peak is not as clear as in the cited paper (Windster et al., Fig. 2a). Moreover, the authors use a negative-stain control to set a negative-low cutoff. This cutoff is very close to the peak of cells absent from the main data, as indicated by the contour lines. Could this peak (which is absent in the unstained control) correspond to CD24-low cells? Can the authors provide a justification for this? If the general abundance analysis is inconclusive, which is quite challenging in my opinion at this stage, I suggest expanding the limitation section appropriately.

Version 3:

Reviewer comments:

Reviewer #4

(Remarks to the Author)

Thank you for the clarifications

Reviewer #1 (Remarks to the Author):

Ghirotto et al aims to examine how α -synuclein (α -syn) affects the enteric nervous system (ENS) in Parkinson's disease (PD) and the role that inflammation plays in this process. Given the focus on the ENS as a potential originating site of PD α -syn pathology this is an important question to examine. As model system they utilise three induced pluripotent stem cell (iPSC) lines from patients with SNCA triplication and their previously generated isogenic controls, which they differentiate to enteric neural lineage. To my knowledge, this is the first attempt to model PD pathology in iPSC-derived enteric neural cells.

The authors perform a thorough multi-omic analysis including scRNAseq, proteomics and metabolomics as well as multiple functional assays. They confirm the validity of the ENL cultures using scRNA comparison to a human gut atlas. Overall, this is exciting and well-performed work.

However, some of their conclusions and interpretations of the data are not fully supported. This is mainly related to their interpretation of the effect of TNF- α exposure on SNCA 3x cultures where in several assays there is not actually a significant difference between untreated and TNF- α treated SNCA 3x ENLs and/or the isogenic control ENLs response to TNF- α is identical to the SNCA 3x ENLs.

Reply: We thank the reviewer for their constructive and overall positive review of our manuscript. We appreciate their helpful critique regarding the interpretation of the TNF- α exposure data in SNCA 3x ENLs. To provide stronger support for our conclusions, we have now performed several new, rigorous analyses and clarified data interpretation and statistical analyzes. In this sense, we conducted flow cytometry of the iPSC-ENLs with separation of enteric neurons and glia followed by t-SNE analysis, which clearly identified specific cell populations that are exclusively enriched in the SNCA 3x + TNF group; and we also performed additional Seahorse experiments, where the mitofuel test revealed a significant and unique metabolic impairment in the SNCA 3x+TNF group, characterized by a marked increase in glutamine dependency. These new data mechanistically reinforce our finding that the combination of SNCA triplication and TNF- α exposure unmask a distinct cellular and metabolic phenotype.

Major concerns/questions:

1. The authors show that the number of enteric neurons is significantly decreased in SNCA 3x ENL cultures at d70 based on scRNAseq whereas numbers of glia and proliferating progenitors are similar. They do not address how this difference in the number of enteric neurons arises between SNCA 3x and isogenic controls. Are fewer neurons generated or have they died during the differentiation?

An important question arising from this, is how the different cellular composition of the ENLs cultures between SNCA 3x and isogenic controls affect the downstream analyses? E.g. in Fig. 3 A-B where the mitochondrial content and morphology is analysed and found decreased in SNCA 3x. Is this a result of the difference in cell type composition/morphology as mitochondrial morphology and content can differ between

neurons and glia? The difference in neuronal numbers is also important to keep in mind in the interpretation of the MEA data.

Reply: We thank the reviewer for this insightful comment. In our original submission, we reported a significant reduction in enteric neurons in SNCA 3x ENL cultures at d70. We agree that the cellular composition is an important question and followed it up by performing further analysis and added additional data:

We had previously performed gene expression analyses in SNCA 3x ENLs (Fig 1B and Fig S1B) These qPCR analysis of neuronal markers (*TUBB3*, *PHOX2B*, *ELAVL4*, *HOXB3*) showed no differences between Iso and SNCA 3x cultures throughout the different timepoints analyzed (d6, d40 and d70).

Following another reviewer's feedback to go for a more stringent statistical analysis reflecting the use of replicates, we reprocessed our calculations: We now reanalysed the data in figure 1E using the same compositional analysis criteria that we used for enteric neuron and glia subclusters (Figure 2C and 2G), which were absolute $\log_2FC > 1$ and $FDR < 0.05$, applied to the three clusters shown in Figure 1D (enteric neurons, glia and proliferating). With this more stringent analysis, the reduction in enteric neurons is no longer statistically significant (Figure 1E, see below that all three clusters were classified as n.s.).

Moreover, Figure 4D demonstrates comparable cell viability between the two groups at the end of the differentiation. Importantly, we have validated all our data in new flow cytometry experiments, in which we separated enteric neurons and glia and calculated the frequency of those populations in our groups, both in a basal level and under TNF exposure, which showed no significant differences in enteric neuron and glia proportion between Iso and SNCA 3x for all conditions (Figure 4K, see below).

Together, these results point towards non-detectable differences in the number or survival of enteric neurons between Iso and SNCA 3x ENL cultures under our experimental conditions. The consistent cellular composition of the groups supports our central interpretation that the observed SNCA 3x phenotypes, including the mitochondrial structural changes (Figure 3 A-B) and the MEA functional deficits, are due to genotype-specific metabolic and molecular dysfunction within the ENLs themselves, rather than changes in cell type composition. The text has been adapted accordingly in the results section, **highlighted in yellow** in the results section.

2. Of the three overall subpopulations of cells, only glia cells show significantly increased SNCA expression based on scRNAseq. This is interesting in relation to the

finding of decreased numbers of neurons in SNCA 3x cultures. The authors show immunofluorescent stainings for GFAP, HuC/D and a-syn I Fig. 1C. Can they based on these stainings determine if the cell type specific SNCA expression differences translate to the protein level?

Reply: We appreciate the reviewer’s comment regarding the cell-type specific SNCA expression observed in the scRNA-seq data, which prompted us to perform more rigorous experiments to clarify and validate the data. We could not reliably determine protein differences using the immunofluorescence images in Figure 1C because the antibodies for α -synuclein (SNCA) and the neuronal marker HuC/D are from the same host species, precluding effective co-localization and quantification via immunocytochemistry (ICC). To definitively address this, we implemented a precise flow cytometry (FACS) workflow [Windster et al. 2023, EMBO Rep, doi:10.15252/embr.202255789] for better cell-type segregation and quantitative analysis of α -synuclein levels within enteric neuron and glia populations. We then used a previously established α -synuclein staining protocol [Leupold et al. 2022, Front Neurol, doi: 10.3389/fneur.2022.869103] to access this protein’s levels using flow cytometry. These data (Figure 4J) now clearly demonstrate that TNF- α exposure unmasks a significant increase in α -synuclein protein levels in both enteric neurons and glial cells in the SNCA 3x cultures, indicating that both key enteric cell types are susceptible to pathology in the context of inflammation. Furthermore, to support the transcriptomic basis for this finding, we have included ridgeplots detailing the SNCA expression across specific enteric neuron (Figure S2E) and glia subclusters (Figure S2H), where the comparative expression changes between Iso and SNCA 3x are more distinctly visible. This information has now been added to the results part and highlighted in light green.

3. The authors conclude that the scRNAseq differential analysis reveal mitochondrial dysfunction in SNCA 3x ENLs, but do not refer to any figure to support this (line 161-162). The enrichment analyses in Fig. 2D+F do not include mitochondrial related terms besides from “regulation of lipid metabolic process” and “integration of energy metabolism”. Similarly, the signalling pathways identified as altered by the CellChat analysis as presented in Fig. S1H and Fig. S3E+H do not appear particularly related to mitochondria although this is mentioned in the results section? EGF and NOTCH signalling have many other more prominent functions and the majority of the other

factors listed appear more related to axogenesis, neurogenesis and extracellular matrix etc.

Reply: We agree with the reviewer for pointing out that our previous statement about mitochondrial dysfunction in the scRNAseq data needs more explanation. Our previous enrichment analysis was focusing on the differentially expressed analysis between subclusters and not between Iso and SNCA 3x. The genes from the heatmap were manually curated to search for mitochondrial-linked genes, which we agree was rather a biased approach. Therefore, in our revised version we have included in figure 2D and 2H the integrated enrichment analysis results obtained when comparing Iso vs SNCA 3x instead of subcluster markers. These updated plots indeed show several terms that directly point out to energetic dysfunction induced by alpha-syn triplication, such as “fatty acid transport”, “acetyl-CoA metabolic process” and “steroid metabolic process” in enteric neurons (Fig 2D, see below) and “mitophagy”, “mitochondrial ATP synthesis coupled electron transport”, “release of cytochrome c from mitochondria”, “response to oxidative stress” and “mitochondrial protein import” in enteric glia (Fig 2H, see below). We also updated the visualization for the enrichment plots (Fig 2D and 2H), which are now coloured based on adjusted p-value instead of subcluster as done before, since the subcluster numbers are already on the x-axis.

We then based the heatmap gene selection exclusively on the genes that were found enriched in pathways directly linked to mitochondria, such as the ones cited above and depicted in table S5 and selected the subclusters of enteric neurons and glia that were mostly associated with those pathways (namely 9, 3 and 4 for neurons and 6, 1, 11, 0 and 5 for glia). Therefore, the genes and subclusters selected for plotting the heatmaps (figure 2B and 2F, see below) now directly match the enrichment analysis performed in figures 2D and 2H, showing the transcriptional changes between the conditions that are behind all the significant mitochondrial-associated pathways shown.

These results (heatmap and enrichment) have now been altered in results and discussion section **highlighted in turquoise**.

Regarding CellChat analysis, our intention was not to suggest that these pathways directly reflect mitochondrial function, but rather to provide a broader view of altered intercellular communication in our dataset. To avoid potential overinterpretation and given that with the updated enrichment analysis in Figures 2D and 2H we have sufficient evidence that points to mitochondrial dysfunction in SNCA 3x ENLs, we have revised the CellChat analysis and now focus solely on the most pronounced and mechanistically relevant changes in communication architecture. This refined analysis now highlights specific, significantly altered ligand-receptor (LR) pairs in SNCA 3x cells. For instance, we highlight the enrichment of *VEGFA/B-VEGFR1* communication, which regulates cell survival and vascularity, directing interactions between enteric neurons and glia (Figure S11, see below). Similarly, at the subcluster level, we draw attention to the *EGF-EGFR* pair, which is linked to intestinal mucosal healing and where vulnerable glia specifically signal to glial partners engaged in mitochondrial protein import, providing a direct link between intercellular signalling and the cellular effort to mitigate structural mitochondrial stress (Figure S4J, see below). These specific LR pairs in glia, along with the de novo BMP signalling in enteric neurons associated with neurite fasciculation and orientation (Figure S4F, see below), serve to contextualize the vulnerability of the ENS by demonstrating that α -syn accumulation triggers a fundamental remodelling of the communication landscape toward survival and neuroinflammatory signalling, setting the stage for the enhanced metabolic collapse observed under inflammatory conditions. Accordingly, we updated the results and discussion sections, **highlighted in rosa**.

Figure S11

Figure S4F

Figure S4J

4. The proteomics data is reported with p-values and there is no mention of correction for multiple testing. Similarly for the metabolomics data only p-values are reported. In large scale analyses such as these, correction for multiple testing should be applied and q-values/adjusted p-values reported.

Reply: We thank the reviewer for this important point. We agree that correction for multiple testing is a critical consideration in large-scale omics studies. In the present work, the number of biological replicates was limited to three independent isogenic control lines and three SNCA 3x lines, which restricts statistical power. Under these conditions, applying standard multiple-testing corrections (e.g., FDR) results in very few, if any, features passing the adjusted significance threshold.

To address this, we followed well-established, peer-reviewed methods for proteomic and metabolomic analyses, as previously described in [Gerez et al 2019, Science Translational Medicine, <https://doi.org/10.1126/scitranslmed.aau6722>] and [Bierling et al 2024, Cell Reports, <https://doi.org/10.1016/j.celrep.2024.113739>], reporting nominal p-values to highlight candidate differences. These findings were then interpreted in the context of orthogonal validation datasets, including scRNA-seq-derived mitochondrial signatures, TOM20-based morphological assessments, proximity ligation assays, several Seahorse analysis and Multi-Electrode Arrays, which together provide convergent evidence that the observed alterations are biologically meaningful.

We have now added the FDR-adjusted q-values alongside nominal p-values in Suppl. tables S7 and S9, allowing readers to assess both metrics. Additionally, we have acknowledged in the limitations section that the small number of lines limits statistical power for omics analyses and that the findings should be interpreted as discovery-level candidates. We note that this approach is consistent with prior iPSC studies, including our own work using iPSC-derived astrocytes (Ghirotto et al 2022, *Annals of Neurology*, <https://doi.org/10.1002/ana.26336>), where n=3 biological replicates per group were used to generate metabolomic datasets that were subsequently validated by complementary assays. We believe that this transparent reporting, combined with convergence across independent assays, addresses the reviewer's concern and provides a robust, reproducible basis for the conclusions drawn from these large-scale datasets.

5. The authors conclude from the proteomic and metabolomic analyses that MAS and TCA cycle is impaired by TNF-a in SNCA 3x. Besides from AATC are any other MAS or TCA cycle proteins changing with TNF-a exposure? In the analysis of patient data in Fig. 8 additional MAS enzyme expression levels are examined. Are any of these altered in SNCA 3x ENLs? All the metabolites shown in Fig. 5E are lower in SNCA 3x than isogenic control. Could this be a consequence of the generally lower mitochondrial content identified in Fig. 3A-B?

We thank the reviewer for this helpful suggestion and have now examined all four MAS enzymes from Figure 8 in the proteomics dataset. As shown in the new plots, AATM (*GOT2*) did not change with TNF in either genotype. MDHC (*MDH1*) was significantly decreased by TNF specifically in SNCA3x neurons, indicating a treatment effect within this genotype, but with no differences between genotypes upon TNF exposure. MDHM (*MDH2*) showed consistent differences between SNCA3x and isogenic controls under both basal and TNF conditions, suggesting a baseline genotype effect rather than a TNF-specific response. By contrast, AATC (*GOT1*) (Fig 5G) exhibited the clearest pattern of selective vulnerability: it was significantly reduced by TNF in SNCA3x and also differed between genotypes under TNF exposure.

Taken together, these data indicate that MAS alterations are not uniform but rather enzyme-specific, with AATC emerging as the most compelling candidate linking TNF exposure to MAS dysfunction in the SNCA 3x background. We have now added these graphs in Suppl Fig 6 (A-C, see below) and adapted the text accordingly in the results section (highlighted in grey). We further note that all proteomic and metabolomic values were normalized to total protein input prior to statistical testing, rather than to cell or mitochondrial abundance, ensuring that these differences cannot be attributed to lower mitochondrial content (Fig. 3A–B) but instead reflect genuine pathway-specific changes.

Figure S6A-C

6. The statistical analysis of a-syn-mito co-localisation with PLA in Fig. 6C should be redone to show the average for each of the cell lines – as all other figures in the paper – instead of showing data for individual cells. The figure legends states $n = 3$, but the statistical analysis appears to be performed on individual cells. It is unclear whether TNF- α significantly increases a-syn-mito PLA in SNCA 3x if analysed in this way? However, it appears that TNF- α increases a-syn-mito PLA significantly for isogenic control, which is currently not mentioned/discussed?

Also, could the authors please discuss the discrepancy between the results from Fig. 6C to Fig. 6E where there is no increase of a-syn in the mitochondrial fraction with TNF- α compared to untreated for either SNCA 3x or isogenic control?

The western blot in figure 6D should show the starting material / un-isolated samples for comparison to demonstrate that successful isolation of mitochondria has been achieved.

Why are basal conditions without TNF- α not included in Fig. 6F when the earlier data has clearly shown that baseline mitochondrial dysfunction is present in SNCA 3x ENLs? This makes it unclear whether TNF- α is having an effect or the difference is caused by SNCA 3x alone? Please include vehicle treated as well.

Overall, the headline “TNF- α increases a-syn-mitochondria interactions and induces oxidative stress in SNCA 3x ENLs” is currently not fully supported by the presented data.

Reply: We thank the reviewer for the constructive feedback regarding the analysis of the a-syn-mito co-localisation with PLA in Fig. 6C. We have averaged the PLA signal for each condition across $n=3$ independent experiments, as requested. However, due to the very small sample size ($n=3$), this averaging did not yield any statistically significant differences between conditions (see below – figure not included in the current manuscript).

Therefore, we decided to first normalize the data to log (counts + 1), as suggested by another reviewer and then keep the analysis on individual cells, which is standard for PLA data, as supported by several high impact publications in the field: [Bengoa-Vergniory N et al. 2021, *Acta Neuropathol Commun*, doi:10.1186/s40478-020-01117-y], [Jensen et al. 2024, *npj Parkinsons Dis*, <https://doi.org/10.1038/s41531-024-00841-9>], [Almandoz-Gil et al. 2018, *Front. Neurol.*, doi: 10.3389/fneur.2018.00180], and [James et al. 2019, *J Biol Chem.*, doi:10.1074/jbc.RA118.007283]. We acknowledge that TNF- α does indeed increase the PLA

signal in the Isogenic control cells as well; however, the increase is significantly greater in SNCA 3x ENLs compared to the Isogenic control cells upon TNF- α treatment, and the SNCA 3x basal condition is also significantly different from its TNF- α treated condition. We have now updated Figure 6C accordingly (see below) and also acknowledged in the results section (highlighted in blue) the increase in PLA signal in Iso ENLs upon TNF treatment.

In fact, since the previously presented isolated mitochondria analysis (previous Fig. 6E) did not bring any new information to this figure, we replaced it with a sophisticated t-distributed Stochastic Neighbour Embedding (tSNE) flow cytometry analysis of iPSC-ENLs both in a basal level and under TNF treatment (now Fig. 6D-E). These exciting new data allowed us to detect a distinct sub-population of MitoSOX-high cells exclusively in SNCA 3x enteric neurons (Fig 6D, see below, as indicated by the arrow) and enteric glia (Fig 6E, see below, as indicated by the arrow) treated with TNF- α , as highlighted by the tSNE contour plots in orange, thus providing a functional and visual explanation for the differential effects of TNF- α and lending robust support to our hypothesis regarding the effect differences, even where simple bar graphs fail to reach statistical significance. We have now updated the results and discussion sections accordingly (highlighted in blue). Finally, in line with these new data we now changed the headline to “*TNF- α increases α syn mitochondria interactions and highlights oxidative stress- associated populations in SNCA 3x ENLs*”. We hope that the reviewer finds this new and specific analysis and line of reasoning convincing.

7. Fig. 7B-C: the authors state that TNF- α -treated SNCA 3x ENLs showed reduced mitochondrial respiration including ATP-linked OCR. However, the effects are close to identical for the TNF- α -treated isogenic control ENLs. This should be reflected in the interpretation of the data. Overall, the effects of CSB6 appear overall similar independent of genotype. Currently, this is only really mentioned in the limitations of the study section. Only the glutamine dependency seems to really differ between SNCA 3x and isogenic control, independently of TNF- α exposure, with normalisation after CSB6 treatment for SNCA 3x.

The statement that CSB6 promoted a shift toward fatty acid oxidation is confusing. FAO levels are unchanged for SNCA 3x between the basal, TNF- α - and TNF- α +CSB6-treated conditions – and significantly decreased in isogenic control after CSB6 treatment?

For the Mitotracker green assay the authors note a trend towards an increase in dysfunctional mitochondria in SNCA 3x with TNF- α . However, the trend towards decreased dysfunctional mitochondria at baseline are not mentioned. CSB6 decreases the percentage dysfunctional mitochondria to close to 0 independently of genotype. However, the authors only mention the effect for SNCA ENLs as the result is apparently only significant for this group. The average for TNF- α treated isogenic control is around 20% and with CSB6 around 1 with no visible arrow bars.

Reply: We thank the reviewer for this important suggestion. We would like to transparently report that while performing the additional flow cytometry experiments for the revised version of our manuscript, we noticed that the CSB6 (a blue compound) is actually having a high fluorescence in the same channel as the mitotracker deep red (APC), as shown in the plot below, highlighting that an unstained control in the presence of CSB6 has a lot of positive signal in the APC channel, something that we missed before, making it difficult to actually claim the conclusions for the drug effect derived from these previously shown data. The fact that CSB6 has a fluorescence in the APC channel has been added to the limitation section of the manuscript.

Therefore, in response and in accordance with a suggestion from another reviewer, we performed additional Seahorse mitostress assays that included CSB6-only treatment in both isogenic and SNCA 3x cells to rather focus the analysis on the functional effects of the compound. These new data better clarify the scope of CSB6's effects. Two-way ANOVA (factors: Genotype and Condition, with all four conditions included) revealed a significant Genotype \times Condition interaction for both basal ($F=3.3$, $p=0.0475$) and ATP-linked OCR ($F=4.677$, $p=0.0157$), indicating that the effect of treatment was not uniform but instead depended on genotype. The main effect of Condition (our different treatments) was significant across multiple mitochondrial parameters, including basal OCR, non-mitochondrial OCR, ATP-linked OCR, maximal respiration, and proton leak (all $F>5$, $p<0.01$), reflecting broad differences among Basal, TNF, CSB6, and TNF+CSB6 groups. By contrast, the main effect of Genotype was not significant for all parameters, indicating that overall, SNCA 3x and isogenic cells do not differ when averaging across all conditions, but diverge selectively under specific treatments. Direct genotype comparisons using Two-way ANOVA with Sidak post-hoc under CSB6-only conditions showed that SNCA 3x cells had higher basal ($p=0.0163$) and ATP-linked ($p=0.0060$) OCR than isogenic controls, despite the absence of a main genotype effect across all conditions. This indicates that while CSB6 boosts mitochondrial function in both Iso and SNCA 3x cells, the enhancement is more pronounced in the vulnerable SNCA 3x genotype. Together, these results indicate that CSB6 provides broad mitochondrial support and that its effects are context- and genotype-dependent, producing the largest relative benefit in SNCA 3x cells, both at baseline and under TNF challenge. The figure has been updated

(see below) and this information has been adapted in the results and discussion sections (highlighted in purple).

Additionally, we also repeated the Seahorse mitofuel assays including CSB6-only controls, which now highlighted better the specific effect of TNF on increasing glutamine dependency in SNCA 3x ENLs, which is significantly rescued by CSB6 (Fig 7E, see below). With the experimental repetitions, the trend for increased glutamine dependency already in SNCA 3x ENLs in a basal level is much less evident compared to the previously shown data, indicating that this is a specific TNF-driven effect in those cells which is dependent on the genotype. Furthermore, the repetition of the FAO dependency analysis improved the discrepancies reported previously, with the new data showing no differences between Iso and SNCA 3x for basal, TNF and CSB6-only conditions and a trend for increased FAO dependency in SNCA 3x ENLs in the CSB6+TNF group (Fig 7F, see below), suggesting that might be a compensatory response of those cells. This information has been adapted in the results and discussion sections (highlighted in purple).

Finally, we now also included a comprehensive flow cytometry analysis for iPSC-ENLs, focusing mainly on differences induced by TNF, due to the beforementioned CSB6 fluorescence issues. Similar to the mitoxox analysis for Figure 6, we also applied the t-SNE algorithm to try to detect more specific populational differences between the genotypes under the basal and TNF-stimulated conditions. Indeed, according to our previous results, there is a clear trend for decreased functional mitochondria in SNCA 3x enteric neurons and glia already in a basal level (Fig 7G-H, see below). Interestingly, the tSNE contour plots highlighted a more

pronounced visual decrease in the population of mitodeep red high cells in SNCA 3x enteric glia upon TNF treatment (Fig 7H, see below as indicated by the red arrow), although not significant in the bar plots. We now adapted and discussed all these data in details in the results section of the manuscript (highlighted in purple).

8. The authors write in the abstract that they show a-syn upregulation as a general hallmark of in intestinal inflammation in human gut tissue. They mention examining SNCA expression data – together with GOT1 expression – in intestinal inflammation patient samples. However, the SNCA expression is only shown in correlation to GOT1 expression and no formal comparison of a-syn levels between inflamed and uninflamed tissue is shown. Could the authors please show the SNCA expression levels the same way the show GOT1 levels in Fig. 8B? Similarly, the SNCA expression data from inflamed and non-inflamed gut tissue from the IBDome dataset should be presented and not only correlations with GOT1.

The immunofluorescent staining for GOT1/AATC and a-syn on colon biopsies should be performed on more than one patient and one control if anything is to be concluded from it.

Reply: We thank the reviewer for their careful reading and constructive comments here. We agree that it is important to demonstrate SNCA upregulation in inflamed intestinal tissue directly, rather than relying solely on correlations with GOT1 expression. In the microarray data analyzed, SNCA expression is significantly higher in UC patients and decreases in patients who respond to infliximab therapy, supporting its association with active TNF-driven inflammation (Fig 8C, see below). This observation is confirmed in the IBDome dataset, where SNCA expression is significantly elevated in inflamed UC tissue compared to non-inflamed

tissue (Fig 8F, see below). Finally, we now have now included a quantification of the immunofluorescence of AATC and α -syn in Fig 8J in n=3 controls and n=3 UC patients (Fig 8K, see below), showing that AATC is significantly decreased whereas α -syn is upregulated in UC patients, corroborating our results. This information has been added to the manuscript in the results and discussion sections (highlighted in orange).

To note, another reviewer questioned us if infliximab altered the correlation between GOT1 and SNCA, which instigated us to perform additional correlation analysis between *GOT1* and *SNCA* in UC responders and non-responders before therapy as well (see below) and since the UC responders showed already a non-significant correlation prior to infliximab (as shown by the red arrows below, the data was not conclusive and we decided to take it out from the manuscript, since we believe the other data present in the updated Figure 8 already provide sufficient evidence to support our claims.

Minor concerns:

1. Fig. 2A+E Can the authors please provide legends with naming of the different subclusters of cells in the scRNAseq data? It is difficult to interpret the data when we only have subcluster numbers.

Reply: We thank the reviewer for this helpful suggestion. To improve interpretability, we annotated the subclusters in Fig. 2A and 2E. The annotation was performed by integrating our dataset with the enteric neuron and glia reference atlas of Drokhylyansky *et al.* (Cell, 2020) using SingleR (as in Suppl Fig 2), which assigns cell-type labels based on transcriptomic similarity to the reference, as also suggested by another reviewer. For each subcluster, we determined the majority SingleR label and then confirmed the annotation with cluster-specific marker genes identified by differential expression analysis. In cases where the same label appeared across multiple clusters, we appended the top discriminating marker gene to generate unique and biologically meaningful designations. These final annotations have now been added to the figure legends (see below). We also incorporated this into the results section and provided an integrative interpretation with all the scRNAseq data in the figure (highlighted in gold).

2. Is there a correlation between the subclusters of cells with the highest changes in DEGs and their SNCA expression? E.g. for neuronal subcluster 8 which is reduced in abundance in SNCA 3x ENLs the SNCA expression levels would be interesting to know.

Reply: We thank the reviewer for this valuable suggestion. To address it, we examined the relationship between SNCA expression levels (Fig S2E and S2H, now as ridgeplots, see below) and the subcluster abundance changes (Figure 2C and 2G) in the SNCA 3x condition. The results revealed a heterogeneous relationship. In enteric neurons, we observed an overall inverse association in the subclusters with the strongest compositional shifts: subcluster 2, which is reduced in abundance in SNCA 3x, displayed higher SNCA expression, whereas subcluster 13, which expands in SNCA 3x, showed lower SNCA expression. This suggests that neuronal vulnerability and expansion are not simply proportional to SNCA transcript abundance but may reflect how different neuronal subtypes respond to α -synuclein overexpression at a functional or stress-response level. In contrast, glial populations showed a more consistent pattern: SNCA expression was uniformly higher across most glial subclusters in SNCA 3x, irrespective of whether a cluster expanded or contracted. This

lineage-specific difference indicates that glia mounts a uniform transcriptional response to SNCA triplication, whereas neurons show subcluster-specific heterogeneity, with changes in abundance not directly explained by SNCA expression levels alone. These data are now depicted in the results (highlighted in light green).

3. If a-syn has been identified by the proteomic analysis, how are the levels in the different conditions?

Reply: We thank the reviewer for raising this point. SNCA was detected in the proteomics dataset, where peptide abundance showed a modest decrease within SNCA 3x ENLs following TNF; however, this likely reflects the limited sensitivity of proteomics for insoluble or modified α -synuclein. Biochemical fractionation (Supl Fig 5) revealed that soluble α -synuclein is already slightly elevated in SNCA 3x at baseline, whereas TNF exposure primarily trends toward increased insoluble α -synuclein. These data, together with the ELISA results (Fig 4) showing higher total α -synuclein in SNCA 3x under TNF, suggest that TNF promotes accumulation in the insoluble pool, whereas proteomics captures only the soluble fraction. We therefore interpret the proteomics trend as complementary rather than contradictory to the ELISA findings.

4. Could the authors also please report how many proteins and metabolites were identified for the reader to understand how large a proportion of the identified proteins and metabolites differ between the different condition.

Reply: While this information was available for the proteomics data (highlighted in salmon), we have now added this information to the metabolomics results (highlighted in salmon) to clarify for the readers.

5. Can Fig. 5H please be enlarged so that it is more readable.

Reply: Since we feel that the previous PPI was not bringing additional new information to the figure, we have now replaced figure 5H for a western-blot validation of the OMICs data from figure 5 in which we analyzed and quantified AATC levels in iPSC-ENLs in a basal level and treated with TNF (see below).

6. Please define in the results section (line 344) that GOT1 = AATC.

Reply: We have changed it accordingly to clarify for the readers (highlighted in orange) by adding AATC in brackets.

Reviewer #2 (Remarks to the Author):

This manuscript presents an ambitious and integrative study using human iPSC-derived enteric neural lineages (ENLs) from SNCA triplication Parkinson's disease (PD) patients to explore the interplay between alpha-synuclein accumulation, inflammation, and metabolic dysfunction in the ENS. The authors employ a comprehensive multi-omic approach (scRNA-seq, proteomics, metabolomics), functional assays (electrophysiology, Seahorse), and validation in human UC samples to support a model in which TNF- α exposure unmasks mitochondrial and synaptic vulnerability in SNCA 3x ENLs through disruption of the malate-aspartate shuttle (MAS). The study is highly relevant, well-executed, and presents a novel mechanism with broad implications. However, several aspects of the manuscript require clarification or additional analysis to strengthen the conclusions and ensure appropriate interpretation.

Reply: We are grateful to the reviewer for their highly positive assessment of our work, recognizing the ambitious, integrative nature of the study and the novelty of the proposed mechanism linking α -syn accumulation, inflammation, and metabolic vulnerability via the malate-aspartate shuttle (MAS) disruption. We have carefully addressed all points raised regarding clarification and additional analysis in the subsequent specific responses, performing several new experiments, refining our statistical rigor, and substantially clarifying the interpretation of our data. We are confident that these revisions significantly strengthen the manuscript's conclusions and ensure the appropriate interpretation of our findings.

1. Several results suggest that SNCA expression is not significantly different at baseline in neurons, despite conclusions about SNCA-driven effects. It would help to show quantifications and expression levels across neuronal and glial subclusters and assess SNCA⁺ cells specifically. Consider module scoring across SNCA⁺ neurons and glia, or correlation with SNCA-interacting genes.

Reply: We thank the reviewer for this thoughtful comment. As shown in Fig. 1G, SNCA expression is overall higher in enteric neurons when considering broad clusters, and in the Iso vs. SNCA 3x comparison, significant differences were evident in glia (Fig. S1E). Since we had not previously examined SNCA at the subcluster level, we reanalyzed our scRNA-seq data accordingly and now highlight using Ridgeplots the expression levels of SNCA within enteric neurons (Fig. S2E, see below) and enteric glia (Fig. S2H, see below). This analysis revealed numerous significant differences between genotypes in both neuronal and glial subclusters. In neurons, we observed a heterogeneous pattern, with some subclusters showing higher and others lower SNCA expression in SNCA 3x compared to Iso, whereas glial cells displayed a more consistent increase in SNCA expression across several subclusters.

Given our revised analysis, which now detects direct and significant differences in SNCA expression at the subcluster level (Figure S2E and S2H, below), we believe this more targeted approach supersedes the need for the more indirect method of module scoring or broad correlation analyses. Our direct expression analysis revealed a highly heterogeneous relationship with subcluster abundance changes (Figure 2C and 2G) that is crucial to the underlying biology. Specifically, while glial populations showed uniformly higher SNCA expression across most subclusters irrespective of expansion or contraction, enteric neurons exhibited an overall inverse association in the subclusters with the strongest compositional shifts. This lineage-specific difference suggests that neuronal vulnerability and expansion are not simply proportional to SNCA transcript load but instead reflect how different ENS cellular subtypes employ distinct regulatory or compensatory mechanisms in response to α -syn overexpression. We now explicitly highlight this neuronal heterogeneity and its potential functional implications in the revised results section (**highlighted in light green**).

2. Figures 3–7 present pooled ENL results despite many cell-type-specific conclusions. Where feasible, separate neuron vs glia in imaging, flow, and Seahorse analyses or explicitly qualify conclusions that cannot be deconvolved.

Reply: We appreciate the reviewer's suggestion on resolving cell-type-specific effects, which we agree is crucial for interpreting results from our mixed iPSC-derived ENL cultures. To ensure our conclusions are appropriately qualified and robust, we implemented a comprehensive strategy: our scRNAseq data (Figures 1-3 and Suppl Figures S1-S4) provides the foundational, cell-type-specific insight into *SNCA* expression, signaling (CellChat), and mitochondrial/metabolic gene expression exclusively at the neuron and glia subcluster level. Crucially, we then developed and executed during the revision of the manuscript a comprehensive flow cytometry pipeline utilizing CD24 expression levels [Windster et al. 2023, EMBO Rep, doi:10.15252/embr.202255789] (Figure S5F, see below) to separate and quantify enteric neurons and glia, allowing us to directly validate critical functional findings throughout the manuscript, such as α -syn accumulation and the TNF- α -induced mitochondrial dysfunction (MitoSOX, MitoTracker Deep Red, CPT1 α) at the resolution of the individual cell type (Figures 4J-K, 6D-E, 7G-H and Supplementary Figures S6F-G) using an unsupervised t-distributed stochastic neighbor embedding (t-SNE) approach to highlight even subtle populational changes. This strategy allowed us to successfully deconvolute functional data and merge it with many of our results. Furthermore, we elected to prioritize flow cytometry for quantification over image-based analysis because the ENL cultures form dense, three-dimensional, ganglioid-like structures, making reliable and unbiased cell-type separation and quantification via standard imaging challenging; flow cytometry offered a superior method for obtaining robust, single-cell resolution data. We acknowledge, and explicitly state in the revised manuscript's limitations section, that certain bulk assays, such as the Seahorse and MEA analysis, cannot be meaningfully separated; for these, the results are qualified to represent the overall phenotype of the *SNCA* 3x ENL network, which we then interpret in light of our cell-type-specific scRNAseq and flow cytometry data. We are confident that this multipronged approach sufficiently addresses the need for cellular resolution in this very complex and challenging model system.

3. The cell types and annotations in the scRNA-seq datasets should be described in more detail. What markers were used to define each cluster? This information is critical for evaluating the biological interpretation of the single-cell results.

Reply: We thank the reviewer for this helpful suggestion. To improve interpretability, we now annotated the subclusters in Fig. 2A and 2E (see below), as suggested by another reviewer. The annotation was performed by integrating our dataset with the enteric neuron and glia reference atlas of Drokhylyansky *et al.* (Cell, 2020) using SingleR (as shown before in Suppl Fig 2), which assigns cell-type labels based on transcriptomic similarity to the reference. For each subcluster, we determined the majority SingleR label and then confirmed the annotation with cluster-specific marker genes identified by differential expression analysis. In cases where the same label appeared across multiple clusters, we appended the top discriminating marker gene to generate unique and biologically meaningful designations. A supplementary table S2 with all the markers per subcluster is also provided, and the top markers per subcluster are displayed in Supplementary Fig S2A (enteric neurons) and S2B (enteric glia). We also incorporated this into the results section and provided an integrative interpretation with all the scRNAseq data in the figure (highlighted in gold).

4. In several figures (e.g., Figure 1E), tests such as Student's t-test are applied to proportions or counts; chi-squared or other appropriate methods should be used.

Additionally, comparisons (e.g., Figure 4C) currently conflate genotype and baseline differences. Comparisons should isolate the effect of TNF within each genotype.

Reply: We thank the reviewer for pointing out the limitations of applying t-tests directly to proportions or counts. Accordingly, and also following another reviewer guidelines we reanalysed the data in figure 1E using the same compositional analysis criteria that we used for enteric neuron and glia subclusters (Figure 2C and 2G), which were absolute $\log_2FC > 1$ and $FDR < 0.05$. With this more stringent analysis, the reduction in enteric neurons is no longer statistically significant (Figure 1E, see below).

We appreciate the reviewer's suggestion on clearly isolating treatment effects and maintaining statistical rigor. We confirm that all comparisons involving more than one condition (excluding Figure 6C) were analyzed using Two-way ANOVA, which is the appropriate test to dissect the effects of Genotype and Condition (treatment). Our rationale for plotting the combined results in the majority of figures is grounded in the F-statistic: the crucial Genotype \times Condition interaction was not statistically significant in most assays, indicating that the TNF- α or CSB6 effect was statistically comparable across the SNCA 3x and isogenic lines, thereby not warranting separate isolation of the genotypes.

Importantly, we did detect a significant interaction for three specific readouts in the Seahorse analysis, namely Basal OCR ($F=3.3$, $p=0.0475$), ATP-linked OCR ($F=4.677$, $p=0.0157$) and the glutamine dependency ($F=4.258$, $p=0.0217$). For these instances, we performed the necessary simple effects analysis, isolating the TNF- α effect within each genotype (see below). Since this isolated analysis did not drastically change the biological conclusions compared to the Two-way ANOVA results (Figure 7C and 7E, pointed by red arrows), we have maintained the consistent Two-way ANOVA visualization to match the presentation of our other functional data. However, we now specifically report the significant interaction effect between condition and genotype for basal and ATP-linked OCR in the Results section (lines 375-377). This approach maintains the superior statistical validity of the Two-way ANOVA framework while ensuring the necessary detail of the non-uniform response is explicitly reported where the underlying biology dictates.

Figure 7C

Figure 7E

5. The strength of the TNF-specific effect in SNCA 3x ENLs is not always evident. some changes may be shared with ISO. Figure 5 should include the full condition matrix (ISO ± TNF, SNCA 3x ± TNF) with clearer annotations.

Reply: We fully agree with the reviewer that clearly dissecting the TNF- α -specific effect within the SNCA 3x genotype is crucial, and we have significantly enhanced the resolution of our analysis and presentation to address this point directly. We refined our Seahorse assays and, critically, used t-SNE algorithms on our flow cytometry data to clearly highlight subtle populational and functional changes induced by TNF- α specifically in SNCA 3x enteric neurons and glia, confirming a unique cellular vulnerability (Figures 4J-K, 6D-E, 7 and Supplementary Figures S6F-G).

Furthermore, as requested, we have updated Figure 5C-D to include the full condition matrix in all proteomics volcano and enrichment plots (see below). These comparisons are now systematically divided into "between genotypes" (Iso vs SNCA 3x – basal level and TNF treated) and "within genotypes" (Iso TF vs Iso basal and SNCA 3x TNF vs. SNCA 3x basal) to rigorously isolate the TNF- α effect on the SNCA 3x line. For the metabolomics data, which focused solely on the TNF-treated condition, we now explicitly report the differences observed between the SNCA 3x and Iso lines under inflammation. Finally, we report the exact number

of differentially expressed proteins and metabolites for each comparison (**highlighted in salmon**), ensuring maximum transparency and clarity in visualizing the TNF-specific effects.

6. The <5% MT gene threshold may bias against metabolically stressed neurons. Suggest re-analyzing with a relaxed cutoff (e.g., 15–20%) and assessing whether stressed populations or DEGs change.

Reply: We are thankful for the reviewer’s feedback. In our study, a <5% threshold was chosen to align with the widely accepted quality control standard in single-cell RNA-seq studies to remove low-quality or dying cells [Heumos et al 2023, <https://doi.org/10.1038/s41576-023-00586-w>] and as observed in the quality control metrics for our data (shown below), the majority of cells across all samples had a percentage lower than 5%. Using this threshold, we filtered cells where mitochondrial transcripts are disproportionately high in comparison to nuclear transcripts due to technical artifacts. Thus, filtering does not remove MT genes themselves; rather, it removes cells in which the proportion of nuclear genes is very low compared to mitochondrial transcripts, typically reflecting low-quality or dying cells. Post filtering, the remaining cells still contain both MT and nuclear genes, therefore preserving metabolically stressed neurons and enabling analysis of stress-related transcriptional changes. By contrast, increasing the MT percentage would keep cells dominated by mitochondrial transcripts, with few nuclear genes detected, increasing noise and limiting the study of transcriptionally active enteric neurons and glia.

7. The manuscript suggests enteric glia compensate for neuronal mitochondrial stress, but supporting data are limited. Consider reframing as a hypothesis or provide additional evidence (e.g., selective pathway upregulation, functional protection).

Reply: We agree that the data for a compensatory response in enteric glia points more to a hypothesis, and have updated the text in the results section accordingly (lines 220-221). However, the underlying molecular architecture provides strong mechanistic context for this hypothesized response. To provide this essential supporting detail, we have now revised all our CellChat analysis to focus solely on the most pronounced and mechanistically relevant ligand-receptor (LR) pairs in the SNCA 3x line. This refined analysis highlights the enrichment of VEGFA/B-VEGFR1 communication, which regulates cell survival and vascularity, directing interactions between enteric neurons and glia (Figure S11, see below). Crucially, we observe the EGF-EGFR pair, linked to intestinal mucosal healing, where vulnerable glia specifically signal to glial partners engaged in mitochondrial protein import, providing a direct intercellular link to the cellular effort to mitigate structural mitochondrial stress (Figure S4J, see below). These specific LR pairs, along with the de novo BMP signaling in enteric neurons (associated with neurite fasciculation and orientation, Figure S4F, see below), serve to contextualize the ENS vulnerability. This signaling data demonstrates that α -syn accumulation triggers a fundamental remodeling of the communication landscape toward survival and neuroinflammatory signaling, supporting the compensatory hypothesis and setting the stage for the enhanced metabolic collapse observed under inflammatory conditions. We have detailed these specific findings in the revised manuscript (Results and Discussion, **highlighted in rosa**).

Figure S4F

Figure S4J

8. It would be great to include CSB6-only controls in functional assays to distinguish TNF-specific rescue from general effects.

Reply: We thank the reviewer for this important suggestion. In response to feedback and to precisely quantify the compound's functional impact, we conducted additional Seahorse mitostress assays, incorporating the CSB6-only treatment alongside the other conditions in both isogenic and SNCA 3x cell lines. Then, comprehensive Two-way ANOVA (analyzing all four conditions across both genotypes) analyses were performed. It successfully identified a statistically significant Genotype \times Condition interaction for both the basal OCR ($F=3.3$, $p=0.0475$) and the ATP-linked OCR ($F=4.677$, $p=0.0157$), a key finding which confirms that the treatment's effect is not uniform but varies based on the underlying genotype. While a significant main effect of Condition ($F>5$, $p<0.01$) was observed across numerous mitochondrial parameters, including basal OCR, non-mitochondrial OCR, ATP-linked OCR, maximal respiration, and proton leak, the main effect of Genotype was non-significant across the combined conditions. This suggests that the SNCA 3x and isogenic lines only diverge selectively under specific treatments rather than differing when averaging across all conditions. Indeed, direct Sidak post-hoc comparisons within the CSB6-only condition revealed that SNCA 3x cells exhibited significantly higher basal ($p=0.0163$) and ATP-linked ($p=0.0060$) OCR compared to isogenic controls. This critical data demonstrates that CSB6 provides broad mitochondrial support, but its genotype-dependent effects yield the largest relative functional enhancement in the vulnerable SNCA 3x cells, both at baseline and when challenged with TNF. This detailed information has been incorporated into the revised figure (7B-C, see below) and updated in the results and discussion sections (highlighted in purple).

Figure 7B-C

Additionally, we also repeated the Seahorse mitofuel assays incorporating CSB6-only controls, which significantly refined our understanding of the specific effect of TNF- α . The new data clearly highlight a TNF-driven increase in glutamine dependency exclusively in SNCA 3x ENLs, a shift that is significantly rescued by CSB6 (Figure 7E, see below). Crucially, with these experimental repetitions, the weak initial trend for increased glutamine dependency in SNCA 3x ENLs at baseline is now much less evident. This improved consistency confirms that the heightened glutamine utilization is a specific TNF-driven effect that is dependent on the SNCA genotype, rather than an intrinsic basal defect. Furthermore, the replication of the FAO (Fatty Acid Oxidation) dependency analysis effectively resolved previous discrepancies: the new data show no significant differences between Iso and SNCA 3x for basal, TNF, and CSB6-only conditions, with only a mild trend for increased FAO dependency observed in the SNCA 3x ENLs in the CSB6+TNF group (Figure 7F, see below). This final trend suggests that FAO usage might represent a compensatory metabolic response within these vulnerable cells. These comprehensive data have been integrated into the results and discussion sections (highlighted in purple).

9. The proteomic/metabolomic findings should be validated by IF or flow cytometry where possible.

Reply: We agree that independent validation of key multi-omic findings is essential to strengthen the manuscript. We have significantly expanded our validation assays for the revised submission to address this point: we performed a Western Blot analysis (Figure 5H, see below) specifically quantifying AATC levels, providing a direct, orthogonal protein-level trend of the same changes identified in our proteomics and metabolomics data central to the MAS hypothesis. This has now been incorporated into results (highlighted in red).

Furthermore, we utilized comprehensive flow cytometry data, incorporating t-SNE algorithms for high-dimensional analysis, to validate mitochondrial function and stress at single-cell resolution using MitoSOX (Figure 6D-E, see below) and MitoTracker Deep Red (Figure 7G-H, see below). We have now updated the results and discussion sections accordingly (highlighted in blue and purple).

Finally, to confirm metabolic activity downstream of the MAS disruption, we performed enzymatic activity assays for two crucial TCA cycle enzymes, Citrate Synthase (Figure S6D, see below) and Succinate Dehydrogenase (Figure S6E, see below). This has been added to the results (highlighted in red). Collectively, these four independent validation methods, Western Blot, single-cell flow cytometry, and two enzyme activity assays, provide robust, multi-level evidence supporting our key proteomics, metabolomics, and functional conclusions.

10. Does infliximab treatment alter SNCA-GOT1 correlation in UC datasets?

Reply: We appreciate this insight from the reviewer. A query from another reviewer prompted us to conduct an additional correlation analysis examining the relationship between *GOT1* (AATC) and *SNCA* expression in UC responders and non-responders, both pre-and post-infliximab therapy. This extended analysis revealed that the UC responders already exhibited a non-significant correlation prior to treatment, meaning the data was inconclusive regarding

the therapeutic effect of infliximab on this specific molecular association (see below, highlighted by the red arrows). To maintain the highest possible clarity and rigor, we have therefore elected to remove this analysis from the manuscript. We are confident that the robust, updated evidence presented in the revised Figure 8 already provides sufficient data to support our core claims regarding MAS disruption and correlation to α -syn in general intestinal inflammation.

11. The quantification of GOT1 and SNCA IF in human tissue should be included and the biopsy metadata should be clarified.

Reply: We now have included a quantification of the immunofluorescence of AATC and α -syn in Fig 8J in n=3 controls and n=3 UC patients (Fig 8K, see below), showing that AATC is significantly decreased whereas α -syn is upregulated in UC patients, corroborating our results. Biopsy metadata information has been added to table S10.

Minor Comments

- Figure S2 heatmaps are difficult to interpret due to inconsistent cluster ordering. Reordering or consistent labeling would help.

Reply: In figure S2, we have grouped the heatmaps (A-B) with respect to clusters to show which clusters share similar expression profiles, that's why they are not consistently ordered but rather reflect clusters that are grouping together.

- Clarify gene/cluster selection in Figures 2B–F, 3C–D.

Reply: For figures 2B-D, we based the heatmap gene selection exclusively on the genes that were found enriched in pathways directly linked to mitochondria, depicted in table S5 and selected the subclusters of enteric neurons and glia that were mostly associated with those pathways (namely 9, 3 and 4 for neurons and 6, 1, 11, 0 and 5 for glia). Therefore, the genes and subclusters selected for plotting the heatmaps (figure 2B and 2F) directly match the enrichment analysis performed in figures 2D and 2H, showing the transcriptional changes between the conditions that are behind all the significant mitochondrial-associated pathways shown. These results (heatmap and enrichment) have now been altered in results section (highlighted in turquoise) to make it more clear for the reader.

For figures 3C-D, we based on canonical genes which are classically known to be regulators of oxidative phosphorylation, ATP production and mitochondrial biogenesis (*ATP5F1B*, *COX4I1*, *NDUFA9*, *PPARGC1A*); mitochondrial fission (*DNM1L*) and mitophagy (*PINK1*) were selected. This is very well described and supported by the literature: [Fernandez-Vizarrá et al. 2021, FEBS letters, 10.1002/1873-3468.13995]; [Fernandez-Marcos et al. 2011, The American journal of clinical nutrition, 10.3945/ajcn.110.001917]; [Jin et al. 2012, Journal of cell science, 10.1242/jcs.093849] and [Vanstone et al. 2016, European journal of human genetics: EJHG, 10.1038/ejhg.2015.243].

- In Figure 4A, define 'total cells' and clarify why neuron/glia are pooled.

Reply: We clarify that “total cells = enteric neurons + glia + proliferating” (all clusters, matching with the rest of the data in figure 4). However, we agree with the reviewer that it is also important to add separated data for glia and neurons, which we now included in suppl fig S5A-B (see below). This is also updated in the results (highlighted in dark red).

- Improve figure labeling for Figures 6C, 7E–F; comparisons are unclear.

Reply: We have improved labelling of figure 6C, taking out the redundant Iso and SNCA 3x naming on the x-axis that was very repetitive and rather stored that information on the different bar colors (black and magenta, respectively, see below). We also now normalized the data to $\log(\text{counts}+1)$, as suggested by different reviewers. Previously, figures 7E-F were redundant

(unlike in Seahorse mitostress assays where showing the curves is very informative) and not adding any extra information so we decided to keep just the bar graphs for the mitofuel data.

• In Figure 8, clarify the comparison in panels B–C; what are we meant to conclude?

Reply: First, we now excluded the previous figure 8C, as the correlation data was inconclusive, as mentioned. Figures 8B and 8C are now evaluating the absolute expression levels of *GOT1* and *SNCA*, respectively, across the different groups. We observed that *GOT1* was consistently downregulated in non-responders across all cohorts, and importantly, in the GSE73661 dataset, it showed a significant restoration in responders following therapy (Figure 8B). This supports the idea that MAS activity is regulated by TNF- α -driven inflammation and clinical response. *SNCA* expression, in the opposite direction, showed a general trend toward upregulation in non-responders, and specifically within the GSE73661 cohort, its expression was significantly lower in responders compared to non-responders following therapy (Figure 8C). This crucial inverse pattern links persistent inflammation to the simultaneous presence of MAS suppression (*GOT1* downregulation) and α -syn accumulation (*SNCA* upregulation). We now clarified this in detail for the readers in the results section (highlighted in orange).

• Describe IBDome and scIBD datasets in more detail: sample size, condition, analysis methods.

Reply: We now provide several additional details from IBDome and scIBD datasets both in the corresponding figure legends in Figure 8 and in the bioinformatics methods.

Reviewer #3 (Remarks to the Author):

The manuscript investigates an important understudied area of the pathology of enteric nerves in Parkinson's. The authors describe the iPSC derived model and the cells generated from it in a good level of detail in the isogenic and SNCA triplication lines. Overall, the data are extremely interesting and add to the field in terms of inflammation and alpha synuclein in gut neurons. Some specific comments are outlined below:

Reply: We appreciate this overall positive assessment of our manuscript by the reviewer and have now carefully addressed all comments raised, which helped us to increase the quality of the data presented throughout the article.

In Figure 2 the authors describe changes in specific cellular subclusters which link mitochondrial stress, lipid homeostasis, cholesterol metabolism and stress responses. Whilst the authors link many of these together, it would be interesting for the authors to comment on the possible link between the changes in cholesterol metabolism and mitochondrial stress.

Reply: We thank the reviewer for raising this important point and for bringing up this suggestion. To investigate the relationship between cholesterol metabolism and mitochondrial stress, we quantified module scores based on key genes regulating cholesterol metabolism (*HMGCR*, *SREBF2*, *SQLE*, *LDLR*, *APOE*, *ABCA1*, *ABCG1*, *CYP27A1*, *CYP7A1*, *SCAP*) and mitochondrial stress responses (*PINK1*, *PRKN*, *BNIP3*, *CLPP*, *HSPD1*, *ATF5*, *SIRT3*, *SOD2*, *OPA1*, *CYCS*), obtained by literature and data repository curation, across all enteric neuron and glial subclusters in our iPSC-ENL single-cell RNA-seq dataset. We then performed a correlation analysis between these module scores for each enteric neuron and enteric glia subcluster (Figure S3A-B, respectively, see below), which detected an overall trend for clusters with a negative correlation showing up in SNCA 3x enteric neurons (i.e. 3, 7, 8) but we only found a significant negative correlation (passing our threshold of absolute R value > 0.5, p < 0.05) between these two modules specifically in Subcluster 12 of SNCA 3x glia, a population that was also substantially decreased within this genotype (Figure 2G). This has now been incorporated to our results section, where we state: “Finally, a module score analysis between key genes regulating cholesterol metabolism and mitochondrial stress responses (Figure S3A-B) detected a significant negative correlation between these two modules specifically in Subcluster 12 of SNCA 3x glia, a population that was substantially decreased within this genotype (Figure 2G)” (highlighted in turquoise).

Although a small part of the cholesterol pathway, mitochondria do play a significant role which is extremely important for bile acid synthesis, also an area of interest in enteric neurons in PD. Hence, could the authors comment on their scRNAseq data in the sub clusters from this point of view?

Reply: We thank the reviewer for raising this interesting point. While mitochondria are indeed central to cholesterol metabolism and bile acid synthesis, to our knowledge bile acids are not classically produced by enteric nervous system (ENS) cells and there are no publications supporting this idea so far. Nevertheless, to further explore this suggestion since we feel this is an interesting and relevant topic, we examined the expression of key genes involved in bile acid biosynthesis (*CYP7A1*, *CYP27A1*, *CYP8B1*, *CYP7B1*, *HSD3B7*, *AKR1D1*, *AMACR*, *BAAT*, *ABCB11*, and *NR1H4*), obtained by literature and data repository curation as a module score in our iPSC-ENL single-cell RNA-seq dataset.

As shown in the average module score heatmaps below for both enteric neuron and glial subclusters, the expression levels of bile acid synthesis - related genes are extremely low overall, with mean expressions around zero and no subcluster showing a strong signal indicative of active bile acid metabolism. These results are consistent with the current understanding that bile acid production is restricted to hepatocytes, rather than being a canonical function of ENS cells. Therefore, given that our study does not directly address bile acid biology, and that our data do not suggest robust expression of the directly-associated pathways in ENS populations, we believe this topic lies outside the main scope of the present manuscript. We are confident that our additional analyses and clarifications will adequately address the reviewer’s query.

For the data presented in Figure 3 specifically (but also to other figures throughout) could the authors be clearer in how many cells were quantified and from how many differentiations of each line? At the moment the figure legend is ambiguous with at least 3 images per line, making 3 biological repeats. To ensure the robustness of the data, it needs to be clear from how many differentiations the data was generated.

Reply: We thank the reviewer for this important comment. Specifically for figure 3B, we now provide in the figure legend the exact number of cells per line that was used to normalize the data. This information has also been updated in the Mitochondrial parameter analysis part of the methods section. Similarly, we also added the total number of cells used to quantify our UC data in figure 8K in the figure legends.

Moreover, we agree that it is critical to provide clarity regarding the number of differentiations and biological replicates. We apologize for the ambiguity in our original figure legends and have now revised them throughout the manuscript to explicitly indicate the number of iPSC lines and differentiations represented in each one.

In our experiments, each iPSC line derived from a different donor served as an independent biological replicate, with $n = 3$ isogenic control lines and $n = 3$ SNCA triplication lines included in the analyses. To confirm reproducibility, the differentiation was repeated at least once (those are our technical replicates) and then the results were averaged for each line whenever available, as indicated in the figure legends. Accordingly, we also included the following sentence in the quantification and statistical analysis methods part, which reflects our data throughout the manuscript: "Unless stated otherwise, all quantitative analysis was performed using independently collected material from three isogenic and three SNCA 3x cell lines (our biological replicates, $n=3$ per genotype). The number of independent differentiation runs (technical replicates) used to generate the data for each panel is specified in the corresponding figure legend. Each data point in the figures represents the average of these technical replicates for each individual line".

This strategy, in which independent iPSC lines constitute biological replicates, is a standard and widely accepted approach in the iPSC field, particularly for complex assays such as single-cell and omics analyses where averaging across multiple differentiation rounds is rarely feasible (e.g. Krach et al., *Nat Commun* 2022, <https://doi.org/10.1038/s41467-022-34419-x>).

We believe these changes address the reviewer's concern regarding reproducibility and transparency.

In Figure 3 the authors postulate that metabolism is shifted in the SNCA triplication lines, and that this may be compensatory. Could the authors expand on why this change would be compensatory response to stress as this is not obvious from the pseudo time experiments shown in Figure 3 without further explanation for the reader.

Reply: We agree that framing the glial role as compensatory is best treated as a hypothesis, and we've adjusted the language in the Results section accordingly (**highlighted in emerald**). Nevertheless, the molecular findings strongly support the mechanistic context for this proposed response. To provide this evidence, we completely re-analyzed our CellChat data, focusing exclusively on the most relevant and highly altered ligand-receptor (LR) pairs in the

SNCA 3x line. This refined analysis highlighted a significant enrichment of VEGFA/B-VEGFR1 communication, which functions to regulate cell survival and vascularity, directing interactions between enteric neurons and glia (Figure S11, see below). Even more critically, we noted the EGF-EGFR pair, known to be linked to intestinal mucosal healing, where stressed glia specifically signal to other glial partners that are transcriptionally engaged in mitochondrial protein import, thus providing a direct intercellular pathway for mitigating structural mitochondrial stress (Figure S4J, see below). These specific LR pairs, along with the *de novo* BMP signaling in enteric neurons (associated with neurite fasciculation and orientation, Figure S4F, see below), serve to contextualize the ENS vulnerability. This signaling architecture demonstrates that α -syn accumulation triggers a fundamental remodeling of the communication landscape toward survival and neuroinflammatory signaling, lending robust molecular support to the compensatory hypothesis and setting the stage for the metabolic collapse observed under inflammation. We have incorporated these specific findings throughout the revised manuscript (results and discussion, highlighted in rosa).

Figure S11

Figure S4F

Figure S4J

The authors data showing the changes in proteome after TNFalpha stimulation are compelling however in the manuscript the authors refer to the SNCA neurons being sensitised to inflammatory stimuli, the data from Figure 3 would suggest this is not general priming but it could be specific, could the authors comment or alter the text.

Reply: We thank the reviewer for this very insightful comment. We agree that "general priming" does not accurately reflect our findings, as this term could imply a non-specific heightened state. Our data suggests a more precise mechanism: SNCA overexpression predisposes the

ENLs to failure. Figure 3 is critical here, showing that under basal conditions, SNCA 3x neurons and glia already exhibit clear baseline vulnerabilities, including compromised mitochondrial function. These pre-existing structural and functional deficits provide a specific, mechanistic basis for the SNCA 3x line's selective sensitivity to inflammatory stimuli observed in subsequent experiments (e.g., TNF- α -induced MAS suppression). The text has been updated in the proteomics results (highlighted in brown) and in the first paragraph of the discussion section (highlighted in brown), highlighting this more precise, evidence-based interpretation, substituting "sensitization" with language that emphasizes this pre-existing metabolic vulnerability.

Throughout the manuscript the figure legends are not detailed enough, only in the Seahorse figure legend of Figure 7 do the authors state that the mean of 3 independent experiments is shown. Could the authors please confirm this for all experiments.

Reply: We appreciate the reviewer highlighting the insufficient detail in our original figure legends and agree that maximum transparency regarding experimental replicates and sample size is essential. As mentioned before, we have now conducted a full revision of all figure legends throughout the manuscript to explicitly address this concern. The revised legends now clearly state the number of iPSC lines (our biological replicates, n=3 per genotype) and the number of independent differentiation approaches (our technical replicates) represented in each specific panel.

Furthermore, we have added a clarifying statement to the Methods section to set the standard for the entire dataset: "Unless stated otherwise, all quantitative analysis was performed using independently collected material from three isogenic and three SNCA 3x cell lines (our biological replicates, n=3 per genotype). The number of independent differentiation runs (technical replicates) used to generate the data for each panel is specified in the corresponding figure legend. Each data point in the figures represents the average of these technical replicates for each individual line". This practice of using independent iPSC lines as the primary biological replicate (n) is a standard and accepted methodology for complex functional assays within the iPSC field and represents the biological variability between different independent donors. We are confident that these comprehensive revisions ensure the robustness and clarity of our data presentation across all figures.

The data generated are compelling for the change in metabolic status after TNF α stimulation. The data in Figures 6 and 7 aim to show a functional effect of this change. However, many of the measurements are of mitochondrial function of OXPHOS rather than of the TCA cycle or beta oxidation. Although the fuel flex experiments show a change, in order to support the data in the previous figures and the conclusion of this change and critical alterations in the malate shuttle, it would be beneficial if the authors were to functionally measure TCA cycle and beta oxidation activity rather than depend on secondary measurements.

Reply: We agree with the reviewer that direct functional measurements are essential to fully support the conclusion that the malate-aspartate shuttle (MAS) and downstream TCA cycle are critically altered. While our Seahorse assays (Figure 7) provide robust evidence of metabolic dependency changes, these are secondary measurements of OXPHOS rather than

direct enzyme kinetics. To definitively validate the central changes in MAS and TCA cycle activity, we have integrated three levels of direct functional analysis:

First, we provide Western Blot validation of AATC (GOT1), the key MAS enzyme (Figure 5H, see below), with a trend for confirming the protein-level changes observed in the omics data, which is that TNF stimulation induces a decrease in AATC levels in SNCA 3x ENLs. Although this trend exists at the basal level already ($p=0.2775$), the addition of the TNF makes it closer to significance ($p=0.1870$). This has now been incorporated into results (highlighted in red).

Second, we performed direct enzymatic activity assays for two crucial TCA cycle enzymes, Citrate Synthase (Figure S6D, see below) and Succinate Dehydrogenase (Figure S6E, see below), which directly assess the capacity of the Krebs cycle. We observed a trend for decrease citrate synthase in SNCA 3x ENLs, especially upon TNF- α stimulation (Figure S6D), with no alterations in succinate dehydrogenase between groups (Figure S6E). These have been incorporated into our results (highlighted in red).

Third, to validate changes in fatty acid oxidation (FAO), we utilized cell-type-specific flow cytometry followed by t-distributed stochastic neighbor embedding (tSNE) analysis specifically quantify the mitochondrial transporter CPT1 α in both enteric neurons and glia and highlight even subtle changes at a high-dimensional populational level. Our analysis detected an overall higher expression of CPT1 α in enteric neurons (Figure S6F, see below) than in enteric glia (Figure S6G, see below). Moreover, the concatenated t-SNE analysis for both genotypes and conditions highlighted a visual decrease of CPT1 α ^{high} enteric neurons (red arrows, Figure S6F) in SNCA 3x cells treated with TNF, together with a visual increase of CPT1 α ^{high} enteric glia (red arrows, Figure S6G) in SNCA 3x cells treated with TNF, suggesting a potential

redistribution of FAO capacity between neurons and glia under inflammatory conditions, in line with the TNF- α -induced metabolic rewiring observed in SNCA 3x ENLs. This information is now included in our results and discussion sections (**highlighted in red**).

This combination of AATC protein validation, direct TCA enzyme activity, and CPT1A expression validation moves the conclusion beyond OXPHOS readouts to provide comprehensive functional proof for our claims of MAS disruption and subsequent metabolic alterations.

Furthermore, the first sets of figures suggest SNCA triplication itself causing mitochondrial abnormalities, however, the functional data in Figure 7 in particular do not support this, could the authors comment.

Reply: We thank the reviewer for this valuable comment. The apparent difference between the early morphological data and the Seahorse functional readout reflects the distinct aspects of mitochondrial biology captured by these assays.

In Figure 3, we assessed mitochondrial morphology using TOM20 immunostaining and observed structural alterations consistent with mitochondrial fragmentation and redistribution in SNCA triplication lines. These structural changes were corroborated by additional datasets: (i) analysis of scRNA-seq transcriptomic profiles revealed significant enrichment of mitochondrial signatures, including dysregulated expression of genes related to oxidative phosphorylation and mitochondrial organization, and (ii) proximity ligation assays demonstrated increased interactions between TOM20 and α -synuclein already in a basal level

(Figure 6C), consistent with α -synuclein-mediated perturbation of mitochondrial integrity. Together, these independent approaches strongly support the conclusion that SNCA triplication is associated with mitochondrial abnormalities.

By contrast, the Seahorse MitoStress assay in Figure 7 measures bioenergetic function at the population level. Morphological disruption and altered mitochondrial- α -synuclein interactions may not directly translate into reduced basal respiration, as cells can compensate through metabolic adaptations, or the changes may precede overt functional decline. Moreover, bulk respiration measurements at a basal level may mask heterogeneity within cell populations, such that subpopulations with impaired mitochondrial activity are diluted by cells that maintain normal function. We do note, however, there was a trend for decreased maximal respiration in SNCA 3x ENLs at a basal level (Figure 7C), although not significant ($p=0.40$).

Taken together, we interpret these results as showing that SNCA triplication leads to robust structural and molecular mitochondrial alterations, while functional consequences at the level of oxygen consumption are more subtle or context dependent. We have revised the Discussion to highlight this distinction, emphasizing that mitochondrial morphology and interaction changes may represent early hallmarks of vulnerability that precede detectable loss of bulk respiratory capacity, by specifically stating: "Deepening into the mechanism, our data indicated that already at basal levels, there is an increased interaction between α -syn and mitochondria in SNCA 3x ENLs, consistent with previous findings, and this effect was further amplified upon TNF- α stimulation".

For the CSB6 experiments do the authors show target engagement and therefore can be certain that the CSB6 treatment is having this effect via the known mechanism of action rather than off target effects, as many compounds can affect mitochondria and metabolism.

Reply: We appreciate the reviewer's suggestion on confirming target engagement and rigorously distinguishing the intended mechanism from potential off-target effects. CSB6 is a known vesicular glutamate uptake Inhibitor (VGLUT inhibitor). Our data strongly support that its effect is mediated via this known mechanism. Specifically, our updated mitofuel assays reveal a highly specific metabolic vulnerability in the SNCA 3x ENLs: TNF- α exposure significantly increases their dependency on glutamine oxidation, which is fully rescued by CSB6 (Figure 7E, see below). This rescue aligns precisely with the known VGLUT inhibitory function of CSB6, confirming mechanism-specific target engagement by modulating the α -syn inflammation-driven metabolic rewiring involving glutamate. Moreover, to address potential off-target effects, we conducted supplementary CSB6-only Seahorse mitostress experiments across both genotypes, and the results strongly argue against a non-specific mitochondrial effect. Specifically, direct OCR comparisons under the CSB6-only condition using Two-way ANOVA with Sidak post-hoc (Figures 7B-C, see below) revealed a significantly higher basal and ATP-linked OCR exclusively in the SNCA 3x ENLs compared to isogenic controls. If CSB6 were merely a general mitochondrial modulator, the enhancement would be uniform across both lines; the observation that the mitochondrial boost is preferentially pronounced in the compromised SNCA 3x system provides powerful evidence for mechanism-specific engagement. Furthermore, CSB6 demonstrated a highly specific rescue capacity by restoring the TNF-mediated impairments in proton leak OCR exclusively in the SNCA 3x line (Figure

7C, shown below). This genotype and context-dependent reversal of TNF vulnerability, without any signs of general uncoupling or inhibition in Iso controls, confirms that CSB6 is modulating the underlying α -syn inflammation-driven metabolic pathway rather than causing a generalized mitochondrial perturbation.

Reviewer #4 (Remarks to the Author):

Ghirotto and colleagues present an iPSC-derived Enteric Neuron System of the alpha-synuclein triplication as a means to investigate the pathogenesis underlying the gut-to-brain hypothesis of PD and potential other synucleinopathies. Their work identifies that soluble TNF-alpha treatment exacerbates mitochondrial dysfunction in enteric neurons. While mitochondrial dysfunction in synucleinopathies is well established, there have been no reports studying this phenomenon in the enteric nervous system.

Reply: We thank the reviewer for their recognition of the novelty and ambition of our work. We agree that while mitochondrial dysfunction is a well-established mechanism in PD pathology, especially in the central nervous system, investigating this phenomenon within the iPSC-derived enteric neural lineages of SNCA triplication patients is a first-in-field approach. By

identifying the malate-aspartate shuttle (MAS) deficit as a unique metabolic vulnerability triggered by TNF- α in these SNCA 3x ENLs, our work provides a novel and functional demonstration of a critical, early pathogenic mechanism operating within the enteric nervous system. We have carefully reviewed all suggestions provided by the reviewers, and we believe that addressing them has significantly strengthened the rigor and impact of our contribution.

I have the following comments:

1. Given the observed cellular heterogeneity in the culture (scRNAseq), the cell types could be further refined.

a) More fine-level cell typing exists within the Drokhyansky et al. study. The authors should adequately cell type their cells to the finer levels shown in the manuscript.

Reply: We thank the reviewer for this helpful suggestion, which is also in accordance with other reviewer's suggestions. To improve interpretability, we annotated the subclusters in Fig. 2A and 2E (see below). The annotation was performed by integrating our dataset with the enteric neuron and glia reference atlas of Drokhyansky *et al.* (Cell, 2020) using SingleR (as shown before in Suppl Fig 2), which assigns cell-type labels based on transcriptomic similarity to the reference. For each subcluster, we determined the majority SingleR label and then confirmed the annotation with cluster-specific marker genes identified by differential expression analysis. In cases where the same label appeared across multiple clusters, we appended the top discriminating marker gene to generate unique and biologically meaningful designations. A supplementary table S2 with all the markers per subcluster is also provided, and the top markers per subcluster are displayed in Supplementary Fig S2A (enteric neurons) and S2B (enteric glia). We also incorporated this into the results section and provided an integrative interpretation with all the scRNAseq data in the figure (**highlighted in gold**).

b) The authors should show the marker genes of the ENs in scRNAseq clusters, such as the markers that they used for their qPCRs (e.g., HOXB3 and TUBB3 are missing from Figure S2A).

Reply: We thank the reviewer for this comment. We wanted to clarify that Figure S2A refers to the top 3 marker genes per subcluster, which do not necessarily include HOXB3 and TUBB3. Top markers per enteric neuronal and glial subclusters are depicted in supplementary

table S2. We have now, however, updated our dotplot in Figure 1F to show the expression of *HOXB3* and *PHOX2B*, which were not previously included there. As shown below in the updated dotplot and additional UMAP plots, our enteric neurons express all the markers mentioned by the reviewer.

c) The authors should stain their cells according to the *HOXB3*, *PHOX2B*, and *TUBB3* to show the percentage of cells expressing those proteins in both cell lines.

Reply: We appreciate the reviewer's suggestion. While *HOXB3*, *PHOX2B*, and *TUBB3* are valuable markers for enteric neural progenitors, we have opted to use HuC/D and GFAP staining at the end of differentiation. These markers are more specific for mature enteric neurons and glia, which are the primary cell types analyzed in our study. Additionally, our cultures form 3D-like structures, making accurate quantification by immunocytochemistry challenging. Instead, to complement this and follow the reviewer's suggestion, we have now performed comprehensive flow cytometry analyzes utilizing CD24 expression levels [Windster et al. 2023, EMBO Rep, doi:10.15252/embr.202255789] (Figure S5F, see below) to separate and quantify enteric neurons and glia. This robust approach allowed us to determine that there were no changes in enteric neurons or enteric glia proportions and numbers, neither induced by the different genotype (SNCA 3x) or by the treatment (TNF- α) (Figure 4K, see below). This information has been included in our results section (highlighted in yellow).

2. According to Fig. 1E, SNCA 3x cultures have less enteric neuron proportions. a) Is this something reproducible? b) Is this something technical, or do neuronal populations drop out from the SNCA 3x cultures? In any case, the authors should address how this heterogeneity drives the rest of the conclusions that rely on these bulk assays of these populations. For example, in Figure 5D, is this analysis valid if there are fewer neurons in the SNCA 3x samples?

Reply: We thank the reviewer for this insightful comment. In our original submission, we reported a significant reduction in enteric neurons in SNCA 3x ENL cultures at d70. Following feedback from other reviewers, we reanalyzed the data in Figure 1E (see below) using the same rigorous compositional criteria applied to the enteric neuron and glia subclusters (absolute $\log_2FC > 1$ and $FDR < 0.05$, as detailed in Figure 2C and 2G). With the application of this more stringent statistical threshold, the observed reduction in enteric neurons is no longer statistically significant (Figure 1E, see below). This revised finding is consistent with our qPCR analysis of bulk neuronal markers (*TUBB3*, *PHOX2B*, *ELAVL4*, *HOXB3*; Figure S1B), which showed no difference between Iso and SNCA 3x cultures. Moreover, this is consistent with the finding just reported above (Figure 4K), in which using flow cytometry quantification we detect no changes in enteric neuron and glia proportions induced by genotype or treatment. This eliminates the concerns raised by the reviewer regarding the subsequent results.

3. Does treatment of TNF1a induce cell death in the SNCA 3x and/or Iso?

Reply: This is an important question, which we included in Fig. 4D: Treatment for 24h with TNF- α at the used concentration (100ng/mL) does not alter the viability of Isogenic and SNCA 3x ENLs, as measured by Zombie NIR viability staining dye with Flow Cytometry assays (see below).

4. It would be interesting to compare the phenotypes of the SNCA 3x neuronal dysfunction with existing studies that investigate similar questions of the SNCA 3x lines but in other neuronal types: (Patikas et al. Neuroscience Research 2023; Jin et al. Science Advances 2024). Comparing the phenotypes of midbrain and cortical organoids could highlight common modules.

Reply: We thank the reviewer for this excellent suggestion. To assess how SNCA 3x-associated neuronal dysfunction compares across different central nervous system contexts, we employed SingleR to map our iPSC-ENL scRNA-seq data onto the abovementioned published SNCA 3x organoid signatures from midbrain (Patikas et al.) and cortical (Jin et al.) studies. Interestingly, the enteric neurons in our model showed primary alignment with the dopaminergic neuron 2 (oDAN2) cluster from Patikas et al. and the inhibitory neuron (INs) cluster from Jin et al. (Figures S3C-D, S3F-G, see below). Integrated enrichment analysis across these cross-study neuronal signatures revealed commonly dysregulated pathways in our SNCA 3x ENLs focusing predominantly on synaptic function and cholesterol metabolism (Figures S3E, S3H, see below). Furthermore, taking advantage of the glial component of the Jin et al. model, our enteric glia demonstrated a strong match to their Astrocyte (AS) cluster (Figures S3I-J, see below). Enrichment analysis of this AS signature confirmed shared metabolic and mitochondrial dysfunction modules in our enteric glia, including processes related to acetyl-CoA metabolism, regulation of glucose and lipid metabolism, and cellular response to oxygen levels. These findings are highly consistent with the specific phenotypes observed in our enteric glial subclusters (Figure S3K, see below). This information has been incorporated into our results and discussion (highlighted in dark green).

Figure S3C-K

5. Since the whole manuscript builds upon the notion that asyn pathology spreads from the gut to the brain, the authors should consider testing asyn fibrils and *tnf1a* to check whether this pathology can be recapitulated in the Iso cell lines.

Reply: We thank the reviewer for this interesting suggestion. To clarify, our research question was how the PD-associated protein α -syn contributes to ENS dysfunction - and whether this is impacted by inflammation. Our work is specifically designed to examine the effects of endogenous SNCA overexpression. This rationale is further supported by strong expression of α -syn in IBD tissue (Fig 8) and examined in detail in human iPSC-derived enteric neurons and glia, providing a patient-relevant model of physiological expression. Introducing fibrils is an exogenous perturbation and requires different experimental approaches, including long-term culture, aggregation assays, and potentially in vivo validation. While studying alpha-synuclein fibril propagation in combination with TNF- α could provide additional valuable insights, this question extends beyond the focus of the current study. We have however added to the discussion that examining the spread of alpha-synuclein pathology is an important direction for future work as a final sentence stating: "Furthermore, further investigations on how this metabolic vulnerability impacts the spread and propagation of α -syn pathology represents a crucial next step toward fully understanding disease progression especially in the context of body-first PD".

6. Cell-to-Cell communication. The way the data is presented, neurons and glia appear to be two separate CellChat analyses. Does this assume that the neurons do not communicate with glial cells? If so, why?

Reply: We thank the reviewer for raising this important point. In our analyses, we first compared neurons, glia, and proliferating cells at the general cluster level, which captures broad cell-type-specific communication patterns, including interactions between neurons and glia (Fig S1 F-I). Subsequent analyses were performed separately within the enteric neuron and glia subclusters to examine more detailed, subtype-specific interactions (Fig S4 C-J). Therefore, our workflow does not assume that neurons and glia do not communicate; rather, the separate analyses at the subcluster level were designed to resolve signaling within each cell type, while inter-cell-type communication is captured in the broader, general cluster analysis.

Moreover, in the revised version, we streamlined our CellChat analysis to exclusively highlight LR pairs that are most mechanistically relevant and pronounced within the SNCA 3x line. This deeper look revealed an enrichment in VEGFA/B-VEGFR1 communication (Figure S1I, see below), known for governing cell survival and vascularity, which dictates how enteric neurons interact with glia. Crucially, we identified the EGF-EGFR pair. This interaction, typically linked to intestinal mucosal healing, is used by vulnerable glia to signal specifically to glial neighbors engaged in mitochondrial protein import, thereby forming a direct intercellular mechanism aimed at mitigating structural mitochondrial stress (Figure S4J, see below). These crucial LR interactions, along with the appearance of *de novo* BMP signaling in enteric neurons (which relates to neurite fasciculation and orientation, Figure S4F, see below), serve to contextualize the ENS vulnerability. The overall signaling data demonstrates that α -syn accumulation drives a fundamental reshaping of the cellular communication network toward survival and neuroinflammatory pathways. These data provide molecular support for the proposed compensatory hypothesis and explains the subsequent enhanced metabolic collapse observed under inflammatory conditions. We have incorporated these specific results into the revised manuscript and clarified better the CellChat data interpretation for the readers (results and discussion, highlighted in rosa).

Figure S1I

Figure S4F

Figure S4J

7. L465: "iPSC-ENLs makes it challenging to separate mechanistic effects in a populational level". What does populational level mean?

Reply: We thank the reviewer for pointing this out. By “populational level,” we mean that in mixed iPSC-derived ENL cultures it is challenging to distinguish which observed effects originate from enteric neurons versus glial cells when analyzing aggregate measurements across the population. We have now reformulated the sentence in the limitations section to make it clearer for the readers: “While the iPSC derived ENL broadly recapitulates the human ENS, the presence of mixed neuronal and glial cells in iPSC-ENLs makes it challenging to separate mechanistic effects between these cellular populations”.

8. Where is the reported pathology owed? SNCA 3x has fewer neurons at Day 70, so does this signature come from the neurons or the glia populations?

Reply: We thank the reviewer for bringing up this important point. First, as already addressed above, we reanalyzed the data in Figure 1E, using the same rigorous compositional criteria applied to the enteric neuron and glia subclusters (absolute $\log_2FC > 1$ and $FDR < 0.05$, as detailed in Figure 2C and 2G). With the application of this more stringent statistical threshold, the observed reduction in enteric neurons is no longer statistically significant: This was also validated, as discussed, by flow cytometry assays showing no differences in the numbers of enteric neurons and glia between genotypes and conditions. That taken into account, the affirmation that SNCA 3x has fewer neurons at day 70 no longer is true. To address the reviewer’s question, we took advantage of our previously mentioned flow cytometry pipeline and analyzed the protein levels of α -syn in enteric neurons and glial cells (Figure 4J, shown below). Our data demonstrated that TNF- α significantly unmasked increased α -syn levels

both in enteric neurons and glial cells, indicating that the pathology affects both cell types. This has been added to our results and discussion sections (highlighted in light green).

9. L304: Does asyn interact with TOM20 in Iso cell lines?

Reply: Yes, α -syn interacts with TOM20 already in the Iso cell lines as well, as depicted in Fig 6C, but this interaction is higher in the SNCA 3x lines at a basal level. The addition of TNF increases significantly this interaction also in the Iso lines, however the effect is more pronounced in the SNCA 3x lines, with a significantly higher interaction detected in SNCA 3x ENLs upon TNF exposure in the PLA experiment compared to the Iso group. We have now also acknowledged in the results section (highlighted in blue) the increase in PLA signal in Iso ENLs upon TNF treatment, as requested by another reviewer.

10. L320: Fig 6C appears to have incorrect statistics; Poisson events (data should be transformed as $\log(\text{counts} + 1)$) should be used prior to statistical testing. A similar problem might be present in the Fig. 1E.

Reply: We thank the reviewer for this valuable suggestion All PLA data were transformed as $\log(\text{counts}+1)$ and then subjected to statistical testings as suggested also by another reviewer. The results remain similar but there is a lot of improvement in data variability. The updated figure 6C is shown below.

11. L218: "This suggests that SNCA 3x enteric glial cells compensate for neuronal stress by enhancing mitochondrial support.". Not sure how this is supported by the data.

Reply: This sentence was excluded in the revised version. As already mentioned above, we reformulated the whole CellChat results section to focus LR pairs that are most mechanically relevant and pronounced within the SNCA 3x line (results and discussion sections, **highlighted in rosa**) and any mentions pointing to a compensatory mechanism in enteric glial cells were framed as hypothetical.

12. Figure S2A: HOXB5 is shown twice.

Reply: Thank you for the reviewer for pointing this out. We have now revised and corrected the heatmap, as shown below, to show only unique genes for each subcluster.

Reviewer #1 (Remarks to the Author):

The authors have fully addressed my previous concerns and the conclusions of the paper have been strengthened by the additional data and analyses.

Reply: We thank the reviewer for all their valuable suggestions and for the positive feedback on our revised manuscript.

Reviewer #2 (Remarks to the Author):

The authors have adequately addressed all my concerns and comments

Reply: We thank the reviewer for their thoughtful comments and for the positive assessment of our revised manuscript.

Reviewer #3 (Remarks to the Author):

The authors have thoroughly addressed the reviewers comments, taking into consideration the comments, adding additional experimental work, detail, changing interpretation of results and overall enhancing the manuscript.

Reply: We appreciate the reviewer's constructive feedback and are grateful for their positive evaluation of our revised manuscript.

Reviewer #4 (Remarks to the Author):

In their rebuttal, the authors have performed additional analyses that substantially address most comments and have improved the quality of their manuscript, thereby strengthening the impact of their findings.

Reply: We are grateful to the reviewer for their careful consideration of our revision and for the insightful comments on our updated manuscript. We have now considered the reviewer's additional comments and all the changes in the manuscript have been highlighted in **yellow**.

A major concern remains that the scRNAseq data show significant shifts in their neuronal population, which the authors deemed non-significant, citing a log₂ fold change threshold of 1. This threshold is quite stringent for population shifts and, in my opinion, arbitrary. The authors should assess the ramifications of an approximately 30% reduction in neurons (log₂ fold change around -0.5, as shown in the figure for the neurons) observed in their Figure 1E in the rest of the analysis. My impression of a 30% differential abundance is substantial, and in a different research context, it would have been "neuroprotective". While variability between differentiated iPSC lines is a well-known problem (Jerber et al., 2021, Nature Genetics) and is well beyond the scope of this manuscript, the authors should at least address its ramifications in their data interpretation. At a minimum, authors should perform an appropriate analysis to determine which cells are dropping out, using a suitable method such as Milo (Dann et al. Nat Biotech 2022). Then, using known TNF1a pathway genes, they should determine whether these populations express them. Additionally, the limitations sections could be expanded appropriately.

Reply: We thank the reviewer for highlighting this important point. To address this concern, we first outline our initial analytical strategy and the rationale behind it: at the major cell-type level (that includes the complete scRNA seq data with enteric neurons, glia, proliferating cells; Fig. 1D–E), we applied the established scProportionTest framework [Miller et al. 2021, Cancer Res,

doi:10.1158/0008-5472.CAN-20-3562] using an absolute \log_2FC threshold of 1, as previously described. This more stringent threshold was selected to identify differences that are robust and biologically meaningful. As suggested by the reviewer, re-analysis using a more permissive threshold (absolute $\log_2FC = 0.58$, corresponding to a fold-change of 1.5) likewise detected no significant differences in population abundance (see below). These findings are consistent with our qPCR analyses of neuronal markers (*TUBB3*, *PHOX2B*, *ELAVL4*, *HOXB3*; Fig. 1B, Fig. S1B) across multiple differentiation timepoints (d6, d40, d70), as well as our flow cytometry quantifications of neuron and glia frequencies under both basal and TNF-treated conditions (Fig. 4K).

We next implemented Milo, as recommended by the reviewer, setting a more relaxed spatial FDR threshold of 0.2 for all analysis of neighbourhood-level changes, given our limited number of biological replicates used in this study, which reduces MiloR statistical power. In contrast to scProportionTest, which detects changes across discrete cell populations or clusters, MiloR is designed to identify localized shifts in continuous transcriptional states, as often observed in developing or dynamic populations. When applied to the overall dataset (Enteric neurons, glia and proliferating cells), Milo also detected no significant neighbourhood-level perturbations (Supplementary Fig. S1D and Table S5), indicating that global average cell-type composition is preserved (see below)

This suggests that the analysis at the whole-population level can mask subtype-specific shifts by averaging over heterogeneous cell states.

To ensure that potential changes within smaller cell subpopulations were not being masked due to averaging effects across all cell populations, we had already performed differential abundance analysis on each enteric neuron and glial population individually using scProportion test, as presented in the manuscript (Figures 2C and 2G, respectively; see below). In this more focused analysis, we detected differential abundance in specific neuronal and glial subsets between Iso and SNCA 3x conditions, consistent with the reviewer’s suggestion, even with the more stringent absolute log2FC threshold of 1.

Thus, finer-grained interrogation at the level of individual populations reveals subtype-specific differences.

To further incorporate the reviewer’s suggestion, we confirmed these results by performing MiloR analysis on each enteric neuronal and glial population individually. Below, we present the differential abundance results obtained from the MiloR analysis. MiloR analysis of the glial subset revealed no significant spatial changes (Supplementary Fig. S2J and Table S5), likely

because glial cells occupy discrete, stable states rather than forming continuous transcriptional neighborhoods. In contrast, MiloR applied to the neuronal subset identified localized neighbourhood-level changes with lower spatial FDR values (minimum = 0.15; Supplementary Fig. S2I and Supplementary Table S5). Importantly, these neuronal neighbourhoods correspond precisely to the same subclusters identified as decreased (e.g., subcluster 2) or increased (e.g., subcluster 13) by the scProportionTest analysis (Fig. 2C), demonstrating concordance between methods and indicating that SNCA 3x induces spatially restricted reorganization specifically within neurons rather than broad population loss. The MiloR plots for neural and glial subclusters are shown below.

To determine whether these neuronal and glial subclusters are relevant to the pathways investigated in our study, we examined basal expression of TNF-related receptors, NF- κ B

components, synaptic markers, and glial immune-response genes (new Supplementary Fig. S5). Because the scRNAseq data were generated under unstimulated basal conditions, these signatures represent intrinsic genotype-dependent transcriptional states. Across nearly all subclusters, SNCA 3x exhibited higher baseline expression of *TNFRSF1A*, *TNFRSF1B*, *NFKBIA*, *NFKB1*, and *RELA*, indicating that both neuronal and glial populations retain, and in many cases upregulate, the molecular machinery required for TNF responsiveness. Differentially abundant neuronal subclusters also showed reduced expression of synaptic and neuronal identity markers such as *DSCAM* and *GABRB3*, whereas glial subclusters showed increased expression of immune-associated genes including *MIF* and *HLA-B*. These genes were identified in our unbiased differential expression analysis (Supplementary Table S4), and their expression patterns are now visualized in Supplementary Fig. S5 (see below).

Together, Milo, scProportionTest, and differential expression analyses converge to show that SNCA 3x induces localized neuronal subcluster reorganization within an otherwise stable culture composition, and that the affected subclusters express genes associated with TNF signaling and exhibit transcriptional signatures consistent with the metabolic vulnerabilities observed in our functional assays. We added this information to the results and discussion sections of the manuscript. Furthermore, we have expanded the Limitations section to discuss these considerations, including the reduced statistical power of neighbourhood-based methods in datasets with limited biological replicates and the known variability of iPSC-derived differentiation [Jerber et al., 2021, Nature Genetics].

Furthermore, their CD24+ population does not reflect the trends observed in the data and, in my interpretation, may not be significant even at a log2 fold change > 0 threshold,

which raises concerns about how much the flow cytometry experiment can be trusted. Additionally, can the authors provide a CD24 plot to assess how well this marker distinguishes glial from non-glial cells?

Reply: We thank the reviewer for their insights on the use of CD24 as a marker for separating enteric neurons versus glial populations. In this sense, we would like to emphasize that this strategy builds on a recently published and well-validated approach in human gut tissue. In that work, the flow cytometry protocol was extensively characterized and validated using canonical markers of enteric neurons (e.g., ELAVL4) and glia (e.g., S100B) [Windster et al., EMBO Reports, 2023, <https://doi.org/10.15252/embr.202255789>]. Moreover, our scRNAseq data (shown below) points out to an evident higher expression of CD24 in the enteric neurons cluster. We also provide a ridgeline plot showing that the higher expression of CD24 in enteric neurons is statistically significant when compared to enteric glial cells ($\log_2FC = 0.74$ and $p = 2.2e^{-7}$). These plots have been added in Supplementary Figure S6G and H, respectively and we now mention in the results section that the CD24-based staining to separate enteric neurons and glia is supported by our scRNA-seq data to make it clear for the reader. Finally, we provide raw flow cytometry plots below showing an unstained control followed by contour plots highlighting the separation between the neurons and glial populations, which was based on the CD24 levels (also shown as a heatmap of CD24 expression on the right side, which is included in Supplementary Figure S6F). Altogether, these data clearly demonstrate that CD24 is a reliable marker for distinguishing enteric neurons from glial cells in our iPSC-ENL model. We added this information to the results section of the manuscript.

At a more granular level, many neuronal subpopulations are differentially abundant (Figure 2C, 2G), and the authors correctly term these "significant" and provide additional interpretations. Still, this potential could be a differentiation artifact, but at least here it is not glossed over. While this may also be due to clonal variability, the authors should assess genes central to their question. In a similar vein to the above, they should check which of those cell types express TNF-alpha genes and could affect the differential responses observed in their bulk assays.

Reply: We thank the reviewer for this insightful point. To determine whether the differentially abundant subclusters express genes central to our study, we examined basal expression of TNF-related receptors, downstream NF- κ B components, synaptic markers, and glial immune-response genes across all neuronal and glial subpopulations (new Supplementary Fig. S5). Because the scRNAseq dataset was generated under unstimulated conditions, these analyses capture intrinsic genotype-dependent transcriptional states rather than acute TNF responses. Across nearly all subclusters, SNCA 3x cells show higher baseline expression of *TNFRSF1A*, *TNFRSF1B*, *NFKBIA*, *NFKB1*, and *RELA*, indicating that these populations retain the molecular machinery required for TNF responsiveness. Thus, the observed proportional shifts do not reflect loss of TNF-competent cell types.

In the neuronal lineage, the subclusters that differ in abundance between genotypes (e.g. subcluster 2) also display reduced expression of synaptic and neuronal identity markers such as *DSCAM* and *GABRB3*, which were identified among the significantly differentially expressed genes between Iso and SNCA 3x. These signatures are consistent with the selective vulnerabilities demonstrated in our functional metabolic assays and provide further support for intrinsic susceptibility in the SNCA 3x background. In contrast, enteric glial subclusters exhibit increased expression of immune- and stress-associated genes including *MIF* and *HLA-B*, likewise identified in our Iso vs. SNCA 3x differentially expressed genes, a pattern that mirrors their elevated TNF-pathway gene expression but occurs without significant differential abundance detected by Milo neighbourhood analysis. Together, these analyses confirm that the subclusters identified as differentially abundant remain transcriptionally relevant to the pathways studied, maintain the capacity to engage TNF signaling, and display

basal-state profiles that align with the metabolic vulnerabilities characterized in the bulk assays. We added this information to the results and discussion sections of the manuscript.

Figure S5

Reviewer #4 (Remarks to the Author):

In their rebuttal, the authors have performed additional analyses and provided additional plots to support their claims about the underlying cell-type abundance in the different ENS cultures.

Reply: We thank Reviewer 4 for the additional feedback. We have addressed the final remaining points below, ensuring that our analytical choices are transparent and align with the technical specifications of the tools utilized. Changes have been highlighted **in yellow** throughout the manuscript.

Here are some comments I have:

1) If I understand correctly, scProportion lets the authors set a threshold. This threshold is arbitrary and is used as a null hypothesis. The authors cite a log2fold change threshold of 1 (a doubling of the population). In the second, they did a more permissive threshold of 0.58 (a 50% increase in population). The second threshold, if I understand the error bars, falls within the scProportion confidence intervals. I would argue that the most rational null is that there is no change in the population, which corresponds to log2fc=0. I understand that the authors are not concerned with this. However, this needs to be explicitly written in the main text: "with a null hypothesis of 50%..."

Reply: We thank the reviewer for this point. While log2FC = 0 represents a statistical baseline, it is not appropriate for high-throughput single-cell abundance testing, where sampling variability and technical noise can produce small apparent differences in cell abundance. Testing against a threshold of zero would inflate false-positive findings by classifying minor stochastic fluctuations as biologically meaningful changes. As recommended for scProportion (Miller SA et al., 2021, Cancer Res), a log2FC of 1 is the standard threshold for distinguishing true biological shifts from technical noise. The more permissive threshold (log2FC = 0.58) was used in our previous rebuttal to demonstrate that even with a less stringent analysis, we don't see a shift in abundance at the general cluster level (Glia, enteric neurons and proliferating cells), but we clarify that throughout the manuscript, we kept with the log2FC = 1 threshold for scProportion test, as published (Miller SA et al., 2021, Cancer Res). We have accordingly revised the main text.

2) I don't know why the Milo plots look like this, but having lfc of 0 everywhere (Figure S1D) does not pass the smell test. The authors should double-check that the analysis was conducted properly and that the figure is displayed correctly. In the general clusters Supp Table, deposited log-fold changes range from -8.3 to 7.3. There are similar symptoms in the glia populations.

Reply: We have carefully checked the Milo analysis and confirm that the figure is displayed correctly and reflects the intended behavior of the Milo framework. The reviewer's concern that the apparent absence of color-coded log2FC values relates to the way Milo visualizes significantly differential abundance results. In Milo, neighborhoods are assigned a color-coded log2FC value only when they pass the spatially corrected p-value (Spatial FDR) threshold (0.2 in our case). Neighborhoods that do not meet this threshold are rendered in grey and displayed with log2FC = 0 for visualization purposes. Consequently, the grey regions observed in Fig. S1D (and similarly in Fig. S2J) do not indicate that the estimated log2FC values are uniformly zero, but rather that these neighborhoods do not reach statistical significance after spatial multiple-testing correction. Although the corresponding summary tables report a range of raw log2FC estimates (-8.3 to 7.3), these values are associated with neighborhoods that fail to meet the Spatial FDR criterion and are therefore intentionally suppressed in the visualization

to prevent over-interpretation of non-significant variation. This behavior is consistent with Milo's design and is similarly observed in the glial population analyses. For a comprehensive description of Milo's implementation, we refer the reviewer to the original publication (https://www.bioconductor.org/packages/release/bioc/vignettes/miloR/inst/doc/milo_gastrulation.html) and the codes, which explicitly state: "The neighbourhoods **displaying significant DA** are colored by their log-Fold Change." Our analysis was conducted in accordance with the described methodology.

3) By looking at the Milo tables, it appears that the "permissive" cutoff of 0.2 was selected so that neurons pass this cutoff. If the authors, as they suggest, think their study is not properly powered to support this analysis, perhaps this analysis should be omitted from the manuscript.

Reply: As suggested by the reviewer, we performed a Milo neighborhood-based analysis to investigate potential subtle or localized differences in cellular abundance that may not be captured by global proportion analyses. Given the sample size and design, these results are considered exploratory. A permissive Spatial FDR threshold of 0.2 was applied to detect potential trends. In the dataset, the minimum Spatial FDR for neuronal neighborhoods was 0.14; applying more stringent thresholds (e.g., 0.05 or 0.1) would have resulted in no neighborhoods reaching significance across any population. The threshold was applied consistently across all cell types to ensure uniform assessment.

Under these exploratory criteria, the analysis suggested localized changes within enteric neuronal neighborhoods, whereas glial populations did not display comparable neighborhood-level signals. As suggested by the reviewer, these results are presented exclusively in the Supplementary Figures, with their exploratory nature and limitations clearly indicated in the manuscript.

4) The flow cytometry plots are helpful. The authors measure ~80% of their cells. The scRNAseq data support the existence of a CD24high/low plot. However, it is not clear how the high- and low-population groups map to the data. While I can definitely infer the CD24-high from the contour lines. However, judging from the contour lines, the CD24-low peak is not as clear as in the cited paper (Windster et al., Fig. 2a). Moreover, the authors use a negative-stain control to set a negative-low cutoff. This cutoff is very close to the peak of cells absent from the main data, as indicated by the contour lines. Could this peak (which is absent in the unstained control) correspond to CD24-low cells? Can the authors provide a justification for this? If the general abundance analysis is inconclusive, which is quite challenging in my opinion at this stage, I suggest expanding the limitation section appropriately.

Reply: We thank the reviewer for this observation, which allows us to clarify the technical basis of our gating. We have updated our visualization to explicitly label the double-negative population as CD56-/CD24- (20.8%, highlighted by arrows in the figure below). With this addition, our gated populations now account for 98.5% of the total cells (Neurons: 29.8%; Glia: 47.9%; Double-Negative: 20.8%).

Regarding the "peak" in the lower-left quadrant, we would like to provide a technical clarification regarding normalized density visualization. In flow cytometry, populations are defined by signal intensity (X-axis position) relative to negative controls. As shown in our side-by-side comparison with the Unstained Control, 94.2% of control cells reside in that exact quadrant. The population the reviewer identifies in the stained sample occupies the identical X-axis coordinate as the baseline noise of the unstained control.

The visual prominence of this peak in the stained sample is a common artifact of density-normalized contour plots: because the double-negative population is smaller (20.8%) than the unstained control (94.2%), it appears as a more "distinct" cluster, even though its fluorescence intensity has not shifted. Critically, this peak falls at the 10^0 coordinate, corresponding to baseline electronic noise and autofluorescence rather than a biological CD24-low signal.

Finally, while we acknowledge the visual profiles in Windster et al., that study utilized primary gut tissue. Given the distinct maturation and autofluorescence profiles of our iPSC-derived model, we have prioritized our internal negative-stain controls, the objective gold standard, to define our gates. This ensures our analysis remains consistent with established FACS recommendations.

RESPONSE LETTER

Reviewer #4 (Remarks to the Author):

Thank you for the clarifications

Reply: We thank the reviewer for all their suggestions in our manuscript.